# DYNAMIC SIMILARITY GRAPH CONSTRUCTION WITH KERNEL DENSITY ESTIMATION

## ABSTRACT

In the kernel density estimation (KDE) problem, we are given a set $X$ of data points in $\mathbb{R}^d$, a kernel function $k : \mathbb{R}^d \times \mathbb{R}^d \to \mathbb{R}$, and a query point $\mathbf{q} \in \mathbb{R}^d$, and the objective is to quickly output an estimate of $\sum_{\mathbf{x} \in X} k(\mathbf{q}, \mathbf{x})$. In this paper, we consider KDE in the dynamic setting, and introduce a data structure that efficiently maintains the *estimates* for a set of query points as data points are added to $X$ over time. Based on this, we design a dynamic data structure that maintains a sparse approximation of the fully connected similarity graph on $X$, and develop a fast dynamic spectral clustering algorithm. We further evaluate the effectiveness of our algorithms on both synthetic and real-world datasets.

## 1 INTRODUCTION

Given a set $X = \{\mathbf{x}_1, \ldots, \mathbf{x}_n\}$ of data points, a set $Q = \{\mathbf{q}_1, \ldots, \mathbf{q}_m\}$ of query points, and a kernel function $k : \mathbb{R}^d \times \mathbb{R}^d \to \mathbb{R}_{\geqslant 0}$, the KDE problem is to quickly approximate $\mu_\mathbf{q} \triangleq \sum_{\mathbf{x}_i \in X} k(\mathbf{q}, \mathbf{x}_i)$ for every $\mathbf{q} \in Q$. As a basic question in computer science, this problem has been actively studied since the 1990s (Greengard & Strain, 1991) and has comprehensive applications in machine learning and statistics (Bakshi et al., 2023; Genovese et al., 2014; Scholkopf & Smola, 2018; Schubert et al., 2014).

In this paper we first study the KDE problem in the dynamic setting, where both the sets of data and query points change over time. The objective is to dynamically update the KDE estimates of $\mu_\mathbf{q}$ as data points are added to $X$. Building on the framework for static KDE developed by Charikar et al. (2020), our algorithm processes: (i) insertions and deletions of query points, and (ii) insertions of data points in $\varepsilon^{-2} \cdot n^{0.25+o(1)}$ time for the Gaussian kernel[1]. In particular, our algorithm maintains $(1 \pm \varepsilon)$-approximate estimates of the kernel densities for *every* query point $\mathbf{q} \in Q$ throughout the sequence of data point insertions; see Theorem 3.1 for the formal statement. To the best of our knowledge, this represents the first dynamic algorithm for the KDE problem that efficiently maintains query estimates under data point insertions.

Among its many applications, an efficient algorithm for the KDE problem on $X = Q \subset \mathbb{R}^d$ can be used to speed up the construction of a similarity graph for $X$, one of the key components used in many graph-based clustering algorithms (e.g., spectral clustering). These clustering algorithms usually have superior performance over traditional geometric clustering techniques (e.g., $k$-means) (Ng et al., 2001; von Luxburg, 2007), but in general lack a dynamic implementation. Our second contribution addresses this challenge, and designs a dynamic algorithm that maintains a similarity graph for the dataset $X$ with expected amortised update time $n^{0.25+o(1)}$ when new data points are added; see Theorem 4.1 for the formal statement. Our algorithm guarantees that, when the set $X_t$ of data points present at any time $t$ has a cluster structure, our dynamically maintained graph will have the same cluster structure as the fully connected graph on $X_t$; hence a downstream graph clustering algorithm will perform well.

Our designed algorithms are experimentally compared against several baseline algorithms on 8 datasets from different domains, and these experiments confirm the sub-linear update time proven in theory. These experiments further demonstrate that

---

[1] Our algorithm generalises to arbitrary kernel functions, with different powers of $n$ in the update time.

- our dynamic KDE algorithm scales better to large datasets than several baseline algorithms, including the fast static KDE algorithm in Charikar et al. (2020), and
- our dynamic similarity graph construction algorithm runs faster than the fully-connected and $k$-nearest neighbour similarity graph baselines, and produces comparable clustering results when applying spectral clustering.

**Related Work.** Efficient algorithms for the kernel density estimation problem in low dimensions have been known for over two decades (Gray & Moore, 2003; Greengard & Strain, 1991; Yang et al., 2003). For the regime of $d = \Omega(\log n)$, there has been some recent progress to develop sub-linear query time algorithms (Charikar et al., 2020; Charikar & Siminelakis, 2017; 2019) based on locality-sensitive hashing (Andoni & Indyk, 2008; Datar et al., 2004) and importance sampling using algorithms for computing approximate nearest neighbors (Backurs et al., 2018; Karppa et al., 2022). There has also been recent work studying the approximation of kernel similarity graphs in the static setting (Macgregor & Sun, 2023; Quanrud, 2021).

Dynamic kernel density estimation has been studied in some restricted settings. Huang et al. (2024) give a dynamic variant of the fast Gauss transform (Greengard & Strain, 1991) for low-dimensional data. Given an initial dataset $X$, Liang et al. (2022) give an efficient algorithm for maintaining a KDE data structure in which some data point $\mathbf{x}_i$ is replaced with a new point $\mathbf{z}$. In the same setting, Deng et al. (2022) present a dynamic data structure that maintains a spectral sparsifier of the kernel similarity graph, for smooth kernels.

Our work also relates to a number of works on incremental spectral clustering (Dhanjal et al., 2014; Laenen & Sun, 2024; Martin et al., 2018; Ning et al., 2007). However, these works either assume a fixed vertex set (Dhanjal et al., 2014; Martin et al., 2018; Ning et al., 2007), or are limited to handling only single edge updates (Laenen & Sun, 2024).

## 2 Preliminaries

This section lists several facts we use in the analysis, and is organised as follows: Section 2.1 gives a brief introduction to Locality Sensitive Hashing, which we apply in Section 2.2 to discuss fast algorithms for Kernel Density Estimation. We informally define an approximate similarity graph in Section 2.3.

### 2.1 Locality Sensitive Hashing

Given data $\mathbf{x}_1, \ldots, \mathbf{x}_n \in \mathbb{R}^d$, the goal of Euclidean locality sensitive hashing (LSH) is to preprocess the data in a way such that, given a query point $\mathbf{y} \in \mathbb{R}^d$, we are able to quickly recover the data points close to $\mathbf{y}$. Informally speaking, a family $\mathcal{H}$ of hash functions $H : \mathbb{R}^d \to \mathbb{Z}$ is *locality sensitive* if there are values $r \in \mathbb{R}$, $c > 1$, and $p_1 > p_2$, such that it holds for $H$ drawn at random from $\mathcal{H}$ that $\mathbb{P}[H(\mathbf{u}) = H(\mathbf{v})] \geqslant p_1$ when $\|\mathbf{u} - \mathbf{v}\| \leqslant r$, and $\mathbb{P}[H(\mathbf{u}) = H(\mathbf{v})] \leqslant p_2$ when $\|\mathbf{u} - \mathbf{v}\| \geqslant c \cdot r$. That is, the collision probability of close points is higher than that of far points. Datar et al. (2004) propose a locality sensitive hash family based on random projections, and their technique is further analysed by Andoni & Indyk (2008):

**Lemma 2.1** (Andoni & Indyk (2008)). *Let $\mathbf{p}$ and $\mathbf{q}$ be any pair of points in $\mathbb{R}^d$. Then, for any fixed $r > 0$, there exists a hash family $\mathcal{H}$ such that, if*

$$p_{\mathrm{near}} \triangleq p_1(r) \triangleq \mathbb{P}_{H \sim \mathcal{H}}[H(\mathbf{p}) = H(\mathbf{q}) \mid \|\mathbf{p} - \mathbf{q}\| \leqslant r]$$

*and*

$$p_{\mathrm{far}} \triangleq p_2(r, c) \triangleq \mathbb{P}_{H \sim \mathcal{H}}[H(\mathbf{p}) = H(\mathbf{q}) \mid \|\mathbf{p} - \mathbf{q}\| \geqslant cr]$$

*for any $c \geqslant 1$, then*

$$\rho \triangleq \frac{\log 1/p_{\mathrm{near}}}{\log 1/p_{\mathrm{far}}} \leqslant \frac{1}{c^2} + O\left(\frac{\log t}{t^{1/2}}\right),$$

*for some $t$, where $p_{\mathrm{near}} \geqslant \mathrm{e}^{-O(\sqrt{t})}$ and each evaluation takes $dt^{O(t)}$ time.*

We follow the work of Charikar et al. (2020) and use $t = \log^{2/3} n$, which results in $n^{o(1)}$ evaluation time and $\rho = \frac{1}{c^2} + o(1)$. In this case, if $c = O\left(\log^{1/7} n\right)$, then $\rho^{-1} = c^2(1 - \beta)$, for $\beta = o(1)$.

**Definition 2.2** (bucket). *For any hash function $H : \mathbb{R}^d \to \mathbb{Z}$ and $\mathbf{x} \in \mathbb{R}^d$, let $B_H(\mathbf{x})$ be the set defined by $B_H(\mathbf{x}) \triangleq \{\mathbf{x}' \mid H(\mathbf{x}) = H(\mathbf{x}')\}$; we call $B_H(\mathbf{x})$ a bucket.*

## 2.2 KERNEL DENSITY ESTIMATION

Given a set $X = \{\mathbf{x}_1, \ldots, \mathbf{x}_n\}$ of data points, a set $Q = \{\mathbf{q}_1, \ldots, \mathbf{q}_m\}$ of query points, and a kernel function $k : \mathbb{R}^d \times \mathbb{R}^d \to \mathbb{R}_{\geqslant 0}$, the KDE problem is to compute

$$\mu_{\mathbf{q}} \triangleq k(\mathbf{q}, X) \triangleq \sum_{\mathbf{x}_i \in X} k(\mathbf{q}, \mathbf{x}_i)$$

for every $\mathbf{q} \in Q$. We assume that[2] $1 \leqslant \mu_{\mathbf{q}} \leqslant n$. While a direct computation of the $m$ values for every $\mathbf{q} \in Q$ requires $mnd$ operations, there are a number of works that develop faster algorithms for approximating these $m$ quantities.

Our designed algorithms are based on the work of Charikar, Kapralov, Nouri, and Siminelakis (Charikar et al., 2020). We refer to their algorithm as CKNS, and provide a brief overview. At a high level, the CKNS algorithm is based on importance sampling and, for any query point $\mathbf{q}$, their objective is to sample a data point $\mathbf{x}_i \in X$ with probability approximately proportional to $k(\mathbf{q}, \mathbf{x}_i)$. To achieve this, they introduce the notion of *geometric weight levels* $\{L_j^{\mathbf{q}}\}_j$ defined as follows:

**Definition 2.3** (Charikar et al. (2020)). *For any query point $\mathbf{q}$, let $J_{\mu_{\mathbf{q}}} \triangleq \left\lceil \log \frac{2n}{\mu_{\mathbf{q}}} \right\rceil$, and for $j \in [J_{\mu_{\mathbf{q}}}]$, let $L_j^{\mathbf{q}} \triangleq \left\{ \mathbf{x}_i \in X : k(\mathbf{q}, \mathbf{x}_i) \in \left( 2^{-j}, 2^{-j+1} \right] \right\}$. We define the corresponding distance levels as*

$$r_j = \max_{\substack{\mathbf{x}, \mathbf{x}': \\ k(\mathbf{x}, \mathbf{x}') \in \left( 2^{-j}, 2^{-j+1} \right]}} \|\mathbf{x} - \mathbf{x}'\|$$

*for any $j \in [J_{\mu_{\mathbf{q}}}]$, and define $L_{J_{\mu_{\mathbf{q}}}+1}^{\mathbf{q}} \triangleq X \setminus \left( \bigcup_{j \in [J_{\mu_{\mathbf{q}}}]} L_j^{\mathbf{q}} \right)$.*

By definition, these $L_j^{\mathbf{q}}$'s for any query point $\mathbf{q}$ partition the data points into groups based on the kernel distances $k(\mathbf{q}, \mathbf{x}_i)$, progressing geometrically away from $\mathbf{q}$. Their key insight is that the number of data points in each level $L_j^{\mathbf{q}}$ is bounded:

**Lemma 2.4** (Charikar et al. (2020)). *It holds for any query point $\mathbf{q}$ and $j \in [J_{\mu_{\mathbf{q}}}]$ that $\left| L_j^{\mathbf{q}} \right| \leqslant 2^j \cdot \mu_{\mathbf{q}}$.*

Hence, one can sub-sample the data points with probability $1/(2^j \cdot \mu_{\mathbf{q}})$ for every $j \in [O(\log(n/\mu_{\mathbf{q}}))]$, and the sampled data points are stored in hash buckets using LSH. This data structure will allow for fast and good estimation of $\mu_{\mathbf{q}}$ for any query point $\mathbf{q}$. We further follow Charikar et al. (2020), and introduce the cost of a kernel $k$.

**Lemma 2.5** (Charikar et al. (2020)). *Assume that kernel $k$ induces weight level sets $L_j^{\mathbf{q}}$'s and corresponding distance levels $r_j$'s. Also, for any query $\mathbf{q}$, integer $i \in [J_{\mu_{\mathbf{q}}} + 1]$, and $j \in [J_{\mu_{\mathbf{q}}}]$ satisfying $i > j$, let $\mathbf{p} \in L_j^{\mathbf{q}}$ and $\mathbf{p}' \in L_i^{\mathbf{q}}$. Assuming that $\mathcal{H}$ is an Andoni-Indyk LSH family designed for near distance $r_j$ (see Lemma 2.1), the following holds for any integer $k \geqslant 1$:*

1. $\mathbb{P}_{H^* \sim \mathcal{H}^k} [H^*(\mathbf{p}) = H^*(\mathbf{q})] \geqslant p_{\text{near},j}^k$,

2. $\mathbb{P}_{H^* \sim \mathcal{H}^k} [H^*(\mathbf{p}') = H^*(\mathbf{q})] \leqslant p_{\text{near},j}^{kc^2(1-\beta)}$,

*where $c \triangleq c_{i,j} \triangleq \min \left\{ \frac{r_{i-1}}{r_j}, \log^{1/7} n \right\}$, $p_{\text{near},j} \triangleq p_1(r_j)$, and $\beta = o(1)$ from Lemma 2.1.*

**Definition 2.6** (Cost of a Kernel). *Suppose that a kernel $k$ induces distance levels $r_j$'s based on the kernel value $\mu_{\mathbf{q}}$ (see Definition 2.3). For any $j \in [J_{\mu_{\mathbf{q}}}]$ we define the cost of kernel $k$ for weight level $L_j^{\mathbf{q}}$ as*

$$\text{cost}_{\mu_{\mathbf{q}}}(k, j) \triangleq \exp_2 \left( \max_{i=j+1, \ldots, J_{\mu_{\mathbf{q}}}+1} \left\lceil \frac{i-j}{c_{i,j}^2(1-\beta)} \right\rceil \right),$$

---

[2]We make this assumption simply for the ease of our presentation, and setting $\mu_{\mathbf{q}} \geqslant \zeta$ for any constant $\zeta$ instead will not influence the asymptotic results of our work.

where $c_{i,j} \triangleq \min\left\{\frac{r_{i-1}}{r_j}, \log^{1/7} n\right\}$ and $\beta = o(1)$ *from Lemma 2.1. We define the general* cost *of a kernel $k$ as* $\mathrm{cost}(k) \triangleq \max_{\mu_{\mathbf{q}}, j \in [J_{\mu_{\mathbf{q}}}]} \mathrm{cost}_{\mu_{\mathbf{q}}}(k, j)$. *For any $j \in [J_{\mu_{\mathbf{q}}}]$ we further define*

$$k_j \triangleq -\frac{1}{\log p_{\mathrm{near},j}} \cdot \max_{i=j+1,\ldots,J_{\mu_{\mathbf{q}}}+1} \left\lceil \frac{i-j}{c_{i,j}^2(1-\beta)} \right\rceil. \tag{2.1}$$

By the assumption that $1 \leqslant \mu_{\mathbf{q}} \leqslant n$, the cost of some popular kernels such as the Gaussian kernel $k_{\mathrm{g}}$, the $t$-student kernel $k_t$, and the exponential kernel $k_e$ are $\mathrm{cost}(k_{\mathrm{g}}) = n^{(1+o(1))\frac{1}{4}}$, $\mathrm{cost}(k_t) = n^{o(1)}$, and $\mathrm{cost}(k_e) = n^{(1+o(1))\frac{4}{27}}$, respectively (Charikar et al., 2020).

## 2.3 APPROXIMATE SIMILARITY GRAPHS

Constructing a similarity graph from a set of data points is the first step of most modern clustering algorithms. For any set $X = \{\mathbf{x}_1, \ldots, \mathbf{x}_n\}$ of data points in $\mathbb{R}^d$ and kernel function $k : \mathbb{R}^d \times \mathbb{R}^d \to \mathbb{R}_{\geqslant 0}$, a similarity graph $F = (V, E, w)$ from $X$ can be constructed as follows: each $\mathbf{x}_i \in X$ is a vertex in $F$, and every pair of vertices $\mathbf{x}_i$ and $\mathbf{x}_j$ is connected by an edge with weight $w(\mathbf{x}_i, \mathbf{x}_j) = k(\mathbf{x}_i, \mathbf{x}_j)$. While this graph $F$ has $\Theta(n^2)$ edges by definition, we can construct in $\widetilde{O}(n)$ time a sparse graph $G$ with $\widetilde{O}(n)$ edges such that (i) every cluster in $F$ has low conductance in $G$, and (ii) the eigenvalue gaps of the normalised Laplacian matrices of $F$ and $G$ are approximately the same (Macgregor & Sun, 2023); these two conditions ensure that a typical clustering algorithm on $F$ and $G$ returns approximately the same result. We call such a sparse graph $G$ an *approximate similarity graph*, and refer the reader to Section A in the appendix for its formal definition.

## 2.4 CONVENTION & ASSUMPTION

For ease of presentation, for any set $X \subset \mathbb{R}^d$ and $\mathbf{z} \in \mathbb{R}^d$, we always use $X \cup \mathbf{z}$ and $X \setminus \mathbf{z}$ to represent $X \cup \{\mathbf{z}\}$ and $X \setminus \{\mathbf{z}\}$. For a similarity graph $G$ constructed for any set $X = \{\mathbf{x}_1, \ldots, \mathbf{x}_n\} \subset \mathbb{R}^d$, we use $\mathbf{x}_i$ to represent both the point in $\mathbb{R}^d$ and the corresponding vertex in $G$, as long as the underlying meaning of $\mathbf{x}_i$ is clear from context. We use $(\mathbf{x}_i, \mathbf{x}_j)$ to represent an edge with $\mathbf{x}_i$ and $\mathbf{x}_j$ as the endpoints, and the graphs studied in our paper are always undirected. We use $\widetilde{O}(n)$ to represent $O(n \cdot \log^c n)$ for some constant $c$. The $\log$ operator takes the base 2.

**Assumption 2.1.** *Let $n_1 = |X|$ denote the number of data points at initialisation. We assume that, if $X_t \subset \mathbb{R}^d$ represents the set of data points present after $t$ updates, then $|X_t| \leqslant n_1^\gamma$ for constant $\gamma > 0$. Moreover, based on the JL Lemma (Johnson, 1984), we always assume that $d = O(\log |X_t|)$, and hence our work ignores the dependency on $d$ in the algorithms' runtime.*

# 3 DYNAMIC KERNEL DENSITY ESTIMATION

In this section we design a data structure to dynamically maintain KDE estimates as new data and query points are added or removed over time. Our data structure supports $\mathrm{INITIALISE}(X, Q, \varepsilon)$, which creates a hash data structure for the KDE estimates based on $X$, and it supports operations for dynamically maintaining the data and query point sets as well as the corresponding estimates. The main components used in updating the data structure and their performance are as follows:

**Theorem 3.1** (**Main Result 1**). *Let $k$ be a kernel function with $\mathrm{cost}(k)$ as defined in Definition 2.6, and $X \subset \mathbb{R}^d$ a set of $n$ data points updated through data point insertions. Assuming $Q = \emptyset$ initially[3], the performance of the procedures in Algorithm 1 is as follows:*

- *Initialisation:* $\mathrm{INITIALISE}(X, \emptyset, \varepsilon)$ *creates a hash data structure for the KDE, and runs in time $\varepsilon^{-2} \cdot n_1^{1+o(1)} \cdot \mathrm{cost}(k)$, where $n_1$ is the number of data points at initialisation.*

- *Query Point Updates: For every query point insertion $Q \leftarrow Q \cup \mathbf{q}$ or deletion $Q \leftarrow Q \setminus \mathbf{q}$, $\mathrm{ADDQUERYPOINT}(\mathbf{q})$ and $\mathrm{DELETEQUERYPOINT}(\mathbf{q})$ update the corresponding sets and*

---

[3]When $Q \neq \emptyset$ with $|Q| \triangleq m_1$, we have an additional additive factor of $m_1 \cdot \varepsilon^{-2} \cdot n_1^{o(1)} \cdot \mathrm{cost}(k)$ and $m_1 \cdot \varepsilon^{-2} \cdot n^{o(1)} \cdot \mathrm{cost}(k)$ in the running time of the initialisation and data point update steps respectively.

*data structures. Moreover, ADDQUERYPOINT($\mathbf{q}$) returns $\hat{\mu}_{\mathbf{q}}$ that achieves a $(1 \pm \varepsilon)$-multiplicative factor approximation of $\mu_{\mathbf{q}}$ with high probability.*

- *Data Point Updates: For every data point insertion $X \leftarrow X \cup \mathbf{z}$, ADDDATAPOINT($\mathbf{z}$) updates the corresponding sets and data structures, and returns the updated estimates $\hat{\mu}_{\mathbf{q}}$ that achieve $(1 \pm \varepsilon)$-multiplicative factor approximations of $\mu_{\mathbf{q}}$ for every maintained query point $\mathbf{q} \in Q$.*

*With high probability, the amortised running time for each update procedure is $\varepsilon^{-2} \cdot n^{o(1)} \cdot \mathrm{cost}(k)$, where $n = |X|$ is the current number of data points.*

To examine the significance of Theorem 3.1, notice that the amortised update time $\varepsilon^{-2} \cdot n^{o(1)} \cdot \mathrm{cost}(k)$ for the data point insertions is *independent* of the number of query points $|Q|$. This makes our algorithm significantly more efficient than re-estimating the query points after every update. While previous work (Liang et al., 2022) has shown that the CKNS KDE *estimator* can extended to the dynamic setting, our result shows that the *estimates* of a set of query points can be efficiently updated.

## 3.1 ANALYSIS FOR THE INITIALISATION

The initialisation step prepares all the data structures used for subsequent data and query point updates. The main component used in INITIALISE($X, Q, \varepsilon$) is the PREPROCESS($X, \varepsilon$) procedure, which preprocesses the data points in $X$ to ensure that the value $\mu_{\mathbf{q}}$ for any query point $\mathbf{q}$ can be fast approximated. To achieve this, PREPROCESS($X, \varepsilon$) defines

$$M \triangleq \left\{ 2^k \mid k \in \mathbb{Z}, 0 \leqslant k \leqslant \log(2n_1) \right\},$$

and indexes every $\mu_i \in M$ such that $\mu_0 \leqslant \ldots \leqslant \mu_{\log(2n_1)}$; note that $\mu_i = 2^i$. Then for $\mu_i \in M$ and $j \in [\log(2 \cdot n_1/\mu_i)]$ it samples every data point in $X$ with probability $\min\left\{1/(2^{j+1}\mu_i), 1\right\}$, and employs a hash function $H_{\mu_i, a, j, \ell}$ chosen uniformly at random from $\mathcal{H}^{k_j}$ with $k_j = \widetilde{O}(1)$ (cf. Lemma 2.5) to add every sampled $\mathbf{x} \in X$ to the buckets $\{B_{H_{\mu_i, a, j, \ell}}(\mathbf{x})\}_{\mu_i, a, j, \ell}$ indexed by all the possible $a \in [K_1]$ with $K_1 = O(\log n_1 \cdot \varepsilon^{-2})$, and $\ell \in [K_2]$ with $K_2 = O(\log(n_1) \cdot \mathrm{cost}(k))$. In addition, PREPROCESS($X, \varepsilon$) samples every data point in $X$ with probability $1/(2n_1)$ for all the possible values of $\mu_i$ and $a$, and adds the sampled point to set $\{\widetilde{X}_{\mu_i, a}\}_{\mu_i, a}$. We remark that our described PREPROCESS($X, \varepsilon$) is almost the same as the one presented in Charikar et al. (2020) and, although this data structure is sufficient to quickly output KDE estimates, we need to store additional *query-hash* buckets to update estimates when new data points arrive.

## 3.2 ANALYSIS FOR UPDATES

When a new query point $\mathbf{q}$ arrives, ADDQUERYPOINT($\mathbf{q}$) performs the following operations:

1. It computes the KDE estimate $\hat{\mu}_{\mathbf{q}}$ of $\mu_{\mathbf{q}}$ using the hash-based data structure from INITIALISE($X, Q, \varepsilon$).

2. It adds $\mathbf{q}$ to the sets $Q_{\mu_i}$ for every $\mu_i \in M$ that satisfies $\hat{\mu}_{\mathbf{q}} \leqslant \mu_i$, and adds $\mathbf{q}$ to the buckets $B^*_{H_{\mu_i, a, j, \ell}}(\mathbf{q})$ for $a \in [K_1]$, $j \in [J_{\mu_i}]$ and $\ell \in [K_2]$, which we call the *query-hash*.

When a new data point $\mathbf{z}$ arrives, ADDDATAPOINT($\mathbf{z}$) first checks whether the number of data points has doubled since the last (re-)construction of the data structure, and re-initialises the data structure if it is the case. Otherwise, ADDDATAPOINT($\mathbf{z}$) performs the following operations:

1. It samples $\mathbf{z}$ with probability $\min\left\{1/(2^{j+1}\mu_i), 1\right\}$ for all possible $\mu_i \in M$, and adds the sampled $\mathbf{z}$ to the buckets $B_{H_{\mu_i, a, j, \ell}}(\mathbf{z})$ for all $a \in K_1$, $j \in [J_{\mu_i}]$, and $\ell \in [K_2]$; it also samples $\mathbf{z}$ with probability $1/(2n_1)$ for all the possible values of $\mu_i$ and $a$, and adds the sampled point to the set $\{\widetilde{X}_{\mu_i, a}\}_{\mu_i, a}$. Notice that the way that $\mathbf{z}$ is added in the buckets is exactly the same as the one when executing INITIALISE($X \cup \mathbf{z}, Q, \varepsilon$), and hence ADDDATAPOINT($\mathbf{z}$) correctly updates all the buckets. We remark that this method of updating the buckets is similar to that of Liang et al. (2022), the difference being that for their dynamic setting instead of adding $\mathbf{z}$ to buckets, $\mathbf{z}$ replaces some $\mathbf{x}_i$.

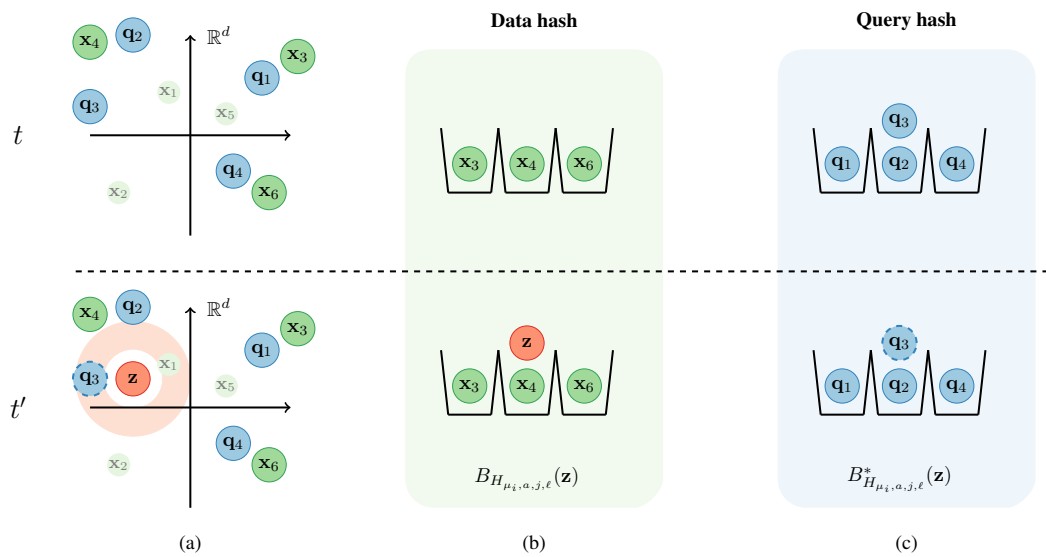

Figure 1: Illustration of ADDDATAPOINT($\mathbf{z}$) for a single iteration $\mu_i \in M$, $a \in K_1$, $j \in [J_{\mu_i}]$, and $\ell \in [K_2]$. The first row illustrates (a) the subsampled data points $Z \triangleq \{\mathbf{x}_3, \mathbf{x}_4, \mathbf{x}_6\}$ and query points $Q_{\mu_i} \triangleq \{\mathbf{q}_i\}_{i=1}^4$, (b) the bucketing of $Z$ by the hash function $H_{\mu_i,a,j,\ell}$, as well as (c) the bucketing of $Q$ by the same hash function. The second row illustrates (a) the relative location of a new arriving data point $\mathbf{z} \in \mathbb{R}^d$, with shaded red region indicating $L_j^{\mathbf{z}}$, (b) $\mathbf{z}$'s inclusion in the bucket $B_{H_{\mu_i,a,j,\ell}}(\mathbf{z})$, as well as (c) the recovery of $\mathbf{q}_3 \in B^*_{H_{\mu_i,a,j,\ell}}(\mathbf{z})$ because $\mathbf{z} \in L_j^{\mathbf{q}_3}$.

2. If $\mathbf{z}$ is sampled, ADDDATAPOINT($\mathbf{z}$) recovers all the points $\mathbf{q} \in B^*_{H_{\mu_i,a,j,\ell}}(\mathbf{z})$ in the query hash that satisfies $\mathbf{q} \in Q_{\mu_i} \setminus \left(\bigcup_{j' < i} Q_{\mu_{j'}}\right)$ and $\mathbf{z} \in L_j^{\mathbf{q}}$. Every such $\mathbf{q}$ is exactly the point whose KDE estimate $\hat{\mu}_{\mathbf{q}}$ would have included $\mathbf{z}$ if ADDQUERYPOINT($\mathbf{q}$) would have been called after running INITIALISE($X \cup \mathbf{z}, Q, \varepsilon$). Hence, the KDE estimates $\hat{\mu}_{\mathbf{q}}$ for the recovered $\mathbf{q}$ are updated appropriately.

See Figure 1 for illustration. The correctness and running time analysis of ADDDATAPOINT($\mathbf{z}$) and ADDQUERYPOINT($\mathbf{q}$) can be found in Section B of the appendix.

Finally, the DELETEQUERYPOINT($\mathbf{q}$) procedure simply removes any stored information about the query point $\mathbf{q}$ throughout all the maintained data, and its running time follows from the running time of ADDQUERYPOINT($\mathbf{q}$).

## 4 DYNAMIC SIMILARITY GRAPH CONSTRUCTION

In this section we design an approximate dynamic algorithm that constructs a similarity graph under a sequence of data point insertions, and analyse its performance. Given a set $X$ of $n_1$ points in $\mathbb{R}^d$ with $d = O(\log n_1)$, and a sequence of points $\{\mathbf{z}\}$ that will be added to $X$ over time, our designed algorithm consists of the CONSTRUCTGRAPH and UPDATEGRAPH procedures, whose performance is summarised as follows:

**Theorem 4.1 (Main Result 2).** *Let $k$ be a kernel function with $\text{cost}(k)$ as defined in Definition 2.6, and $X \subset \mathbb{R}^d$ a set of data points updated through point insertions. Then, the following statements hold:*

1. *The Initialisation Step: with probability at least $9/10$, the CONSTRUCTGRAPH procedure constructs an approximate similarity graph $G = (X, E, w_G)$ with $|E| = \widetilde{O}(n_1)$ edges, where $n_1 = |X|$ is the number of data points at initialisation. The running time of the initialisation step is $n_1^{1+o(1)} \cdot \text{cost}(k)$.*

> 2. *The Dynamic Update Step: for every new arriving data point* **z**, *the* UPDATEGRAPH *procedure updates the approximate similarity graph $G$, and with probability at least $9/10$ $G$ is an approximate similarity graph for $X \cup \mathbf{z}$. The expected amortised update time is $n^{o(1)} \cdot \mathrm{cost}(k)$, where $n$ is the number of currently considered data points.*

On the significance of Theorem 4.1, first notice that the algorithm achieves an update time of $n^{o(1)} \cdot \mathrm{cost}(k)$. For the Gaussian kernel, this corresponds to an update time of $n^{(1+o(1))\frac{1}{4}}$, which is much faster than the $\widetilde{O}(n)$ time needed to update the fully connected similarity graph. Secondly, our result demonstrates that, as long as the dynamically changing set $X \subset \mathbb{R}^d$ of points presents a clear cluster structure, an approximate similarity graph $G$ for $X$ can be dynamically maintained, and the conductance of every cluster in $G$ can be theoretically analysed, due to the formal definition of an approximate similarity graph (Definition A.3). This is another difference between our work and many heuristic clustering algorithms that lack a theoretical guarantee on their performance.

### 4.1 ANALYSIS FOR THE INITIALISATION

The main component of the initialisation step is our designed CONSTRUCTGRAPH$(X)$ procedure, which builds a (complete) binary tree $\mathcal{T}$ for $X$, such that each leaf corresponds to a data point $\mathbf{x}_i \in X$, and each internal node of $\mathcal{T}$ corresponds to a dynamic KDE data structure (described in Section 3) on the descendant leaves/data points. At a high level, CONSTRUCTGRAPH$(X)$ applies the SAMPLE$(X, \mathcal{T}, \ell)$ procedure to recursively traverse $\mathcal{T}$ and sample $L = O(\log |X|)$ neighbours for every vertex based on the KDEs maintained by the internal nodes. It also stores the paths $\mathcal{P}_{\mathbf{x}_i, \ell}$ ($\mathbf{x}_i \in X, \ell \leqslant L$), each of which records the internal nodes that $\mathbf{x}_i$ visits when sampling its $\ell$th neighbour; with these stored paths the algorithm can adaptively resample the tree as new data points arrive. In addition, the query points whose KDE estimates are dynamically maintained at any internal node $\mathcal{T}'$ are the data points $\mathbf{x}_i$ whose sample path $\mathcal{P}_{\mathbf{x}_i, \ell}$ visit $\mathcal{T}'$.

Our proof follows Macgregor & Sun (2023) at a high level. However, one crucial difference between our analysis and theirs is that the weight of every edge $(\mathbf{x}_i, \mathbf{x}_j)$ added by our algorithm is set to be $k(\mathbf{x}_i, \mathbf{x}_j)/\hat{w}(i,j)$. Here, $\hat{w}(i,j)$ depends on $\min\{\mathcal{T}.\mathsf{kde}.\hat{\mu}_{\mathbf{x}_i}, \mathcal{T}.\mathsf{kde}.\hat{\mu}_{\mathbf{x}_j}\}$, where $\mathcal{T}.\mathsf{kde}$ is the dynamic KDE data structure maintained at the root of $\mathcal{T}$. In particular, every sampled edge $(\mathbf{x}_i, \mathbf{x}_j)$ is added with this weight independent of the edge being sampled from $\mathbf{x}_i$ or $\mathbf{x}_j$. This modification allows for correct reweighting and resampling after a sequence of data point insertions.

### 4.2 ANALYSIS FOR DYNAMIC UPDATES

We design the UPDATEGRAPH$(G, \mathcal{T}, \mathbf{z})$ procedure to dynamically update our constructed graph such that the updated graph is an approximate similarity graph for $X \cup \mathbf{z}$. At a high level, UPDATEGRAPH$(G, \mathcal{T}, \mathbf{z})$ works as follows (see Figure 2 for illustration):

1. for every arriving data point $\mathbf{z}$, UPDATEGRAPH$(G, \mathcal{T}, \mathbf{z})$ creates a new leaf node for $\mathbf{z}$, and places it appropriately in $\mathcal{T}$ ensuring that the new tree is a complete binary tree;

2. UPDATEGRAPH$(G, \mathcal{T}, \mathbf{z})$ inspects the internal nodes from the new leaf $\mathbf{z}$ to the root of the tree, and for every such internal node it adds $\mathbf{z}$ as a new data point in the corresponding dynamic KDE estimators;

3. UPDATEGRAPH$(G, \mathcal{T}, \mathbf{z})$ further checks in every internal node along the sample path $\mathcal{P}_{\mathbf{x}_i, \ell}$ whether the KDE estimate of any $\mathbf{x}_i$ has changed due to the insertion of $\mathbf{z}$. If so, $\mathcal{P}_{\mathbf{x}_i, \ell}$ is added to the set $\mathcal{A}$ of paths that need to be updated. For every $\mathcal{P}_{\mathbf{x}_i, \ell} \in \mathcal{A}$, UPDATEGRAPH$(G, \mathcal{T}, \mathbf{z})$ finds the highest internal node $\mathcal{T}'$ where the KDE estimate of $\mathbf{x}_i$ has changed, removes the path from all nodes below $\mathcal{T}'$, and resamples the corresponding edges; this is achieved through RESAMPLE$(\mathcal{T}, \mathcal{P}_{\mathbf{x}_i, \ell})$. Additionally, it employs SAMPLE$(\{\mathbf{z}\}, \mathcal{T}, \ell)$ to sample $L$ new edges adjacent to $\mathbf{z}$.

From this description, it is easy to see that the total time complexity of UPDATEGRAPH$(G, \mathcal{T}, \mathbf{z})$ is dominated by (i) the time complexity of SAMPLE$(\{\mathbf{z}\}, \mathcal{T}, \ell)$ and RESAMPLE$(\mathcal{T}, \mathcal{P}_{\mathbf{x}, \ell})$, and (ii) the total number of paths $\mathcal{A}$ that need to be reconstructed. First, we study the time complexity of these two procedures, and our result is as follows:

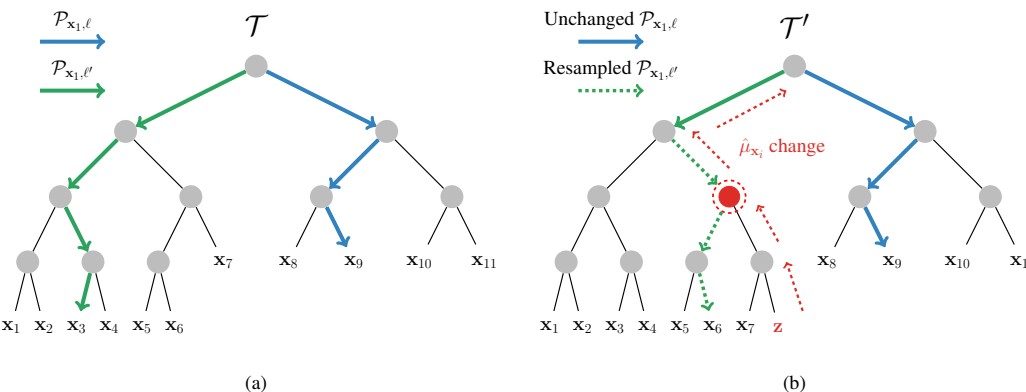

(a)                                          (b)

Figure 2: Illustration of updating $\mathcal{T}$ after performing UPDATEGRAPH($\mathbf{z}$). In (a), $\mathcal{P}_{\mathbf{x}_1,\ell}$ and $\mathcal{P}_{\mathbf{x}_1,\ell'}$ are generated by SAMPLE($\{\mathbf{x}_1\}, \mathcal{T}, \ell$) and SAMPLE($\{\mathbf{x}_1\}, \mathcal{T}, \ell'$), and correspond to edges $(\mathbf{x}_1, \mathbf{x}_9)$ and $(\mathbf{x}_1, \mathbf{x}_3)$. (b) Illustrates that, after adding $\mathbf{z}$, part of $\mathcal{P}_{\mathbf{x}_1,\ell'}$ is updated due to RESAMPLE($\mathcal{T}', \mathcal{P}_{\mathbf{x}_1,\ell'}$), and $(\mathbf{x}_1, \mathbf{x}_3)$ is replaced by $(\mathbf{x}_1, \mathbf{x}_6)$; however, the update on $\mathbf{z}$ doesn't change $\mathcal{P}_{\mathbf{x}_1,\ell}$.

**Lemma 4.2.** *For any* $\mathbf{x} \in \mathbb{R}^d$ *and* $\ell \in \mathbb{N}$*, the running time of* SAMPLE($\{\mathbf{x}\}, \mathcal{T}, \ell$) *(Algorithm 8) and* RESAMPLE($\mathcal{T}, \mathcal{P}_{\mathbf{x},\ell}$) *(Algorithm 10) is* $n^{o(1)} \cdot \mathrm{cost}(k)$.

Given that SAMPLE($\mathbf{z}, \mathcal{T}, \ell$) is called $L = \widetilde{O}(1)$ times by UPDATEGRAPH($G, \mathcal{T}, \mathbf{z}$), and RESAMPLE($\mathcal{T}, \mathcal{P}_{\mathbf{x},\ell}$) is called $|\mathcal{A}|$ times, Lemma 4.2 implies that the running time of UPDATEGRAPH depends on the number of re-sampled paths $|\mathcal{A}|$. Therefore, to prove the time complexity of UPDATE-GRAPH, it remains to show that $\mathbb{E}[|\mathcal{A}|]$ is sufficiently small.

Bounding the expected number of re-sampled paths corresponds to bounding the number of query points whose KDE estimates are updated at each affected internal node $\mathcal{T}'$. However, applying Theorem 3.1 directly is *insufficient* because, in our approximate similarity graph, the dynamic KDE data structures start with $Q = X$ rather than $Q = \emptyset$. As such, more careful analysis is needed, for which the following notation for the query points $Q$ will be used.

**Definition 4.3.** *Let* $\mathcal{T}$ *be the* KDE *tree constructed from* CONSTRUCTGRAPH($X$)*, and* $\mathcal{T}'$ *an internal node of* $\mathcal{T}$*. Then, for* $0 \leqslant j \leqslant i \leqslant \lceil \log(2 \cdot \mathcal{T}.kde.n) \rceil$*, we define the set*

$$Q_{\mu_i \to \mu_j}(\mathcal{T}') \triangleq \{\mathbf{q} \in \mathcal{T}.kde.Q \mid \mu_i \leqslant k(\mathbf{q}, \mathcal{T}.kde.X) < 2\mu_i \text{ and } k(\mathbf{q}, \mathcal{T}'.kde.X) \leqslant \mu_j\},$$

*where* $\mathcal{T}'.kde$ *is the dynamic* KDE *data structure maintained at* $\mathcal{T}'$*.*

The set $Q_{\mu_i \to \mu_j}(\mathcal{T}')$ represents the set of query points $\mathbf{q} \in X$ whose KDE estimates are bounded by $\mu_i$ when computed with respect to the data points $X$ at the root of the tree, and bounded by $\mu_j$ for $j \leqslant i$ when computed with respect to the data points $X'$ represented at the internal node $\mathcal{T}'$. Intuitively, this set captures the query points whose KDE estimates decrease when moving from the root of the tree to the internal node $\mathcal{T}'$. These sets exhibit the following useful property.

**Lemma 4.4.** *It holds for any* $\mathbf{q} \in Q_{\mu_i \to \mu_j}(\mathcal{T}')$ *that* $\mathbb{P}[\mathbf{q} \in \mathcal{T}'.kde.Q] = \widetilde{O}\left(\frac{\mu_j}{\mu_i}\right)$.

To bound the number of maintained query points whose estimate is updated, we look at the expected number of collisions caused by hashing $\mathbf{z}$ in the dynamic KDE data structure $\mathcal{T}'.kde$ at every affected internal node $\mathcal{T}'$. Crucially, by separately analysing the contributions from query points in $Q_{\mu_{i'} \to \mu_i}(\mathcal{T}')$ for $i' \geqslant i$ and applying Lemma 4.4, we can bound the expected number of colliding points in the buckets $\mathcal{T}'.kde.B_{H_{\mu_i,a,j,\ell}}(\mathbf{z})$ sufficiently tightly at each affected internal node $\mathcal{T}'$.

**Lemma 4.5** (Informal version of Lemma C.2)**.** *Let* $\mathbf{z}$ *be the data point that is added to* $\mathcal{T}$ *through our designed update procedures, and* $\mathcal{T}'$ *be any internal node that lies on the path from the new leaf* LEAF($\mathbf{z}$) *to the root of* $\mathcal{T}$*. Then it holds for any* $i, a, j,$ *and* $\ell$ *that that*

$$\mathbb{E}_{H_{\mu_i,a,j,\ell}}\left[|\{\mathbf{q} \in \mathcal{T}'.kde.Q_{\mu_i} \mid \mathcal{T}'.kde.H_{\mu_i,a,j,\ell}(\mathbf{z}) = \mathcal{T}'.kde.H_{\mu_i,a,j,\ell}(\mathbf{q})\}|\right] = \widetilde{O}\left(\mu_i \cdot 2^{j+1}\right).$$

Combining Lemma 4.5 with the fact that $\mathbf{z}$ is sampled with probability $\min\left\{1/(2^{j+1}\mu_i), 1\right\}$ for all possible $i \in [\lceil\log(2 \cdot \mathcal{T}.\mathsf{kde}.n)\rceil]$ and $j \in J_{\mu_i}$ along every affected internal node $\mathcal{T}'$, and noting that there are $\widetilde{O}(1)$ such nodes, we obtain the following result.

**Lemma 4.6.** *For every added $\mathbf{z}$, the expected number of paths $\mathcal{A}$ that needs to be resampled by* $\mathrm{UPDATEGRAPH}(G, \mathcal{T}, \mathbf{z})$ *satisfies* $\mathbb{E}[|\mathcal{A}|] = \widetilde{O}(1)$.

Combining Lemmas 4.2 and 4.6 with the running time analysis of other involved procedures proves the time complexity in the second part of Theorem 4.1. To show that our dynamically maintained $G$ is an approximate similarity graph, we prove in Lemma C.6 that running $\mathrm{CONSTRUCTGRAPH}(X)$ followed by $\mathrm{UPDATEGRAPH}(G, \mathcal{T}, \mathbf{z})$ is equivalent to running $\mathrm{CONSTRUCTGRAPH}(X \cup \mathbf{z})$; hence the correctness of our constructed $G$ follows from the one for $\mathrm{CONSTRUCTGRAPH}(X)$.

## 5 EXPERIMENTS

In this section, we experimentally evaluate our proposed dynamic algorithms for KDE and approximate similarity graph construction on the Gaussian kernel. All experiments are performed on a compute server with 64 AMD EPYC 7302 16-Core Processors and 500 Gb of RAM. We report the 2-sigma errors for all numerical results based on 3 repetitions of each experiment, and Section D gives additional experimental details and results. The code to reproduce the results is included as part of the supplementary material.

We evaluate the algorithms on a variety of real-world and synthetic data, and we summarise their properties in Table 1. The datasets cover a variety of domains, including synthetic data (blobs (Pedregosa et al., 2011)), images (mnist (Lecun et al., 1998), aloi (Geusebroek et al., 2005)), image embeddings (cifar10 (He et al., 2016; Krizhevsky, 2009)), word embeddings (glove (Pennington et al., 2014)), and mixed numerical datasets (msd (Bertin-Mahieux et al., 2011), covtype (Blackard & Dean, 1999), census (Meek et al., 1990)).

Table 1: Dataset size information

| Dataset | $n$ | $d$ |
|---------|-----|-----|
| blobs | 20,000 | 10 |
| mnist | 70,000 | 728 |
| cifar10 | 50,000 | 2,048 |
| aloi | 108,000 | 128 |
| msd | 515,345 | 90 |
| covtype | 581,012 | 54 |
| glove | 1,193,514 | 100 |
| census | 2,458,285 | 68 |

### 5.1 DYNAMIC KDE

To our knowledge, our proposed algorithm is the first which solves the dynamic kernel density estimation problem. For this reason, we compare against the following baseline approaches:

- EXACT: the exact kernel density estimate, computed incrementally as data points are added.

- DYNAMICRS: a dynamic KDE estimator based on uniform random sampling of the data. For all experiments, we uniformly subsample the data with sampling probability 0.1.

- CKNS: we use the fast kernel density estimation algorithm proposed by Charikar et al. (2020), and fully re-compute the estimates every time the data is updated.

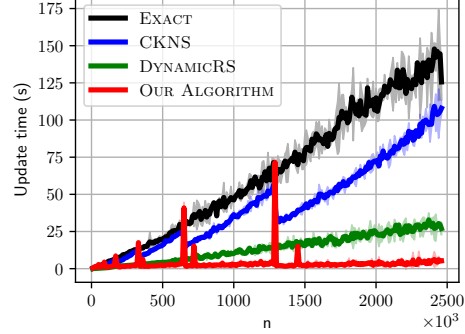

Figure 3: Dynamic KDE update times on the census dataset

For each dataset, we set the parameter $\sigma$ of the Gaussian kernel such that the average kernel density $\mu_{\mathbf{q}} \cdot n^{-1} \approx 0.01$ (Karppa et al., 2022). We split the data points into chunks of size 1,000 for aloi, msd, and covtype, and size 10,000 for glove and census. Then, we add one chunk at a time to the set of data points $X$ and the set of query points $Q$. At each iteration, we evaluate the kernel density estimates $\hat{\mu}_{\mathbf{q}}$ produced by each algorithm with the relative

Table 2: The experimental results for the dynamic KDE algorithms. For each dataset, we shade the cells corresponding to the algorithm with the lowest running time. The running times of the exact algorithm are 164.5, 2715.6, 2179.9, 5251.7, and 16279.9.

| | CKNS | | DYNAMICRS | | OUR ALGORITHM | |
|---|---|---|---|---|---|---|
| dataset | Time (s) | Err | Time (s) | Err | Time (s) | Err |
| aloi | $619.0_{\pm 10.7}$ | $0.050_{\pm 0.006}$ | $19.7_{\pm 0.3}$ | $0.010_{\pm 0.003}$ | $46.9_{\pm 0.7}$ | $0.060_{\pm 0.021}$ |
| msd | $14,360.0_{\pm 0.0}$ | $0.385_{\pm 0.000}$ | $1,887.8_{\pm 0.0}$ | $5.430_{\pm 0.000}$ | $306.4_{\pm 0.0}$ | $0.388_{\pm 0.000}$ |
| covtype | $5,650.3_{\pm 109.0}$ | $0.159_{\pm 0.002}$ | $309.2_{\pm 2.4}$ | $0.018_{\pm 0.003}$ | $151.7_{\pm 4.5}$ | $0.196_{\pm 0.017}$ |
| glove | $2,640.8_{\pm 1677.7}$ | $0.221_{\pm 0.229}$ | $1,038.6_{\pm 26.5}$ | $0.004_{\pm 0.005}$ | $445.6_{\pm 214.6}$ | $0.296_{\pm 0.469}$ |
| census | $10,471.5_{\pm 160.6}$ | $0.080_{\pm 0.000}$ | $3,424.8_{\pm 5.2}$ | $0.005_{\pm 0.001}$ | $836.5_{\pm 44.6}$ | $0.102_{\pm 0.021}$ |

Table 3: Running time and NMI results for the dynamic similarity graph algorithms. For each dataset, the shaded cells correspond to the algorithm with the lowest running time.

| | FULLYCONNECTED | | KNN | | OUR ALGORITHM | |
|---|---|---|---|---|---|---|
| dataset | Time (s) | NMI | Time (s) | NMI | Time (s) | NMI |
| blobs | $72.8_{\pm 2.2}$ | $1.000_{\pm 0.000}$ | $383.6_{\pm 3.9}$ | $0.933_{\pm 0.095}$ | $21.2_{\pm 0.8}$ | $1.000_{\pm 0.000}$ |
| cifar10 | $19,158.2_{\pm 231.6}$ | $0.001_{\pm 0.000}$ | $3,503.0_{\pm 490.6}$ | $0.227_{\pm 0.002}$ | $1,403.5_{\pm 152.4}$ | $0.339_{\pm 0.021}$ |
| mnist | $1,328.3_{\pm 159.5}$ | $0.460_{\pm 0.000}$ | $5,796.3_{\pm 234.3}$ | $0.812_{\pm 0.003}$ | $1,470.3_{\pm 77.9}$ | $0.523_{\pm 0.011}$ |

error (Karppa et al., 2022)

$$\text{err} = \frac{1}{|Q|} \sum_{\mathbf{q} \in Q} \left| \frac{\hat{\mu}_{\mathbf{q}} - \mu_{\mathbf{q}}}{\mu_{\mathbf{q}}} \right|.$$

Table 2 gives the total running time and final relative error for each algorithm, and Figure 3 shows the time taken to update the data structure for the census dataset at each iteration. From these results, we observe that our algorithm scales better to large datasets than the baseline algorithms, while maintaining low relative errors. Figure 3 further shows that the update time of our algorithm is sub-linear in the number of data points, as shown theoretically in Theorem 3.1. The update time of the other algorithms is linear in $n$ and their total running time is quadratic.

## 5.2 DYNAMIC CLUSTERING

For the dynamic similarity graph algorithm, we compare against the two baseline algorithms:

- FULLYCONNECTED: the fully-connected similarity graph with the Gaussian kernel;
- KNN: the $k$-nearest neighbour graph, for $k = 20$.

We split the datasets into chunks of $1,000$ and add each chunk to the dynamically constructed similarity graph, adding one complete ground-truth cluster at a time. At each iteration, we apply the spectral clustering algorithm to the constructed similarity graph and report the normalised mutual information (NMI) (Lancichinetti et al., 2009) with respect to the ground truth clusters. Table 3 shows the total running times and final NMI values for each algorithm on each dataset. From these results, we see that our algorithm achieves a competitive NMI value with faster running time than the baseline algorithms.

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

# A    ADDITIONAL BACKGROUND KNOWLEDGE

This section presents additional background knowledge used in our analysis, and is organised as follows: Section A.1 lists further notation for graphs and useful facts in spectral graph theory; Section A.2 formally defines the concept of an approximate similarity graph.

## A.1    NOTATION

Let $G = (V, E, w)$ be an undirected graph of $n$ vertices and weight function $w : V \times V \to \mathbb{R}_{\geqslant 0}$. For any edge $e = (u, v) \in E$, we write $w_G(u, v)$ or $w_G(e)$ to express the weight of $e$. For a vertex $u \in V$, we denote its *degree* by $\deg_G(u) \triangleq \sum_{v \in V} w_G(u, v)$, and the volume for any $S \subseteq V$ is defined as $\mathrm{vol}_G(S) \triangleq \sum_{u \in S} \deg_G(u)$. For any two subsets $S, T \subset V$, we define the *cut value* between $S$ and $T$ by $w_G(S, T) \triangleq \sum_{e \in E_G(S,T)} w_e$, where $E_G(S, T)$ is the set of edges between $S$ and $T$. Moreover, for any $S \subset V$, the conductance of $S$ is defined as

$$\Phi_G(S) \triangleq \frac{w_G(S, V \setminus S)}{\min\{\mathrm{vol}_G(S), \mathrm{vol}_G(V \setminus S)\}}$$

if $S \neq \emptyset$, and $\Phi_G(S) = 1$ if $S = \emptyset$. For any integer $k \geqslant 2$, we call subsets of vertices $A_1, \dots, A_k$ a $k$-way partition of $G$ if $\bigcup_{i=1}^k A_i = V$ and $A_i \cap A_j = \emptyset$ for different $i$ and $j$. We define the *k-way expansion* of $G$ by

$$\rho_G(k) \triangleq \min_{\text{partitions } A_1, \dots, A_k} \max_{1 \leqslant i \leqslant k} \Phi_G(A_i).$$

Part of our analysis is based on algebraic properties of graphs, hence we define graph Laplacian matrices. For a graph $G = (V, E, w)$, let $D_G \in \mathbb{R}^{n \times n}$ be the diagonal matrix defined by $D_G(u, u) = \deg_G(u)$ for all $u \in V$. We denote by $A_G \in \mathbb{R}^{n \times n}$ the *adjacency matrix* of $G$, where $A_G(u, v) = w_G(u, v)$ for all $u, v \in V$. The *normalised Laplacian matrix* of $G$ is defined as

$$\mathcal{L}_G \triangleq I - D_G^{-1/2} A_G D_G^{-1/2},$$

where $I$ is the $n \times n$ identity matrix. The normalised Laplacian $\mathcal{L}_G$ is symmetric and real-valued, and has $n$ real eigenvalues which we write as $\lambda_1(\mathcal{L}_G) \leqslant \dots \leqslant \lambda_n(\mathcal{L}_G)$. We sometimes refer to the $i$th eigenvalue of $\mathcal{L}_G$ as $\lambda_i$ if it is clear from the context. It is known that $\lambda_1(\mathcal{L}_G) = 0$ and $\lambda_n(\mathcal{L}_G) \leqslant 2$ (Chung, 1997). The following result will be used in our analysis.

**Lemma A.1** (higher-order Cheeger inequality, Lee et al. (2014)). *It holds for any graph $G$ and $k \geqslant 2$ that*

$$\frac{\lambda_k(\mathcal{L}_G)}{2} \leqslant \rho_G(k) = O\left(k^3\right)\sqrt{\lambda_k(\mathcal{L}_G)}. \tag{A.1}$$

## A.2    APPROXIMATE SIMILARITY GRAPH

We first introduce the notion of cluster-preserving sparsifiers.

**Definition A.2** (Cluster-preserving sparsifier, Sun & Zanetti (2019)). *Let $F = (V, E, w)$ be any graph with $k$ clusters, and $\{S_i\}_{i=1}^k$ a $k$-way partition of $F$ corresponding to $\rho_F(k)$. We call a re-weighted subgraph $G = (V, E' \subset E, w_G)$ a cluster-preserving sparsifier of $F$ if (i) $\Phi_G(S_i) = O(k \cdot \Phi_F(S_i))$ for $1 \leqslant i \leqslant k$, and (ii) $\lambda_{k+1}(\mathcal{L}_G) = \Omega(\lambda_{k+1}(\mathcal{L}_F))$.*

Notice that graph $F = (V, E, w)$ has exactly $k$ clusters if (i) $F$ has $k$ disjoint subsets $S_1, \dots, S_k$ of low conductance, and (ii) any $(k + 1)$-way partition of $F$ would include some $A \subset V$ of high conductance, which would be implied by a lower bound on $\lambda_{k+1}(\mathcal{L}_F)$ due to (A.1). Together with the well-known eigen-gap heuristic (von Luxburg, 2007) and theoretical analysis on spectral clustering (Peng et al., 2017), these two conditions ensure that the $k$ optimal clusters in $F$ have low conductance in $G$ as well. Based on this, we define approximate similarity graphs as follows:

**Definition A.3** (Approximate Similarity Graph). *For any set $X \subset \mathbb{R}^d$ of $n$ data points and the fully connected similarity graph $F$ on $X$, we call a sparse graph $G$ with $\widetilde{O}(n)$ edges an approximate similarity graph on $X$ if $G$ is a cluster-preserving sparsifier of $F$.*

We call $G$ an approximate similarity graph in the extended abstract if $G$ satisfies the properties of Definition A.3.

## B   OMITTED DETAIL FROM SECTION 3

This section provides the detailed explanations omitted from Section 3, and is organised as follows: Section B.1 analyses the initialisation and querying procedures. Section B.2 analyses the dynamic update step for adding data points. Finally, Section B.3 proves Theorem 3.1.

Algorithm 1 describes all the used procedures and corresponding subprocedures, whose performance is summarised in Theorem 3.1.

---

**Algorithm 1** DYNAMICKDE$(X, Q, \varepsilon)$

---

1: **Members**
2:    $\hat{\mu}_{\mathbf{q}}$                                            ▷ Query estimates for every point $\mathbf{q} \in Q$
3:    $\varepsilon$                                            ▷ Precision parameter for KDE estimate
4:    For $\mu_i \in M$, create set $Q_{\mu_i}$         ▷ Set of data points with query estimate less than $\mu_i$
5: **procedure** INITIALISE$(X, Q, \varepsilon)$
6:    $n', n \leftarrow |X|, \bar{m} \leftarrow \lceil \frac{C}{\varepsilon^2} \rceil, \bar{N} \leftarrow \lceil \log(2n') \rceil$                   ▷ $C$ is a universal constant
7:    $K_1 \leftarrow \bar{m} \cdot \bar{N}$                       ▷ Number of independent estimators used
8:    $J_{\mu_i} \leftarrow \left\lceil \log \frac{2n'}{\mu_i} \right\rceil$ for $\mu_i \in M$                              ▷ See Definition 2.3
9:
10:    PREPROCESS$(X, \varepsilon)$      ▷ Initialise the Charikar et al. (2020) data structure (Algorithm 2)
11:    PREPROCESSQUERYPOINTS$(X, Q, \varepsilon)$                              ▷ (Algorithm 4)
12: **procedure** ADDQUERYPOINT$(\mathbf{q})$
13:    $\hat{\mu}_{\mathbf{q}} \leftarrow$ QUERYPOINT$(X, \mathbf{q}, \varepsilon)$                              ▷ (Algorithm 3)
14:    Store $\hat{\mu}_{\mathbf{q}}$
15:    Add $\mathbf{q}$ to all sets $Q_{\mu_i}$ such that $\hat{\mu}_{\mathbf{q}} \leqslant \mu_i$
16:    ADDFULLHASH$(\mathbf{q})$                              ▷ (Algorithm 4)
17: **procedure** ADDDATAPOINT$(\mathbf{z})$
18:    $n \leftarrow n + 1$
19:    **if** $n - n' > n'$ **then**
20:        INITIALISE$(X \cup \mathbf{z}, Q, \varepsilon)$                              ▷ We reconstruct the data structure
21:    **else**
22:        ADDPOINTANDUPDATEQUERIES$(\mathbf{z}, Q, \varepsilon)$                              ▷ (Algorithm 6)
23: **procedure** DELETEQUERYPOINT$(\mathbf{q})$
24:    DELETEFROMDATA$(\mathbf{z})$                              ▷ (Algorithm 5)
25:    Delete $\hat{\mu}_{\mathbf{q}}$

---

**Algorithm 2** DynamicKDE Preprocessing

---

1: **procedure** PREPROCESS$(X, \varepsilon)$
2:    **Input:** the set $X$ of data points, and the precision estimate $\varepsilon$
3:    **for** $\mu_i \in M$ **do**
4:        **for** $a = 1, 2, \ldots, K_1$ **do**                   ▷ $K_1 = O(\log n'/\varepsilon^2)$ independent repetitions
5:            **for** $j = 1, 2, \ldots, J_{\mu_i}$ **do**                   ▷ $J_{\mu_i} = \left\lceil \log \frac{2n'}{\mu_i} \right\rceil$ geometric weight levels
6:                $K_{2,j} \leftarrow 200 \log n' \cdot p_{\text{near},j}^{-k_j}$ ▷ See Lemma 2.5 and (2.1) for def. of $p_{\text{near},j}$ and $k_j$
7:                $p_{\text{samp}} \leftarrow \min \left\{ \frac{1}{2^{j+1}\mu_i}, 1 \right\}$
8:                Sample every $\mathbf{x} \in X$ w.p. $p_{\text{samp}}$, and let $Z_{\mu_i,j}$ be the set of sampled elements
9:                **for** $\ell = 1, 2, \ldots, K_{2,j}$ **do**
10:                   Draw a hash function $H_{\mu_i,a,j,\ell}$ from hash family $\mathcal{H}^{k_j}$ (Lemma 2.5)
11:                   **for** $\mathbf{x} \in Z_{\mu_i,j}$ **do**
12:                       Store $\mathbf{x}$ in the bucket $B_{H_{\mu_i,a,j,\ell}}(\mathbf{x})$
13:               Sample every $\mathbf{x} \in X$ w.p. $\frac{1}{2n'}$, and let $\widetilde{X}_{\mu_i,a}$ be the set of sampled elements.
14:               Store $\widetilde{X}_{\mu_i,a}$                   ▷ Set $\widetilde{X}_{\mu_i,a}$ will be used to recover points beyond $L_{J+1}$

---

---

**Algorithm 3** DynamicKDE Query Procedures

---

1: **procedure** QUERYMUESTIMATE$(X, \mathbf{q}, \varepsilon, \mu_i)$
2:     **Input:** set $X$ of data points, query point $\mathbf{q}$, precision estimate $\varepsilon$, and KDE estimate $\mu_i$.
3:     **for** $a = 1, 2, \ldots, K_1$ **do**             $\triangleright O(\log n'/\varepsilon^2)$ independent repetitions
4:         **for** $j = 1, 2, \ldots, J_{\mu_i}$ **do**
5:             $K_{2,j} \leftarrow 200 \log n' \cdot p_{\text{near},j}^{-k_j}$
6:             **for** $\ell = 1, 2, \ldots, K_{2,j}$ **do**
7:                 Recover points $\mathbf{x} \in B_{H_{\mu_i,a,j,\ell}}(\mathbf{q})$ such that $\mathbf{x} \in L_j^{\mathbf{q}}$
8:         Recover points $\mathbf{x} \in \widetilde{X}_{\mu_i,a}$ such that $\mathbf{x} \in L_{J_{\mu_i}+1}^{\mathbf{q}}$.
9:         $S \leftarrow$ set of all recovered points in this iteration
10:         **for** $\mathbf{x}_{i'} \in S$ **do**
11:             $w_{i'} \leftarrow k(\mathbf{x}_{i'}, \mathbf{q})$
12:             **if** $\mathbf{x}_{i'} \in L_j^{\mathbf{q}}$ for some $j \in [J_{\mu_i}]$ **then**
13:                 $p_{i'} \leftarrow \min\left\{\frac{1}{2^{j+1}\mu_i}, 1\right\}$
14:             **else if** $\mathbf{x}_i \in X \setminus \left(\bigcup_{j \in [J_{\mu_i}]} L_j^{\mathbf{q}}\right)$ **then**
15:                 $p_{i'} \leftarrow \frac{1}{2n'}$
16:         $Z_{\mathbf{q},a} \leftarrow \sum_{\mathbf{x}_{i'} \in S} w_{i'}/p_{i'}$
17:         Store $Z_{\mathbf{q},a}$
18:     **for** $b = 1, 2, \ldots, \bar{N}$ **do**         $\triangleright$ Get median of $\bar{N} = O(\log n)$ means of size $O(1/\varepsilon^2)$
19:         $\bar{Z}_{\mathbf{q},b} \leftarrow \frac{1}{\bar{m}} \sum_{a=(b-1)\bar{m}+1}^{b\bar{m}} Z_{\mathbf{q},a}$
        **return** Median $\left(\bar{Z}_{\mathbf{q},1}, \bar{Z}_{\mathbf{q},2}, \ldots, \bar{Z}_{\mathbf{q},\bar{N}}\right)$
20: **procedure** QUERYPOINT$(X, \mathbf{q}, \varepsilon)$
21:     **Input:** set $X$ of data points, query point $\mathbf{q}$, precision estimate $\varepsilon$.
22:     **for** $\mu_i \in [\mu_{\log(2n')}, \mu_{\log(2n')-1}, \ldots, \mu_1, \mu_0]$ **do**
23:         **if** QUERYMUESTIMATE$(X, \mathbf{q}, \varepsilon, \mu_i) > \mu_i$ **then**
24:             **return** QUERYMUESTIMATE$(X, \mathbf{q}, \varepsilon, \mu_{i+1})$   $\triangleright$ If estimate is larger than $\mu_i$, return the previous estimate.
25:     **return** $0$

---

---

**Algorithm 4** DynamicKDE Full Hash Procedures

---

1: **procedure** PREPROCESSQUERYPOINTS$(X, Q, \varepsilon)$
2:     **Input:** the set $X$ of data points, the set of query points $Q$, and the precision estimate $\varepsilon$.
3:     **for** $\mathbf{q} \in Q$ **do**
4:         $\hat{\mu}_{\mathbf{q}} \leftarrow$ QUERYPOINT$(X, \mathbf{q}, \varepsilon)$
5:         Store $\hat{\mu}_{\mathbf{q}}$
6:         Add $\mathbf{q}$ to all sets $Q_{\mu_i}$ such that $\hat{\mu}_{\mathbf{q}} \leqslant \mu_i$
7:     **for** $\mu_i \in M$ **do**
8:         **for** $a = 1, 2, \ldots, K_1$ **do**
9:             **for** $j = 1, 2, \ldots, J_{\mu_i}$ **do**
10:                 $K_{2,j} \leftarrow 200 \log n' \cdot p_{\text{near},j}^{-k_j}$
11:                 **for** $\ell = 1, 2, \ldots, K_{2,j}$ **do**
12:                     **for** $\mathbf{q} \in Q_{\mu_i}$ **do**
13:                         Store $\mathbf{q}$ in full bucket $B^*_{H_{\mu_i,a,j,\ell}}(\mathbf{q})$ corresponding to hash value $H_{\mu_i,a,j,\ell}(\mathbf{q})$
14: **procedure** ADDFULLHASH$(\mathbf{q})$
15:     **Input:** New query point $\mathbf{q}$
16:     **for** $\mu_i \geqslant \hat{\mu}_{\mathbf{q}}$ **do**
17:         **for** $a = 1, 2, \ldots, K_1$ **do**
18:             **for** $j = 1, 2, \ldots, J_{\mu_i}$ **do**
19:                 $K_{2,j} \leftarrow 200 \log n' \cdot p_{\text{near},j}^{-k_j}$
20:                 **for** $\ell = 1, 2, \ldots, K_{2,j}$ **do**
21:                     Store $\mathbf{q}$ in full bucket $B^*_{H_{\mu_i,a,j,\ell}}(\mathbf{q})$ corresponding to hash value $H_{\mu_i,a,j,\ell}(\mathbf{q})$

---

**Algorithm 5** DynamicKDE Delete Procedures

---

1: **procedure** DELETEFROMDATA$(\mathbf{z})$
2:     **Input:** Query point $\mathbf{q}$ to remove
3:     **for** $\mu_i \in M$ **do**
4:         Remove $\mathbf{q}$ from $Q_{\mu_i}$
5:         **for** $a = 1, 2, \ldots, K_1$ **do**
6:             **for** $j = 1, 2, \ldots, J_{\mu_i}$ **do**
7:                 $K_{2,j} \leftarrow 200 \log n' \cdot p_{\text{near},j}^{-k_j}$
8:                 **for** $\ell = 1, 2, \ldots, K_{2,j}$ **do**
9:                     Remove $\mathbf{q}$ from full bucket $B^*_{H_{\mu_i,a,j,\ell}}(\mathbf{q})$

---

### B.1   ANALYSIS OF THE INITIALISE AND ADDQUERYPOINT PROCEDURES

We first analyse the INITIALISE$(X, Q, \varepsilon)$ and ADDQUERYPOINT$(\mathbf{z})$ procedures in Algorithm 1, whose corresponding subprocedures are presented in Algorithms 2, 3, and 4.

We assume that $\mu_i$ is an estimate satisfying $\mu_{\mathbf{q}} \leqslant \mu_i$; we will justify this assumption in Remark 1. We first analyse the expected number of data points to be sampled in each bucket $B_{H_{\mu_i,a,j,\ell}}(\mathbf{q})$.

**Lemma B.1.** *For any $a$, $\mu_i$, $j$, $\ell$, it holds for $\mathbf{q} \in Q_{\mu_i}$ that*

$$\mathbb{E}_{H_{\mu_i,a,j,\ell}} \left[ \left| \left\{ \mathbf{x} \in \widetilde{X}_{\mu_i,j} \mid H_{\mu_i,a,j,\ell}(\mathbf{q}) = H_{\mu_i,a,j,\ell}(\mathbf{x}) \right\} \right| \right] = \widetilde{O}(1),$$

*any for $\mathbf{q} \in Q_{\mu_{i'}}$ that*

$$\mathbb{E}_{H_{\mu_i,a,j,\ell}} [|\{\mathbf{x} \in X \mid H_{\mu_i,a,j,\ell}(\mathbf{x}) = H_{\mu_i,a,j,\ell}(\mathbf{q})|\}|] = \widetilde{O}\left(2^{j+1}\mu_{i'}\right).$$

*Proof.* We compute the expected number of collisions in the bucket $B_{H_{\mu_i,a,j,\ell}}(\mathbf{q})$, and our analysis is by case distinction.

*Case 1: $j' \leqslant j$.* It holds by Lemma 2.4 that $\left| L_{j'}^{\mathbf{q}} \right| \leqslant 2^{j'} \mu_{\mathbf{q}} \leqslant 2^{j'} \mu_i$, which upper bounds the number of points that could collide from these geometric weight levels. Since every data point is sampled with probability $1/\left(2^{j+1}\mu_i\right)$ in this iteration, the expected number of sampled data points is $O(1)$.

*Case 2: $j < j' \leqslant J_{\mu_i} + 1$.* We analyse the effect of the LSH. Note that in the $j$th iteration, we choose an LSH function whose corresponding distance level is $r_j$, and use

$$k \triangleq k_j = -\frac{1}{\log p_{\text{near},j}} \cdot \max_{i=j+1,\ldots,J_{\mu_i}+1} \left\lceil \frac{i-j}{c_{i,j}^2(1-\beta)} \right\rceil.$$

as the number of concatenations. Then, it holds for $\mathbf{p} \in L_{j'}^{\mathbf{q}}$ that

$$\mathbb{P}_{H^* \in \mathcal{H}^k}\left[H^*(\mathbf{p}) = H^*(\mathbf{q})\right] \leqslant p^{kc^2(1-\beta)},$$

where $c \triangleq c_{i,j} = \min\left\{\frac{r_{i-1}}{r_j}, O\left(\log^{1/7} n'\right)\right\}$ and $p \triangleq p_{\text{near},j}$. Hence, the expected number of points from weight level $L_{j'}^{\mathbf{q}}$ in the query hash bucket is $O\left(2^{j'-j}\right) \cdot p^{kc^2(1-\beta)} = \widetilde{O}(1)$, where the last line holds by the choice of $k$. Combining the two cases proves the first statement.

The second statement holds for the same analysis, but we have instead that $|L_{j'}^{\mathbf{q}}| \leqslant 2^{j'}\mu_{\mathbf{q}} \leqslant 2^{j'}\mu_{i'}$ for $\mathbf{q} \in Q_{\mu_{i'}}$. $\qquad\square$

The query time complexity for QUERYMUESTIMATE$(X, \mathbf{q}, \varepsilon, \mu_i)$ follows from Charikar et al. (2020).

**Lemma B.2** (Query Time Complexity, Charikar et al. (2020)). *For any kernel $k$, the expected running time of* QUERYMUESTIMATE$(X, \mathbf{q}, \varepsilon, \mu_i)$ *(Algorithm 3) is $\varepsilon^{-2} \cdot n_1^{o(1)} \cdot \text{cost}(k)$.*

Next we show that our returned estimator gives a good approximation with high probability.

**Lemma B.3.** *For any $\mathbf{q} \in \mathbb{R}^d$, $\mu_{\mathbf{q}} \in (0, 2n_1]$, $\mu_i \geqslant \mu_{\mathbf{q}}$, $\varepsilon \in \left(1/n_1^5, 1\right)$, the estimator $Z_{\mathbf{q},a}$ for $a \in [K_1]$ constructed in* QUERYMUESTIMATE$(X, \mathbf{q}, \varepsilon, \mu_i)$ *(Algorithm 3) satisfies that $\left(1 - n_1^{-9}\right) \cdot \mu_{\mathbf{q}} \leqslant \mathbb{E}[Z_{\mathbf{q},a}] \leqslant \mu_{\mathbf{q}}$.*

*Proof.* We first fix arbitrary $j = j^*$ and $a = a^*$, and sample some point $\mathbf{p} \in L_{j^*}^{\mathbf{q}}$. By Lemma 2.5 we have

$$\mathbb{P}_{H^* \sim \mathcal{H}^{k_j}}\left[H^*(\mathbf{p}) = H^*(\mathbf{q})\right] \geqslant p_{\text{near},j}^{k_j}.$$

Since we repeat this process for $K_{2,j} = 200 \log n_1' \cdot p_{\text{near},j}^{-k_j}$ times, it holds with high probability that any sampled point $\mathbf{p}$ from band $L_{j^*}^{\mathbf{q}}$ is recovered in at least one phase. By applying the union bound, the probability that all the sampled points are recovered is at least $1 - n_1^{-9}$.

We define $Z \triangleq Z_{\mathbf{q},a}$, and have that

$$\mathbb{E}[Z] = \sum_{i=1}^{n_1} \frac{\mathbb{E}[\chi_i]}{p_i} \cdot w_i,$$

where $\chi_i = 1$ if point $\mathbf{x}_i$ is sampled and $\chi_i = 0$ otherwise. Hence, it holds that $\left(1 - n_1^{-9}\right)p_i \leqslant \mathbb{E}[\chi_i] \leqslant p_i$, which implies that $\left(1 - n_1^{-9}\right)\mu_{\mathbf{q}} \leqslant \mathbb{E}[Z] \leqslant \mu_{\mathbf{q}}$. $\qquad\square$

**Remark 1.** *Lemma B.3 shows that the estimator $Z_{\mathbf{q},a}$ is unbiased (up to some small inverse polynomial error) for any choice of $\mu_i \geqslant \mu_{\mathbf{q}}$. Therefore, when $\mu_i \geqslant 4\mu_{\mathbf{q}}$, by Markov's inequality the probability that the estimator's returned value is larger than $\mu_i$ is at most $1/4$. By taking $O(\log n_1)$ independent estimates, one can conclude that $\mu_i$ is higher than $\mu_{\mathbf{q}}$ if the median of the estimated values is below $\mu_i$, and this estimate is correct with high probability. This is achieved on Lines 18–19 of Algorithm 3. To ensure that we find a value of $\mu_i$ that satisfies $\mu_i/4 < \mu_{\mathbf{q}} \leqslant \mu_i$ with high probability, on Lines 18–24 the algorithm starts with $\mu_i = 2n_1$ and repeatedly halves the estimate until finding an estimate $\hat{\mu}_{\mathbf{q}} > \mu_i$; at this point the algorithm returns the previous estimate based on $\mu_{i+1}$.*

**Lemma B.4** (Charikar et al. (2020))**.** *For every* $\mathbf{q} \in \mathbb{R}^d$, $\mu_\mathbf{q} \in (0, 2n_1]$, $\varepsilon \in \left(1/n_1^5, 1\right)$, *and* $\mu_i$ *satisfying* $\mu_i/4 \leqslant \mu_\mathbf{q} \leqslant \mu_i$, *the procedure* QUERYMUESTIMATE$(X, \mathbf{q}, \varepsilon, \mu_i)$ *(Algorithm 3 in the appendix) outputs a* $(1 \pm \varepsilon)$*-approximation to* $\mu_\mathbf{q}$ *with high probability.*

*Proof.* Let $Z \triangleq Z_{\mathbf{q},a}$, and we have that

$$
\mathbb{E}[Z^2] = \mathbb{E}\left[\left(\sum_{\mathbf{p}_i \in X} \chi_i \cdot \frac{w_i}{p_i}\right)^2\right]
$$

$$
= \sum_{i \neq j} \mathbb{E}\left[\chi_i \chi_j \cdot \frac{w_i w_j}{p_i p_j}\right] + \sum_{i \in [n_1]} \mathbb{E}\left[\chi_i \cdot \frac{w_i^2}{p_i^2}\right]
$$

$$
\leqslant \sum_{i \neq j} w_i w_j + \sum_{i \in [n_1]} \frac{w_i^2}{p_i} \cdot \mathbb{I}[p_i = 1] + \sum_{i \in [n_1]} \frac{w_i^2}{p_i} \cdot \mathbb{I}[p_i \neq 1]
$$

$$
\leqslant \left(\sum_{i \in [n_1]} w_i\right)^2 + \max_{i \in [n_1]} \left\{\frac{w_i}{p_i} \cdot \mathbb{I}[p_i \neq 1]\right\} \sum_{i \in [n_1]} w_i
$$

$$
\leqslant 2\left(\mu_\mathbf{q}\right)^2 + \max_{j \in [J_{\mu_i}], \mathbf{p}_i \in L_j^\mathbf{q}} \left\{w_i \cdot 2^{j+1}\right\} \cdot \mu_i \cdot \mu_\mathbf{q}
$$

$$
\leqslant 8\mu_i^2, \tag{B.1}
$$

where the second inequality follows from

$$
\frac{w_i^2}{p_i} \cdot \mathbb{I}[p_i = 1] \leqslant w_i^2
$$

and

$$
\left(\sum_i w_i\right)^2 = \sum_{i \neq j} w_i w_j + \sum_{i \in [n_1]} w_i^2,
$$

and the third inequality follows from $(\mu_\mathbf{q})^2 = (\sum_i w_i)^2$ and $p_j \geqslant 1/\left(2^{j+1}\mu_i\right)$.

Let

$$
\bar{Z} \triangleq \bar{Z}_{\mathbf{q},b} = \frac{1}{\bar{m}} \sum_{a=(b-1)\bar{m}+1}^{b\bar{m}} Z_{\mathbf{q},a}
$$

be the empirical mean of $\bar{m}$ such estimates, as computed on Line 19 of Algorithm 3. We have that

$$
\mathbb{P}\left[|\bar{Z} - \mu_\mathbf{q}| \geqslant \varepsilon\mu_\mathbf{q}\right] \leqslant \mathbb{P}\left[|\bar{Z} - \mathbb{E}[Z]| \geqslant \varepsilon\mu_\mathbf{q} - |\mathbb{E}[Z] - \mu_\mathbf{q}|\right]
$$

$$
\leqslant \mathbb{P}\left[|\bar{Z} - \mathbb{E}[Z]| \geqslant (\varepsilon - n_1^{-9})\mu_\mathbf{q}\right]
$$

$$
\leqslant \frac{\mathbb{E}[\bar{Z}^2]}{\left(\varepsilon - n_1^{-9}\right)^2 (\mu_\mathbf{q})^2}
$$

$$
\leqslant \frac{1}{\bar{m}} \frac{128(\mu_\mathbf{q})^2}{\left(\varepsilon - n_1^{-9}\right)^2 (\mu_\mathbf{q})^2},
$$

where the first inequality follows from $|\bar{Z} - \mu_\mathbf{q}| \leqslant |\bar{Z} - \mathbb{E}[Z]| + |\mathbb{E}[Z] - \mu_\mathbf{q}|$, the second one follows from $\mathbb{E}[Z] \geqslant (1 - n_1^{-9})\mu_\mathbf{q}$ (Lemma B.3), the third one follows from Chebyshev's inequality and the last one follows from $\mathbb{E}[\bar{Z}^2] \leqslant \mathbb{E}[Z^2]/\bar{m} \leqslant 8\mu_i^2/\bar{m}$ and $\mu_i \leqslant 4\mu_\mathbf{q}$. By setting $\bar{m} = \frac{C}{\varepsilon^2}$ for a large enough constant $C$ and taking the median of $O(\log(1/\delta))$ of these means we achieve a $(1 \pm \varepsilon)$-approximation with probability at least $1 - \delta$ per query. $\qquad\square$

## B.2 ANALYSIS OF THE ADDDATAPOINT PROCEDURE

We now analyse the ADDDATAPOINT($\mathbf{z}$) procedure in Algorithm 1. If the number of data points has doubled, ADDDATAPOINT($\mathbf{z}$) calls the INITIALISE$(X, Q, \varepsilon)$ procedure. Otherwise, ADDDATAPOINTANDUPDATEQUERIES$(\mathbf{z}, Q, \varepsilon)$ is called, which we describe in Algorithm 6.

---

**Algorithm 6** DynamicKDE Update Procedures

---

1: **procedure** ADDPOINTANDUPDATEQUERIES($\mathbf{z}, Q, \varepsilon$)
2:     **Input:** New data point $\mathbf{z}$, the set of query points $Q$, and the precision estimate $\varepsilon$.
3:     **for** $\mu_i \in M$ **do**
4:         **for** $a = 1, 2, \ldots, K_1$ **do**
5:             **for** $j = 1, 2, \ldots, J_{\mu_i}$ **do**
6:                 $p_{\text{sampling}} \leftarrow \min\left\{\frac{1}{2^{j+1}\mu_i}, 1\right\}$
7:                 **if** $\mathbf{z}$ is sampled with probability $p_{\text{sampling}}$ **then**
8:                     $K_{2,j} \leftarrow 200 \log n' \cdot p_{\text{near},j}^{-k_j}$
9:                     **for** $\ell = 1, 2, \ldots, K_{2,j}$ **do**
10:                       Store $\mathbf{z}$ in bucket $B_{H_{\mu_i, a, j, \ell}}(\mathbf{z})$
11:                       Recover $\mathbf{q} \in B_{H_{\mu_i, a, j, \ell}}^*(\mathbf{z})$ such that $\mathbf{q} \in Q_{\mu_i} \setminus \bigcup_{j' < i} Q_{\mu_{j'}}$, and $\mathbf{q} \in L_j^{\mathbf{z}}$
12:         Sample $\mathbf{z}$ with probability $\frac{1}{2n'}$.
13:         **if** $\mathbf{z}$ is sampled **then**
14:             Add $\mathbf{z}$ to $\widetilde{X}_{\mu_i, a}$
15:             Recover $\mathbf{q} \in Q$ such that $\mathbf{q} \in Q_{\mu_i} \setminus \bigcup_{j' < i} Q_{\mu_{j'}}$, and $\mathbf{q} \in L_{J_{\mu_i}+1}^{\mathbf{z}}$
16:     $S \leftarrow$ Set of all recovered points from the full hash in the current iteration
17:     **for** $\mathbf{q} \in S$ **do**
18:         $w_{\mathbf{q}} \leftarrow k(\mathbf{z}, \mathbf{q})$
19:         **if** $\mathbf{z} \in L_j^{\mathbf{q}}$ for some $j \in [J_{\mu_i}]$ **then**
20:             $p_{\mathbf{q}} \leftarrow \min\left\{\frac{1}{2^{j+1}\mu_i}, 1\right\}$,
21:         **else if** $\mathbf{z} \in X \setminus \bigcup_{j \in [J_{\mu_i}]} L_j^{\mathbf{q}}$ **then**
22:             $p_{\mathbf{q}} \leftarrow \frac{1}{2n'}$
23:         $Z_{\mathbf{q}, a} \mathrel{+}= w_{\mathbf{q}}/p_{\mathbf{q}}$
24:         $\bar{Z}_{\mathbf{q}, \lceil a/\bar{m} \rceil} \mathrel{+}= w_{\mathbf{q}}/(\bar{m} p_{\mathbf{q}})$         $\triangleright$ Update empirical mean (Line 19, Algorithm 3))
25:         $\hat{\mu}_{\mathbf{q}} \leftarrow \text{Median}\left(\bar{Z}_{\mathbf{q},1}, \bar{Z}_{\mathbf{q},2}, \ldots, \bar{Z}_{\mathbf{q},\bar{N}}\right)$         $\triangleright$ Update median
26:         **if** $\hat{\mu}_{\mathbf{q}} > \mu_i$ **then**
27:             $Q_{\mu_i} \leftarrow Q_{\mu_i} \setminus \{\mathbf{q}\}$
28:             Remove $\mathbf{q}$ from every $B_{H_{\mu_i, a', j', \ell'}}^*(\mathbf{q})$ for all $a' \in [K_1], j' \in [J_{\mu_i}]$, and $\ell' \in K_{2,j}$.
29:             $\hat{\mu}_{\mathbf{q}} \leftarrow \text{QUERYPOINT}(X, \mathbf{q}, \varepsilon)$

---

We first show that ADDDATAPOINT($\mathbf{z}$) procedure updates the estimates correctly.

**Lemma B.5.** *After running the* ADDDATAPOINT($\mathbf{z}$) *procedure, it holds with high probability for every* $\mathbf{q} \in Q$ *that* $\hat{\mu}_{\mathbf{q}}$ *is a* $(1 \pm \varepsilon)$-*approximation of* $\mu_{\mathbf{q}}$.

*Proof.* We prove that running the initialisation procedure INITIALISE($X \cup \mathbf{z}, Q, \varepsilon$) is the same as running INITIALISE($X, Q, \varepsilon$), and then running ADDDATAPOINT($\mathbf{z}$).

- We first examine the ADDPOINTANDUPDATEQUERIES($\mathbf{z}, Q, \varepsilon$) procedure (Algorithm 6). On Lines 7–10 and Lines 12–14, the algorithm samples the point $\mathbf{z}$ with probability $\min\left\{\frac{1}{2^{j+1}\mu_i}, 1\right\}$ and $\frac{1}{2n'}$, respectively: If $\mathbf{z}$ is sampled in Lines 7–10, the algorithm stores $\mathbf{z}$ in the bucket corresponding to hash value $H_{\mu_i, a, j, \ell}(\mathbf{z})$; if $\mathbf{z}$ is sampled in Lines 12–14, the algorithm adds $\mathbf{z}$ to the set $\widetilde{X}_{\mu_i, a}$. This is the same as sampling and storing $\mathbf{z}$ in the PREPROCESS($X \cup \mathbf{z}, \varepsilon$) procedure (Algorithm 2) on Lines 8 and 13, which is called during INITIALISE($X \cup \mathbf{z}, Q, \varepsilon$). Hence, after running INITIALISE($X, Q, \varepsilon$) followed by ADDPOINTANDUPDATEQUERIES($\mathbf{z}, Q, \varepsilon$), the stored points in $H_{\mu_i, a, j, \ell}$ and $\widetilde{X}_{\mu_i, a}$ for all $\mu_i \in M$, $a \in [K_1]$, $j \in [J_{\mu_i}]$, and $\ell \in [K_{2,j}]$ are the same as the ones after running INITIALISE($X \cup \mathbf{z}, Q, \varepsilon$).

- Next, we prove that the estimates $\hat{\mu}_{\mathbf{q}}$ are updated correctly for every $\mathbf{q} \in Q$. Without loss of generality, let $\mathbf{q}$ be a query point such that $\mathbf{q} \in Q_{\mu_i} \setminus \bigcup_{j' < i} Q_{\mu_{j'}}$.

    – We first note that, when running INITIALISE$(X, Q, \varepsilon)$ (Algorithm 1), the KDE estimate for $\mathbf{q}$ is returned by running QUERYMUESTIMATE$(X, \mathbf{q}, \varepsilon, \mu_i)$ (Line 24 of Algorithm 3).

    – When running INITIALISE$(X \cup \mathbf{z}, Q, \varepsilon)$, if $\mathbf{z}$ is sampled on Line 8 during PREPROCESS$(X \cup \mathbf{z}, \varepsilon)$ for any iteration $a \in [K_1]$, $j \in [J_{\mu_i}]$, then $\mathbf{z}$ is stored in the bucket $B_{H_{\mu_i,a,j,\ell}}(\mathbf{z})$ for all $\ell \in [K_{2,j}]$. Moreover, if $\mathbf{z}$ is sampled on Line 13 for any iteration $a \in [K_1]$, then $\mathbf{z}$ is stored in $\widetilde{X}_{\mu_i,a}$. In this case, if $H_{\mu_i,a,j,\ell}(\mathbf{q}) = H_{\mu_i,a,j,\ell}(\mathbf{z})$ and $\mathbf{z} \in L_j^{\mathbf{q}}$ for some $\ell \in [K_{2,j}]$, or if $\mathbf{z} \in L_{J_{\mu_i}+1}^{\mathbf{q}}$, then $\mathbf{z}$ would be included in the set of recovered points $S$ for the iteration $a \in [K_1]$, and consequently in the estimator $Z_{\mathbf{q},a}$ when QUERYMUESTIMATE$(X \cup \mathbf{z}, \mathbf{q}, \varepsilon, \mu_i)$ is called during PREPROCESSQUERYPOINTS$(X \cup \mathbf{z}, Q, \varepsilon)$ (Algorithm 4).

    – On the other hand, we notice that, when running ADDPOINTANDUPDATEQUERIES$(\mathbf{z}, Q, \varepsilon)$, $\mathbf{q}$ is recovered if (i) $H_{\mu_i,a,j,\ell}(\mathbf{q}) = H_{\mu_i,a,j,\ell}(\mathbf{z})$ and $\mathbf{q} \in L_j^{\mathbf{z}}$ (Line 11 of Algorithm 6) or (ii) $\mathbf{q} \in L_{J_{\mu_i}+1}^{\mathbf{z}}$ (Line 15 of Algorithm 6). Furthermore, (i) $\mathbf{q} \in L_{j'}^{\mathbf{z}}$ if and only if $\mathbf{z} \in L_{j'}^{\mathbf{q}}$ for any $j' \in [J_{\mu_i} + 1]$, and (ii) the buckets $B_{H_{\mu_i,a,j,\ell}}^*(\mathbf{q})$ and $B_{H_{\mu_i,a,j,\ell}}(\mathbf{q})$ are populated using the same hash function (Line 13 of Algorithm 4). Therefore, $\mathbf{q}$ is recovered at iteration $a \in [K_1]$ when running ADDPOINTANDUPDATEQUERIES$(\mathbf{z}, Q, \varepsilon)$ if and only if $\mathbf{z}$ is included in the estimator $Z_{\mathbf{q},a}$ when QUERYMUESTIMATE$(X \cup \mathbf{z}, \mathbf{q}, \varepsilon, \mu_i)$ is called during PREPROCESSQUERYPOINTS$(X \cup \mathbf{z}, Q, \varepsilon)$. Then, the estimator $Z_{\mathbf{q},a}$ is updated accordingly by adding $\mathbf{z}$ through Lines 17–25 of Algorithm 6 as it would be done through Lines 10–19 of Algorithm 3.

    – Finally, if $\hat{\mu}_{\mathbf{q}} > \mu_i$, then we re-estimate the query point on Line 23, to ensure we have the correct estimate $\hat{\mu}_{\mathbf{q}}$. We also update the set $Q_{\mu_i}$ accordingly, and remove $\mathbf{q}$ from every $B_{H_{\mu_i,a',j',\ell'}}^*(\mathbf{z})$ for all $a' \in [K_1]$, $j' \in [J_{\mu_i}]$, and $\ell' \in [K_{2,j}]$.

Combining everything together, we have shown that performing the initialisation procedure INITIALISE$(X \cup \mathbf{z}, Q, \varepsilon)$ is the same as running INITIALISE$(X, Q, \varepsilon)$, followed by ADDDATAPOINT$(\mathbf{z})$. $\qquad\square$

Next, we prove how many times any individual query point $\mathbf{q}$ is updated as the data points are inserted using the ADDDATAPOINT$(\mathbf{z})$ procedure. Let $X_1^{\mathbf{q}} \triangleq \{\mathbf{x}_1, \ldots, \mathbf{x}_{n_1}\}$ be the set of points presented at the time when $\mathbf{q}$ is added, and let $Z_T \triangleq \{\mathbf{z}_1, \ldots, \mathbf{z}_T\}$ be the points added up until the query time $T$. We use $\mathbf{z}_t$ to denote the new point added at time $t$. Note that it holds for the points $X_T$ at time $T \geqslant 1$ that $X_T^{\mathbf{q}} = X_1^{\mathbf{q}} \cup Z_T$. Next, we define the event $\mathcal{F}_t^{\mathbf{q}}$ that

$$\widehat{\mu}_{\mathbf{q},t} \in (1 \pm \varepsilon) \cdot k(\mathbf{q}, X_t), \tag{B.2}$$

where $\widehat{\mu}_{\mathbf{q},t}$ is the maintained query estimate for $\mathbf{q}$ at time $t$ from Algorithm 1. By Lemma B.4, we know that $\mathcal{F}_t^{\mathbf{q}}$ happens with high probability. Moreover, for a large enough constant $C$ on Line 6 of Algorithm 1, we can ensure that this happens with high probability at every time step $t$. Therefore in the following we assume $\mathcal{F}_t^{\mathbf{q}}$ happens. We also introduce the following notation.

**Definition B.6.** *We define $T_{\mu_i}^{\mathbf{q}}$ to be the time step such that*

$$k(\mathbf{q}, X_{T_{\mu_i}^{\mathbf{q}}}^{\mathbf{q}}) = k(\mathbf{q}, X_1^{\mathbf{q}}) + \sum_{t=1}^{T_{\mu_i}^{\mathbf{q}}} k(\mathbf{q}, \mathbf{z}_t) \leqslant \mu_i$$

*and*

$$k(\mathbf{q}, X_{T_{\mu_i}^{\mathbf{q}}+1}^{\mathbf{q}}) = k(\mathbf{q}, X_1^{\mathbf{q}}) + \sum_{t=1}^{T_{\mu_i}^{\mathbf{q}}+1} k(\mathbf{q}, \mathbf{z}_t) > \mu_i.$$

By definition, $T_{\mu_i}^{\mathbf{q}}$ is the last time step at which the KDE value of $\mathbf{q}$ is at most $\mu_i$. The next lemme analyses the number of times a query point $\mathbf{q}$ is updated.

**Lemma B.7.** *Let* $\mathbf{q}$ *be a maintained query point by our algorithm. Then the total number of updates* $\mathcal{U}_T^{\mathbf{q}}$ *during* $T$ *insertions is, with high probability,* $\mathcal{U}_T^{\mathbf{q}} = \widetilde{O}(1)$.

*Proof.* Since the KDE data structure can be re-initialised at most $\log(T) = \widetilde{O}(1)$ times (cf. Line 19 of Algorithm 1) through the sequence of $T$ updates, it suffices for us to analyse the number of times $\mathbf{q}$ is updated between different re-initialisations; we assume this in the remaining part of the proof. To analyse the expected number of times that $\mathbf{q}$ (Line 23 of Algorithm 6) is updated throughout the sequences of updates $Z_T$, we define the random variable $Y_{a,t}^{\mathbf{q}}$ by

$$Y_{a,t}^{\mathbf{q}} \triangleq \begin{cases} 1 & \text{if estimate } Z_{\mathbf{q},a} \text{ is updated at time } t \\ 0 & \text{otherwise.} \end{cases}$$

Let $\mathcal{E}_{a,t}^{\mathbf{q}}$ be the event that estimate $Z_{\mathbf{q},a}$ is updated at time $t$, and we assume without loss of generality that $T_{\mu_{i'}-1}^{\mathbf{q}} < t \leqslant T_{\mu_{i'}}^{\mathbf{q}}$ for some $\mu_{i'}$. First note that estimate $Z_{\mathbf{q},a}$ is updated if $\mathbf{q} \in S$ (Line 16 of Algorithm 6). Furthermore, because $\widehat{\mu}_{\mathbf{q},t-1} \in (1 \pm \varepsilon) \cdot k(\mathbf{q}, X_{t-1})$ by (B.2) and $T_{\mu_{i'}-1}^{\mathbf{q}} < t \leqslant T_{\mu_{i'}}^{\mathbf{q}}$, one of the following holds:

(i) $\mathbf{q} \in Q_{\mu_{i'}} \setminus \left( \bigcup_{j' < i'} Q_{\mu_{j'}} \right)$;

(ii) $\mathbf{q} \in Q_{\mu_{i'-1}} \setminus \left( \bigcup_{j' < i'-1} Q_{\mu_{j'}} \right)$;

(iii) $\mathbf{q} \in Q_{\mu_{i'+1}} \setminus \left( \bigcup_{j' < i'+1} Q_{\mu_{j'}} \right)$.

Additionally, it holds that $\mathbf{q} \in L_{j'}^{\mathbf{z}_t}$ for some $j' \in J_{\mu_{i'}}$.

By these conditions, $\mathbf{q}$ is included in $S$ at time $t$ if and only if $\mathbf{z}_t$ is sampled at either the iteration for $\mu_{i'} \in M$, $\mu_{i'-1} \in M$ or $\mu_{i'+1} \in M$ (Line 3 of Algorithm 6), and the corresponding iteration $j' \in J_{\mu_{i'+1}}$ on Line 7 of Algorithm 6. Therefore, it holds for $t \in (T_{\mu_{i'}-1}^{\mathbf{q}}, T_{\mu_{i'}}^{\mathbf{q}}]$ that

$$\mathbb{P}\left[ \mathcal{E}_{a,t}^{\mathbf{q}} \right] \leqslant \frac{1}{2^{j'+1} \cdot \mu_{i'-1}} \leqslant \frac{1}{2^{j'} \cdot \mu_{i'}} \leqslant \frac{2k(\mathbf{q}, \mathbf{z}_t)}{\mu_{i'}}, \tag{B.3}$$

where the last inequality uses the fact that $\mathbf{q} \in L_{j'}^{\mathbf{z}_t}$ (Definition 2.3). Similarly, we have that

$$\mathbb{P}\left[ \mathcal{E}_{a,t}^{\mathbf{q}} \right] \geqslant \frac{k(\mathbf{q}, \mathbf{z}_t)}{4\mu_{i'}}.$$

Let $\mathcal{U}_T^{\mathbf{q}} \triangleq \sum_{a=1}^{K_1} \sum_{t=1}^{T} Y_{a,t}^{\mathbf{q}}$ be the total number of times that the query point $\mathbf{q}$ is updated, and we have that

$$\mathbb{E}\left[ \mathcal{U}_T^{\mathbf{q}} \right] = \sum_{a=1}^{K_1} \sum_{t=1}^{T} \mathbb{E}\left[ Y_{a,t}^{\mathbf{q}} \right]$$

$$= \sum_{a=1}^{K_1} \sum_{t=1}^{T} \mathbb{P}\left[ \mathcal{E}_{a,t}^{\mathbf{q}} \right]$$

$$= \sum_{a=1}^{K_1} \sum_{\mu_{i'} \in M} \sum_{t=T_{\mu_{i'}-1}^{\mathbf{q}}}^{T_{\mu_{i'}}^{\mathbf{q}}} \mathbb{P}\left[ \mathcal{E}_{a,t}^{\mathbf{q}} \right]$$

$$\leqslant \sum_{a=1}^{K_1} \sum_{\mu_{i'} \in M} \sum_{t=T_{\mu_{i'}-1}^{\mathbf{q}}}^{T_{\mu_{i'}}^{\mathbf{q}}} \frac{2k(\mathbf{q}, \mathbf{z}_t)}{\mu_{i'}}$$

$$\leqslant \sum_{a=1}^{K_1} \sum_{\mu_{i'} \in M} \frac{2\mu_{i'}}{\mu_{i'}}$$

$$= 2 \cdot K_1 \cdot |M|$$

$$= \widetilde{O}(1),$$

where the first inequality follows by (B.3) and the second one holds by the fact that

$$\sum_{t=T^{\mathbf{q}}_{\mu_{i'}-1}}^{T^{\mathbf{q}}_{\mu_{i'}}} k(\mathbf{q}, \mathbf{z}_t) \leqslant \mu_{i'}.$$

Similarly, we have that

$$\mathbb{E}\left[\mathcal{U}^{\mathbf{q}}_T\right] \geqslant \frac{1}{4} \cdot K_1 \cdot |M|.$$

By the Chernoff bound, it holds that

$$\mathbb{P}\left[\mathcal{U}^{\mathbf{q}}_T \geqslant 10 \cdot \mathbb{E}\left[\mathcal{U}^{\mathbf{q}}_T\right]\right] \leqslant \left(\frac{\mathrm{e}^9}{10^{10}}\right)^{K_1 \cdot |M|/4} \leqslant \exp\left(-K_1 \cdot |M|\right) = o(n^{-c})$$

for some constant $c$, and we have with high probability that

$$\mathcal{U}^{\mathbf{q}}_T \leqslant 20 \cdot K_1 \cdot |M| = \widetilde{O}(1),$$

which proves the statement. $\qquad\square$

### B.3 Proof of Theorem 3.1

*Proof.* We start with proving the first statement. Notice that PREPROCESS$(X, \varepsilon)$ goes through $M \cdot K_1 \cdot J_{\mu_i} \cdot K_{2,j}$ iterations, where $M = O(\log(n_1))$, $K_1 = O(\varepsilon^{-2} \cdot \log(n_1))$, $J_{\mu_i} = O(\log(n_1))$ and $K_{2,j} = O(\log(n_1) \cdot \mathrm{cost}(k))$. Since $k_j = \widetilde{O}(1)$ by definition, the algorithm concatenates $\widetilde{O}(1)$ LSH functions. By Lemma 2.1, the evaluation time of $H^*(\mathbf{x})$ for any $H^* \in \mathcal{H}^{k_j}$ is $n_1^{o(1)}$, and hashing all $n_1$ points yields the running time of $\varepsilon^{-2} \cdot n_1^{1+o(1)} \cdot \mathrm{cost}(k)$ for PREPROCESS$(X, \varepsilon)$ in the worst case. Since we start with an empty set of query points $Q = \emptyset$, the running time of PREPROCESSQUERYPOINTS$(X, Q, \varepsilon)$ can be omitted. This proves the first statement.

The guarantees for ADDQUERYPOINT$(\mathbf{q})$ in the second statement follow from Lemma B.2 and Lemma B.4. The running time for DELETEQUERYPOINT$(\mathbf{q})$ follows from the running time guarantee for ADDQUERYPOINT$(\mathbf{q})$.

Now we prove the third statement. The correctness of the updated estimate of $\mu_{\mathbf{q}}$ follows from Lemma B.5. To prove the time complexity, we notice that, when running ADDDATAPOINT$(\mathbf{z})$, the procedure goes through $M \cdot K_1 \cdot J_{\mu_i} \cdot K_{2,j}$ iterations, where $M = O(\log n)$, $K_1 = O(\log(n)/\varepsilon^2)$, $J_{\mu_i} = O(\log n)$, and $K_{2,j} = O(\log(n) \cdot \mathrm{cost}(k))$. In the worst case, $\mathbf{z}$ is sampled in every iteration on Line 7 of Algorithm 6, and needs to be stored in the bucket $B_{H_{\mu_i,a,j,\ell}}(\mathbf{z})$. Therefore, the running time of updating all the buckets $B_{H_{\mu_i,a,j,\ell}}(\mathbf{z})$ for a new $\mathbf{z}$ is at most $\varepsilon^{-2} \cdot n^{o(1)} \cdot \mathrm{cost}(k)$. To analyse Lines 17–29 of Algorithm 6, we perform an amortised analysis. By Lemma B.7, it holds with high probability that every $\mathbf{q} \in Q$ is updated $\widetilde{O}(1)$ times throughout the sequence of data point updates. When $\mathbf{q} \in Q$ is updated, the total running time for Lines 17–29 is

$$\widetilde{O}(\varepsilon^{-2} \cdot K_{2,j} + \varepsilon^{-2} \cdot \mathrm{cost}(k)) = \widetilde{O}(\varepsilon^{-2} \cdot \mathrm{cost}(k)),$$

due to Lines 28 and 29. Let $T$ be the total number of query and data point insertions at any point throughout the sequence of updates, and $m \triangleq |Q|$. Then the amortised update time is

$$\widetilde{O}\left(\frac{m \cdot \varepsilon^{-2} \cdot \mathrm{cost}(k)}{T}\right) = \widetilde{O}\left(\frac{m \cdot \varepsilon^{-2} \cdot \mathrm{cost}(k)}{m}\right) = \widetilde{O}\left(\varepsilon^{-2} \cdot \mathrm{cost}(k)\right),$$

where the second inequality follows from $T \geqslant m$, as the algorithm started with an empty query set $Q = \emptyset$. Combining everything together proves the running time. $\qquad\square$

## C Omitted Detail from Section 4

This section presents all the detail omitted from Section 4, and is organised as follow: Section C.1 presents and analyses the algorithm for the initialisation step; Section C.2 presents and analyses the algorithm for the dynamic update step.

## C.1 THE INITIALISATION STEP

In this subsection we present the algorithms used in the initialisation step, and analyse its correctness as well as complexity. The following tree data structure will be used in the design of our procedures.

---

**Algorithm 7** Tree Data Structure

---

1: LEAF($\mathbf{x}_i$)
2:     **Input:** data point $\mathbf{x}_i$
3:     data $\leftarrow \mathsf{x}_i$                                                   $\triangleright$ Stores the data point
4:     paths $\leftarrow$ NIL                          $\triangleright$ Stores the sampling paths ending at this leaf
5: NODE($X$)
6:     **Input:** set of data points $X$
7:     data $\leftarrow X$                    $\triangleright$ Stores the data points in the subtree rooted at this node
8:     size $\leftarrow |X|$               $\triangleright$ Number of data points in the subtree rooted at this node
9:     kde $\leftarrow$ NIL                              $\triangleright$ Stores the DynamicKDE structure
10:    left $\leftarrow$ NIL                                      $\triangleright$ Left child node
11:    right $\leftarrow$ NIL                                 $\triangleright$ Right child node
12:    parent $\leftarrow$ NIL                              $\triangleright$ Parent node
13:    paths $\leftarrow$ NIL                   $\triangleright$ Stores the sampling paths passing through this node

---

Based on this data structure, the main procedures used for constructing an approximate similarity graph in the initialisation step are presented in Algorithm 8. We remark that, for any set $X$ of data points, we always set $\varepsilon = 1/\log^3 |X|$ when running the DYANMICKDE.INITIALISE procedure. Choosing a fixed value of $\varepsilon$ in this section allows us to simplify the presentation of the analysis without loss of generality.

To analyse the correctness and time complexity of the algorithm, we first prove that, for any data point $\mathbf{x}_i \in X$, the probability that its sampling path $\mathcal{P}_{\mathbf{x}_i,\ell}$ passes through any internal node $\mathcal{T}'$ depends on the KDE value of $\mathbf{x}_i$ with respect to $\mathcal{T}'.\mathsf{data}$.

**Lemma C.1.** *For any point $\mathbf{x}_i \in X$, tree $\mathcal{T}$ constructed by* CONSTRUCTGRAPH *(Algorithm 8), and sampling path $\mathcal{P}_{\mathbf{x}_i,\ell}$ (for any $\ell \in [L]$), the probability that $\mathcal{P}_{\mathbf{x}_i,\ell}$ passes through any internal node $\mathcal{T}'$ of $\mathcal{T}$ is given by*

$$\frac{k(\mathbf{x}_i, \mathcal{T}'.\mathsf{data})}{2k(\mathbf{x}_i, \mathcal{T}.\mathsf{data})} \leqslant \mathbb{P}\left[\mathcal{P}_{\mathbf{x}_i,\ell} \in \mathcal{T}'.\mathsf{paths}\right] \leqslant \frac{2k(\mathbf{x}_i, \mathcal{T}'.\mathsf{data})}{k(\mathbf{x}_i, \mathcal{T}.\mathsf{data})}.$$

*Proof.* Let $X = \{\mathbf{x}_1, \ldots, \mathbf{x}_{n_1}\}$ be the input data points for the CONSTRUCTGRAPH$(X, \varepsilon)$ procedure in Algorithm 8. Then, in each recursive call (at some internal root $\mathcal{T}''$) to SAMPLE (Algorithm 8) we are given the data points $X_L \triangleq \mathcal{T}''.\mathsf{left.data}$ and $X_R \triangleq \mathcal{T}''.\mathsf{left.data}$ as input and assign $\mathcal{P}_{\mathbf{x}_i,\ell}$ to either $\mathcal{T}''_L \triangleq \mathcal{T}''.\mathsf{left}$ or $\mathcal{T}''_R \triangleq \mathcal{T}''.\mathsf{right}$. By Line 31 of Algorithm 8, we have that the probability of assigning $\mathcal{P}_{\mathbf{x}_i,\ell}$ to $\mathcal{T}''_L.\mathsf{paths}$ is

$$\mathbb{P}\left[\mathcal{P}_{\mathbf{x}_i,\ell} \in \mathcal{T}''_L.\mathsf{paths} \mid \mathcal{P}_{\mathbf{x}_i,\ell} \in \mathcal{T}''.\mathsf{paths}\right] = \frac{\mathcal{T}''_L.\hat{\mu}_{\mathbf{x}_i}}{\mathcal{T}''.\hat{\mu}_{\mathbf{x}_i}}.$$

By the performance guarantee of the KDE algorithm (Theorem 3.1), we have that $\mathcal{T}''.\hat{\mu}_{\mathbf{x}_i} \in (1 \pm \varepsilon) \cdot k(\mathcal{T}''.\mathsf{data}, \mathbf{x}_i)$. This gives

$$\left(\frac{1-\varepsilon}{1+\varepsilon}\right)\frac{k(\mathbf{x}_i, \mathcal{T}''_L.\mathsf{data})}{k(\mathbf{x}_i, \mathcal{T}''.\mathsf{data})} \leqslant \mathbb{P}\left[\mathcal{P}_{\mathbf{x}_i,\ell} \in \mathcal{T}''_L.\mathsf{paths} \mid \mathcal{P}_{\mathbf{x}_i,\ell} \in \mathcal{T}''.\mathsf{paths}\right]$$

$$\leqslant \left(\frac{1+\varepsilon}{1-\varepsilon}\right)\frac{k(\mathbf{x}_i, \mathcal{T}''_L.\mathsf{data})}{k(\mathbf{x}_i, \mathcal{T}''.\mathsf{data})}. \tag{C.1}$$

Next, notice that it holds for a sequence of internal nodes $\mathcal{T}_1, \mathcal{T}_2, \ldots, \mathcal{T}_r$ with $\mathcal{T}_i.\mathsf{parent} = \mathcal{T}_{i+1}$ ($1 \leqslant i \leqslant r - 1$) that

$$\mathbb{P}\left[\mathcal{P}_{\mathbf{x}_i,\ell} \in \mathcal{T}_1.\mathsf{paths}\right] = \prod_{1 \leqslant j \leqslant r-1} \mathbb{P}\left[\mathcal{P}_{\mathbf{x}_i,\ell} \in \mathcal{T}_j.\mathsf{paths} \mid \mathcal{P}_{\mathbf{x}_i,\ell} \in \mathcal{T}_{j+1}.\mathsf{paths}\right],$$

---

**Algorithm 8** Initialisation Procedures for Constructing an Approximate Similarity Graph

---

1: **procedure** INITIALISETREE($X$)
2:     **Input**: set $X$ of data points
3:     **if** $|X| = 1$ **then**
4:         **return** LEAF($X$)                         $\triangleright$ Leaves store individual data points $\mathbf{x}_i \in X$
5:     **else**
6:         $\mathcal{T} \leftarrow$ NODE($X$)
7:         $m \leftarrow 2^{\lfloor \log(|X|/2) \rfloor}, \hat{n} \leftarrow |X|$           $\triangleright$ Nearest power of 2 less than or equal to $\hat{n}/2$
8:         $X_L \leftarrow X[1 : \hat{n} - m], X_R \leftarrow X[\hat{n} - m + 1 : \hat{n}]$         $\triangleright$ Split the dataset into two
9:         $\mathcal{T}_L \leftarrow$ INITIALISETREE($X_L$), $\mathcal{T}_R \leftarrow$ INITIALISETREE($X_R$)
10:        $\mathcal{T}$.left $\leftarrow \mathcal{T}_L, \mathcal{T}$.right $\leftarrow \mathcal{T}_R$
11:        $\mathcal{T}_L$.kde $\leftarrow$ DYNAMICKDE.INITIALISE($X_L, \emptyset, 1/\log^3 n$)         $\triangleright$ (Algorithm 1)
12:        $\mathcal{T}_R$.kde $\leftarrow$ DYNAMICKDE.INITIALISE($X_R, \emptyset, 1/\log^3 n$)
13:        **return** $\mathcal{T}$
14: **procedure** SAMPLE($S, \mathcal{T}, \ell$)
15:     **Input**: set $S$ of $\mathbf{x}_i$, KDE tree $\mathcal{T}$ representing data points $X$, parameter $\varepsilon$, sample index $\ell$
16:     **Output**: $E = \{(\mathbf{x}_i, \mathbf{x}_j) \text{ for some } i \text{ and } j\}$
17:     **for** $\mathbf{x}_i \in S$ **do**
18:         $\mathcal{P}_{\mathbf{x}_i, \ell} \leftarrow \mathcal{P}_{\mathbf{x}_i, \ell} \cup \mathcal{T}$
19:         $\mathcal{T}$.paths $\leftarrow \mathcal{T}$.paths $\cup \{\mathcal{P}_{\mathbf{x}_i, \ell}\}$         $\triangleright$ Update and store sample paths
20:     **if** ISLEAF($\mathcal{T}$) **then**
21:         **return** $S \times \mathcal{T}$.data
22:     **else**
23:         $\mathcal{T}_L = \mathcal{T}$.left, $X_L = \mathcal{T}$.left.data
24:         $\mathcal{T}_R = \mathcal{T}$.right, $X_R = \mathcal{T}$.right.data
25:         **for** $\mathbf{x}_i \in S$ **do**
26:            $\mathcal{T}_L.\hat{\mu}_{\mathbf{x}_i} \leftarrow \mathcal{T}_L$.kde.ADDQUERYPOINT($\mathbf{x}_i$)         $\triangleright$ (Algorithm 1)
27:            $\mathcal{T}_R.\hat{\mu}_{\mathbf{x}_i} \leftarrow \mathcal{T}_R$.kde.ADDQUERYPOINT($\mathbf{x}_i$)
28:         $S_L \leftarrow \emptyset, S_R \leftarrow \emptyset$
29:         **for** $\mathbf{x}_i \in S$ **do**
30:            $r \sim \text{Unif}[0, 1]$
31:            **if** $r \leqslant \mathcal{T}_L.\hat{\mu}_{\mathbf{x}_i}/(\mathcal{T}_L.\hat{\mu}_{\mathbf{x}_i} + \mathcal{T}_R.\hat{\mu}_{\mathbf{x}_i})$ **then**
32:                $S_L \leftarrow S_L \cup \mathbf{x}_i$
33:            **else**
34:                $S_R \leftarrow S_R \cup \mathbf{x}_i$
35:         **return** SAMPLE($S_L, \mathcal{T}_L, \ell$) $\cup$ SAMPLE($S_R, \mathcal{T}_R, \ell$)
36: **procedure** CONSTRUCTGRAPH($X$)
37:     **Input**: set of data points $X$
38:     $\mathcal{T} \leftarrow$ INITIALISETREE($X$)
39:     $\mathcal{T}$.kde $\leftarrow$ DYNAMICKDE.INITIALISE($X, X, 1/\log^3 |X|$)         $\triangleright$ (Algorithm 1)
40:     $E \leftarrow \emptyset$
41:     **for** $\ell \in [L]$ **do**
42:         $E_\ell =$ SAMPLE($X, \mathcal{T}, \ell$)
43:         $E \leftarrow E \cup E_\ell$
44:         **for** $(\mathbf{x}_i, \mathbf{x}_j) \in E_\ell$ **do**
45:            $\widehat{w}(i, j) \leftarrow L \cdot k(\mathbf{x}_i, \mathbf{x}_j)/\min\{\mathcal{T}.\text{kde}.\hat{\mu}_{\mathbf{x}_i}, \mathcal{T}.\text{kde}.\hat{\mu}_{\mathbf{x}_j}\}$
46:            **if** $\min\{\mathcal{T}.\text{kde}.\hat{\mu}_{\mathbf{x}_i}, \mathcal{T}.\text{kde}.\hat{\mu}_{\mathbf{x}_j}\} = \mathcal{T}.\text{kde}.\hat{\mu}_{\mathbf{x}_j}$ **then**
47:                $\mathcal{B}_{\mathbf{x}_j} \leftarrow \mathcal{B}_{\mathbf{x}_j} \cup \mathbf{x}_i$         $\triangleright$ Keep track of the neighbours with higher degree
48:            $w_G(\mathbf{x}_i, \mathbf{x}_j) \mathrel{+}= k(\mathbf{x}_i, \mathbf{x}_j)/\widehat{w}(i, j)$
49:     **return** $\mathcal{T}, G \triangleq (X, E, w_G)$

---

where each term in the right hand side above corresponds to one level of recursion of the SAMPLE procedure in Algorithm 8 and there are at most $\lceil \log_2(n_1) \rceil$ terms. Then, by setting $\mathcal{T}_r = \mathcal{T}, \mathcal{T}_1 = \mathcal{T}'$,

(C.1), and the fact that the denominator and numerator of adjacent terms cancel out, we have

$$\left(\frac{1-\varepsilon}{1+\varepsilon}\right)^{\lceil\log(n_1)\rceil}\frac{k(\mathbf{x}_i,\mathcal{T}'.\mathsf{data})}{k(\mathbf{x}_i,\mathcal{T}.\mathsf{data})} \leqslant \mathbb{P}\left[\mathcal{P}_{\mathbf{x}_i,\ell}\in\mathcal{T}'.\mathsf{paths}\right]$$

$$\leqslant \left(\frac{1+\varepsilon}{1-\varepsilon}\right)^{\lceil\log(n_1)\rceil}\frac{k(\mathbf{x}_i,\mathcal{T}'.\mathsf{data})}{k(\mathbf{x}_i,\mathcal{T}.\mathsf{data})}.$$

For the lower bound, we have that

$$\left(\frac{1-\varepsilon}{1+\varepsilon}\right)^{\lceil\log(n_1)\rceil} \geqslant (1-2\varepsilon)^{\lceil\log(n_1)\rceil} \geqslant 1 - 3\varepsilon\log(n_1) \geqslant 1/2,$$

where the final inequality follows by the condition of $\varepsilon$ that $\varepsilon \leqslant 1/\log^3(n_1)$.

For the upper bound, we similarly have

$$\left(\frac{1+\varepsilon}{1-\varepsilon}\right)^{\lceil\log(n_1)\rceil} \leqslant (1+3\varepsilon)^{\lceil\log(n_1)\rceil} \leqslant \exp\left(3\varepsilon\lceil\log(n_1)\rceil\right) \leqslant e^{2/3} \leqslant 2,$$

where the first inequality follows since $\varepsilon \leqslant 1/\log^3(n_1)$. $\qquad\square$

The remaining part of our analysis is very similar to the proof presented in Macgregor & Sun (2023). For each added edge, CONSTRUCTGRAPH($X$) computes the estimate defined by

$$\widehat{w}(i,j) \triangleq 6C \cdot \frac{\log n_1}{\lambda_{k+1}} \cdot \frac{k(\mathbf{x}_i,\mathbf{x}_j)}{\min\{\hat{\mu}_{\mathbf{x}_i},\hat{\mu}_{\mathbf{x}_j}\}},$$

where for the ease of notation we denote $\mathcal{T}.\mathsf{kde}.\hat{\mu}_{\mathbf{x}_i} \triangleq \hat{\mu}_{\mathbf{x}_i}$ and $\mathcal{T}.\mathsf{kde}.\hat{\mu}_{\mathbf{x}_j} \triangleq \hat{\mu}_{\mathbf{x}_j}$. If an edge $(\mathbf{x}_i,\mathbf{x}_j)$ is sampled, then the edge is included with weight $k(\mathbf{x}_i,\mathbf{x}_j)/\widehat{w}(i,j)$. The algorithm in Macgregor & Sun (2023) is almost the same as our algorithm, with the only difference that in their case the edge is included with weight $k(\mathbf{x}_i,\mathbf{x}_j)/\widehat{p}(i,j)$, where

$$\widehat{p}(i,j) \triangleq 6C \cdot \frac{k(\mathbf{x}_i,\mathbf{x}_j)\cdot\log n_1}{\lambda_{k+1}}\cdot\left(\frac{1}{\hat{\mu}_{\mathbf{x}_i}}+\frac{1}{\hat{\mu}_{\mathbf{x}_j}}\right) - \left(6C\cdot\frac{k(\mathbf{x}_i,\mathbf{x}_j)\cdot\log n_1}{\lambda_{k+1}}\right)^2\cdot\frac{1}{\hat{\mu}_{\mathbf{x}_i}\cdot\hat{\mu}_{\mathbf{x}_j}}.$$

Notice that

$$\widehat{p}(i,j) \leqslant 6C\cdot\frac{k(\mathbf{x}_i,\mathbf{x}_j)\cdot\log n_1}{\lambda_{k+1}}\cdot\left(\frac{1}{\hat{\mu}_{\mathbf{x}_i}}+\frac{1}{\hat{\mu}_{\mathbf{x}_j}}\right) \leqslant 2\widehat{w}(i,j).$$

Assuming without loss of generality that

$$6C\cdot\frac{k(\mathbf{x}_i,\mathbf{x}_j)\cdot\log n_1}{\lambda_{k+1}}\cdot\left(\frac{1}{\hat{\mu}_{\mathbf{x}_i}}+\frac{1}{\hat{\mu}_{\mathbf{x}_j}}\right) < 1,$$

we have that

$$\widehat{p}(i,j) \geqslant 3C\cdot\frac{k(\mathbf{x}_i,\mathbf{x}_j)\cdot\log n_1}{\lambda_{k+1}}\cdot\left(\frac{1}{\hat{\mu}_{\mathbf{x}_i}}+\frac{1}{\hat{\mu}_{\mathbf{x}_j}}\right) \geqslant \frac{\widehat{w}(i,j)}{2}.$$

As such, the scaling factor $\widehat{w}(i,j)$ in our algorithm and $\widehat{p}(i,j)$ in the algorithm of Macgregor & Sun (2023) are within a constant factor of each other. Therefore, to prove the first statement of Theorem 4.1, one can follow the proof of Theorem 2 of Macgregor & Sun (2023), while replacing each $\widehat{p}(i,j)$ with $\widehat{w}(i,j)$ appropriately.

### C.2 THE DYNAMIC UPDATE STEP

In this subsection we present the algorithm used in the dynamic update step, and analyse its correctness as well as complexity. Our main algorithm for dynamically updating an approximate similarity graph is described in Algorithm 9, and the RESAMPLE procedure can be found in Algorithm 10.

---

**Algorithm 9** Dynamic Update Algorithm for Constructing an Approximate Similarity Graph

---

1: **procedure** UPDATEGRAPH($G = (X, E, w_G), \mathcal{T}, \mathbf{z}$)
2:      **Input**: an approximate similarity graph $G$, KDE tree $\mathcal{T}$, new data point $\mathbf{z}$
3:      $\mathcal{A} \leftarrow$ ADDDATAPOINTTREE($\mathcal{T}, \mathbf{z}$)                 ▷ (Algorithm 10)
4:      $E_{\text{new}} \leftarrow \emptyset$
5:      **for** $\ell \in [1, \ldots, L]$ **do**              ▷ Sample $L$ neighbours from the new vertex $\mathbf{z}$
6:          $(\mathbf{z}, \mathbf{x}_j) \leftarrow$ SAMPLE($\{\mathbf{z}\}, \mathcal{T}, \ell$)
7:          $E_{\text{new}} \leftarrow E_{\text{new}} \cup \{(\mathbf{z}, \mathbf{x}_j)\}$
8:          $\widehat{w}(i,j) \leftarrow L \cdot k(\mathbf{z}, \mathbf{x}_j) / \min\{\mathcal{T}.\text{kde}.\hat{\mu}_{\mathbf{z}}, \mathcal{T}.\text{kde}.\hat{\mu}_{\mathbf{x}_j}\}$
9:          $w_G(\mathbf{z}, \mathbf{x}_j) \mathrel{+}= k(\mathbf{z}, \mathbf{x}_j) / \widehat{w}(i,j)$
10:         **if** $\min\{\mathcal{T}.\text{kde}.\hat{\mu}_{\mathbf{z}}, \mathcal{T}.\text{kde}.\hat{\mu}_{\mathbf{x}_j}\} = \mathcal{T}.\text{kde}.\hat{\mu}_{\mathbf{x}_j}$ **then**
11:             $\mathcal{B}_{\mathbf{x}_j} \leftarrow \mathcal{B}_{\mathbf{x}_j} \cup \mathbf{z}$
12:      $E \leftarrow E \cup E_{\text{new}}$
13:      **for** $\mathbf{x}_i$ such that $\mathcal{T}.\text{kde}.\hat{\mu}_{\mathbf{x}_i}$ has changed **do**
14:          Let $\deg_{\text{old}}$ be the old estimate of $\mathcal{T}.\text{kde}.\hat{\mu}_{\mathbf{x}_i}$
15:          **for** $\mathbf{x}_j \in \mathcal{B}_{\mathbf{x}_i}$ **do**
16:             $w_G(\mathbf{x}_i, \mathbf{x}_j) \leftarrow w_G(\mathbf{x}_i, \mathbf{x}_j) \cdot \deg_{\text{old}} / \min\{\mathcal{T}.\text{kde}.\hat{\mu}_{\mathbf{x}_i}, \mathcal{T}.\text{kde}.\hat{\mu}_{\mathbf{x}_j}\}$
                                    ▷ Update scaling factor of adjacent edges
17:            **if** $\min\{\mathcal{T}.\text{kde}.\hat{\mu}_{\mathbf{x}_i}, \mathcal{T}.\text{kde}.\hat{\mu}_{\mathbf{x}_j}\} = \mathcal{T}.\text{kde}.\hat{\mu}_{\mathbf{x}_j}$ **then**
18:               $\mathcal{B}_{\mathbf{x}_i} \leftarrow \mathcal{B}_{\mathbf{x}_i} \setminus \mathbf{x}_j$
19:      **for** $\mathcal{P}_{\mathbf{x}_i, \ell} \in \mathcal{A}$ **do**
20:          Let $\mathcal{T}'$ be the parent of the highest internal node where $\mathcal{P}_{\mathbf{x}_i, \ell}$ was fetched
21:          **for** $\mathcal{T}^*$ below $\mathcal{T}'$ such that $\mathcal{P}_{\mathbf{x}_i, \ell} \in \mathcal{T}^*.\text{paths}$ **do**
22:             Remove $\mathcal{P}_{\mathbf{x}_i, \ell}$ from $\mathcal{T}^*.\text{paths}$
23:             Remove $\mathbf{x}_i$ from the query set of $\mathcal{T}^*.\text{left.kde}$ and $\mathcal{T}^*.\text{right.kde}$
24:             Remove $\mathcal{T}^*$ from $\mathcal{P}_{\mathbf{x}_i, \ell}$
25:          Let $\mathbf{x}_j$ be the previous sampled neighbour of $\mathbf{x}_i$ (i.e., leaf in $\mathcal{P}_{\mathbf{x}_i, \ell}$)
26:          $w_G(\mathbf{x}_i, \mathbf{x}_j) \mathrel{-}= k(\mathbf{x}_i, \mathbf{x}_j) / \widehat{w}(i,j)$ where $\widehat{w}(i,j)$ is previous used re-scale factor
27:          $E \leftarrow E \setminus (\mathbf{x}_i, \mathbf{x}_j)$ if $w_G(\mathbf{x}_i, \mathbf{x}_j) = 0$
28:          $(\mathbf{x}_i, \mathbf{x}_j^*) \leftarrow$ RESAMPLE($\mathcal{T}', \mathcal{P}_{\mathbf{x}_i, \ell}, \varepsilon$)        ▷ Resample path (Algorithm 10)
29:          $E \leftarrow E \cup \{(\mathbf{x}_i, \mathbf{x}_j^*)\}$ if $(\mathbf{x}_i, \mathbf{x}_j^*) \notin E$
30:          $\widehat{w}^*(i,j) \leftarrow L \cdot k(\mathbf{x}_i, \mathbf{x}_j^*) / \min\{\mathcal{T}.\text{kde}.\hat{\mu}_{\mathbf{x}_i}, \mathcal{T}.\text{kde}.\hat{\mu}_{\mathbf{x}_j^*}\}$
31:          **if** $\min\{\mathcal{T}.\text{kde}.\hat{\mu}_{\mathbf{x}_i}, \mathcal{T}.\text{kde}.\hat{\mu}_{\mathbf{x}_j^*}\} = \mathcal{T}.\text{kde}.\hat{\mu}_{\mathbf{x}_j^*}$ **then**
32:             $\mathcal{B}_{\mathbf{x}_j^*} \leftarrow \mathcal{B}_{\mathbf{x}_j^*} \cup \mathbf{x}_i$        ▷ Update neighbours with higher degrees
33:          $w_G(\mathbf{x}_i, \mathbf{x}_j) \mathrel{+}= k(\mathbf{x}_i, \mathbf{x}_j) / \widehat{w}^*(i,j)$

---

### C.2.1 RUNTIME ANALYSIS

Now we analyse the performance of the algorithms used in the dynamic update step. We first prove Lemmas 4.2 and 4.4

*Proof of Lemma 4.2.* The running time of the two procedures is dominated by the recursive calls to ADDQUERYPOINT($\mathbf{x}$). By Theorem 3.1, the running time of adding a query point is $\varepsilon^{-2} \cdot n^{o(1)} \cdot \text{cost}(k)$. Since the depth of the tree $\mathcal{T}$ is at most $\lceil \log n \rceil$, there are at most $\lceil \log n \rceil$ recursive calls to SAMPLE and RESAMPLE. Hence, the total running time of SAMPLE and RESAMPLE is $\varepsilon^{-2} \cdot n^{o(1)} \cdot \text{cost}(k)$. $\qquad\square$

*Proof of Lemma 4.4.* By the tree construction, we have that $\mathbb{P}[\mathbf{q} \in \mathcal{T}'.\text{kde}.Q] = \mathbb{P}[\exists \ell' \text{ such that } \mathcal{P}_{\mathbf{q}, \ell'} \in \mathcal{T}'.\text{paths}]$, and

$$\mathbb{P}[\exists \ell' \text{ such that } \mathcal{P}_{\mathbf{q}, \ell'} \in \mathcal{T}'.\text{paths}] \leqslant L \cdot \mathbb{P}[\mathcal{P}_{\mathbf{q}, \ell} \in \mathcal{T}'.\text{paths}] \leqslant L \cdot \frac{2k(\mathbf{q}, \mathcal{T}'.\text{data})}{k(\mathbf{q}, \mathcal{T}.\text{data})}$$

$$= L \cdot \frac{2k(\mathbf{q}, \mathcal{T}'.\text{kde}.X)}{k(\mathbf{q}, \mathcal{T}.\text{kde}.X)} = \widetilde{O}\left(\frac{\mu_j}{\mu_i}\right),$$

---

**Algorithm 10** Tree Update Procedures for Constructing an Approximate Similarity Graph

---

1: **procedure** ADDDATAPOINTTREE($\mathcal{T}, \mathbf{z}$)
2:     **Input**: KDE tree/node $\mathcal{T}$, new data point $\mathbf{z}$
3:     **if** ISLEAF($\mathcal{T}$) **then**
4:         $\mathbf{x} \leftarrow \mathcal{T}.\mathsf{data}$
5:         $\mathcal{A} \leftarrow \mathcal{T}.\mathsf{parent.paths}$                  ▷ Store paths that need to be resampled
6:         $\mathcal{T}_{\text{new}} \leftarrow$ NODE($\{\mathbf{x}, \mathbf{z}\}$)
7:         $\mathcal{T}_{\text{new}}.\mathsf{left} \leftarrow$ LEAF($\mathbf{x}$), $\mathcal{T}_{\text{new}}.\mathsf{right} \leftarrow$ LEAF($\mathbf{z}$)
8:         $\mathcal{T}_{\text{new}}.\mathsf{kde} \leftarrow$ DYNAMICKDE.INITIALISE($\{\mathbf{x}, \mathbf{z}\}, \emptyset, \varepsilon$)
                                                     ▷ Initialise new KDE data structure
9:         Replace the leaf $\mathcal{T}$ with node $\mathcal{T}_{\text{new}}$
10:        **return** $\mathcal{A}$
11:     **else**
12:         $\mathcal{T}.\mathsf{kde}.$ADDDATAPOINT($\mathbf{z}$) (Algorithm 1)
13:         Let $\widetilde{A}$ be the set of points $\mathbf{x}_i \in \mathcal{T}.\mathsf{kde}.Q$ such that $\mathcal{T}.\mathsf{kde}.\hat{\mu}_{\mathbf{x}_i}$ changes after adding $\mathbf{z}$.
14:         $\mathcal{A} \leftarrow \{\mathcal{P}_{\mathbf{x}_i, \ell} \in \mathcal{T}.\mathsf{parent.paths} \mid \mathbf{x}_i \in \widetilde{A}\}$
15:         **if** $\mathcal{T}.\mathsf{left.size} \leqslant \mathcal{T}.\mathsf{right.size}$ **then**
16:             **return** $\mathcal{A} \cup$ ADDDATAPOINTTREE($\mathcal{T}.\mathsf{left}, \mathbf{z}$)
17:         **else**
18:             **return** $\mathcal{A} \cup$ ADDDATAPOINTTREE($\mathcal{T}.\mathsf{right}, \mathbf{z}$)
19: **procedure** RESAMPLE($\mathcal{T}, \mathcal{P}_{\mathbf{x}_i, \ell}$)
20:     **Input**: KDE tree/node $\mathcal{T}$, and sampling path $\mathcal{P}_{\mathbf{x}_i, \ell}$
21:     $\mathcal{P}_{\mathbf{x}_i, \ell} \leftarrow \mathcal{P}_{\mathbf{x}_i, \ell} \bigcup \mathcal{T}$
22:     $\mathcal{T}.\mathsf{paths} \leftarrow \mathcal{T}.\mathsf{paths} \bigcup \{\mathcal{P}_{\mathbf{x}_i, \ell}\}$              ▷ Update and store sample paths
23:     **if** ISLEAF($\mathcal{T}$) **then**
24:         **return** $\mathbf{x}_i \times \mathcal{T}.\mathsf{data}$
25:     **else**
26:         $\mathcal{T}_L = \mathcal{T}.\mathsf{left}, X_L = \mathcal{T}.\mathsf{left.data}$
27:         $\mathcal{T}_R = \mathcal{T}.\mathsf{right}, X_R = \mathcal{T}.\mathsf{right.data}$
28:         $\mathcal{T}_L.\mathsf{kde}.$ADDQUERYPOINT($\mathbf{x}_i$) if $\mathbf{x}_i \notin \mathcal{T}_L.\mathsf{kde}.Q$              ▷ (Algorithm 1)
29:         $\mathcal{T}_R.\mathsf{kde}.$ADDQUERYPOINT($\mathbf{x}_i$) if $\mathbf{x}_i \notin \mathcal{T}_R.\mathsf{kde}.Q$
30:         $r \sim \text{Unif}[0, 1]$
31:         **if** $r \leqslant \mathcal{T}_L.\hat{\mu}_{\mathbf{x}_i}/(\mathcal{T}_L.\hat{\mu}_{\mathbf{x}_i} + \mathcal{T}_R.\hat{\mu}_{\mathbf{x}_i})$ **then**
32:             **RETURN** RESAMPLE($\mathcal{T}_L, \mathcal{P}_{\mathbf{x}_i, \ell}$)
33:         **else**
34:             **RETURN** RESAMPLE($\mathcal{T}_R, \mathcal{P}_{\mathbf{x}_i, \ell}$)

---

where the first inequality holds by the union bound, the second inequality follows by Lemma C.1, and the last line holds by the definition of $Q_{\mu_i \to \mu_j}(\mathcal{T}')$ and $L = \widetilde{O}(1)$.         $\square$

Next, we state Lemma 4.5 more precisely and provide its proof.

**Lemma C.2.** *Let $\mathbf{z}$ be the data point that is added to $\mathcal{T}$ through the* ADDDATAPOINTTREE($\mathcal{T}, \mathbf{z}$) *procedure in Algorithm 10, and $\mathcal{T}'$ be any internal node that lies on the path from the new leaf* LEAF($\mathbf{z}$) *to the root of $\mathcal{T}$. Then it holds for any $i \in [\lceil \log(2 \cdot \mathcal{T}.\mathit{kde}.n') \rceil]$, $a \in \mathcal{T}'.\mathit{kde}.K_1$, $j \in [J_{\mu_i}]$ and $\ell$ that*

$$\mathbb{E}_{H_{\mu_i, a, j, \ell}} [|\{\mathbf{q} \in \mathcal{T}'.\mathit{kde}.Q_{\mu_i} \mid \mathcal{T}'.\mathit{kde}.H_{\mu_i, a, j, \ell}(\mathbf{z}) = \mathcal{T}'.\mathit{kde}.H_{\mu_i, a, j, \ell}(\mathbf{q})\}|] = \widetilde{O}\left(\mu_i \cdot 2^{j+1}\right).$$

*Proof.* We first remark that, except for the dynamic KDE structure stored at the root $\mathcal{T}.\mathsf{kde}$, it does not necessarily hold that $\mathcal{T}'.\mathsf{kde}.Q = \mathcal{T}'.\mathsf{kde}.X$; this is because that the query points stored at internal nodes are the ones whose sample paths passed through this node, and the data points are the leaves of the subtree $\mathcal{T}'$. Hence, to analyse the expected number of colliding points in the bucket $\mathcal{T}'.\mathsf{kde}.B_{H_{\mu_i, a, j, \ell}}(\mathbf{z})$, we need to separately analyse the contributions from $\mathbf{q} \in Q_{\mu_{i'} \to \mu_i}(\mathcal{T}')$ for $i' \geqslant i$. To achieve this, we apply Lemma B.1 and have for $i' \geqslant i$ that

$$\mathbb{E}_{H_{\mu_i, a, j, \ell}} \left[ \left| \left\{ \mathbf{q} \in \mathcal{T}.\mathsf{kde}.Q_{\mu_{i'}} \mid \mathcal{T}'.\mathsf{kde}.H_{\mu_i, a, j, \ell}(\mathbf{z}) = \mathcal{T}'.\mathsf{kde}.H_{\mu_i, a, j, \ell}(\mathbf{q}) \right\} \right| \right] = O(2^{j+1} \cdot \mu_{i'}).$$
$$\text{(C.2)}$$

Therefore, it holds that

$$\mathbb{E}_{H_{\mu_i,a,j,\ell}}\left[|\{\mathbf{q} \in \mathcal{T}'.\mathsf{kde}.Q_{\mu_i} \mid \mathcal{T}'.\mathsf{kde}.H_{\mu_i,a,j,\ell}(\mathbf{z}) = \mathcal{T}'.\mathsf{kde}.H_{\mu_i,a,j,\ell}(\mathbf{q})\}|\right]$$

$$= \mathbb{E}_{H_{\mu_i,a,j,\ell}}$$

$$\left[\sum_{i' \geqslant i}|\{\mathbf{q} \in \mathcal{T}'.\mathsf{kde}.Q_{\mu_i} \mid \mathcal{T}'.\mathsf{kde}.H_{\mu_i,a,j,\ell}(\mathbf{z}) = \mathcal{T}'.\mathsf{kde}.H_{\mu_i,a,j,\ell}(\mathbf{q}) \text{ and } \mathbf{q} \in Q_{\mu_{i'} \to \mu_i}(\mathcal{T}')\}|\right]$$

$$= \sum_{i' \geqslant i} \mathbb{E}_{H_{\mu_i,a,j,\ell}}$$

$$\left[|\{\mathbf{q} \in \mathcal{T}'.\mathsf{kde}.Q_{\mu_i} \mid \mathcal{T}'.\mathsf{kde}.H_{\mu_i,a,j,\ell}(\mathbf{z}) = \mathcal{T}'.\mathsf{kde}.H_{\mu_i,a,j,\ell}(\mathbf{q}) \text{ and } \mathbf{q} \in Q_{\mu_{i'} \to \mu_i}(\mathcal{T}')\}|\right]$$

$$\leqslant \sum_{i' \geqslant i} \widetilde{O}\left(\frac{\mu_i}{\mu_{i'}}\right) \cdot \mathbb{E}_{H_{\mu_i,a,j,\ell}}\left[|\{\mathbf{q} \in \mathcal{T}.\mathsf{kde}.Q_{\mu_{i'}} \mid \mathcal{T}'.\mathsf{kde}.H_{\mu_i,a,j,\ell}(\mathbf{z}) = \mathcal{T}'.\mathsf{kde}.H_{\mu_i,a,j,\ell}(\mathbf{q})\}|\right]$$

$$\tag{C.3}$$

$$= \sum_{i' \geqslant i} \widetilde{O}\left(\frac{\mu_i}{\mu_{i'}} \cdot 2^{j+1}\mu_{i'}\right) \tag{C.4}$$

$$= \widetilde{O}\left(\mu_i \cdot 2^{j+1}\right), \tag{C.5}$$

where (C.3) follows by Lemma 4.4, and (C.4) holds by (C.2). $\qquad\square$

**Lemma C.3.** *The expected total running time for $\mathcal{T}'.\mathsf{kde}.\text{ADDDATAPOINT}(\mathbf{z})$ (Line 12 of Algorithm 10) over all internal nodes $\mathcal{T}'$ along the path from the new leaf $\text{LEAF}(\mathbf{z})$ to the root of $\mathcal{T}$ is $n^{o(1)} \cdot \text{cost}(k)$. Moreover, the expected number of paths $\mathcal{A}$ (Line 3, Algorithm 9) that need to be resampled satisfies that $\mathbb{E}[|\mathcal{A}|] = \widetilde{O}(1)$.*

*Proof.* We first study the update time and the total number of paths that need to be updated at a single internal node $\mathcal{T}'$. Notice that, when $\mathcal{T}'.\mathsf{kde}.\text{ADDDATAPOINT}(\mathbf{z})$ (Line 12 of Algorithm 10) is called, the procedure ADDPOINTANDUPDATEQUERIES in Algorithm 6 is called in the dynamic $\mathsf{KDE}$ data structure $\mathcal{T}'.\mathsf{kde}$. Hence, we analyse the expected running time of ADDPOINTANDUPDATEQUERIES in Algorithm 6.

First, we have that executing Lines 7–11 of in Algorithm 6 takes $K_{2,j} \cdot |\mathcal{T}'.\mathsf{kde}.B^*_{H_{\mu_i,a,j,\ell}}(\mathbf{z})| \cdot n'^{o(1)}$ time, where $n' = |\mathcal{T}'.\mathsf{data}|$ is the number of data points stored at $\mathcal{T}'$, and these five lines are executed with probability at most $1/(2^{j+1}\mu_i)$. Since we only consider the collisions with points in $\mathcal{T}'.\mathsf{kde}.Q_{\mu_i}$, it holds by Lemma C.2 that

$$\mathbb{E}_{H_{\mu_i,a,j,\ell}}\left[|\mathcal{T}'.\mathsf{kde}.B^*_{H_{\mu_i,a,j,\ell}}(\mathbf{z})|\right] = \widetilde{O}\left(2^{j+1}\mu_i\right).$$

Hence, by our choice of $K_{2,j} = O(\log(n') \cdot \text{cost}(k))$, the expected total running time over all $\mu_i$, $a$, and $j$ of Lines 7–11 of Algorithm 6 is $\varepsilon^{-2} \cdot \text{cost}(k) \cdot n'^{o(1)}$. The same analysis can also be applied for Lines 12–15 of Algorithm 6. Moreover, the expected number of recovered points in $S$ (Line 16 of Algorithm 6) is $\widetilde{O}(1)$, as the expected number of collisions we consider is

$$\mathbb{E}_{H_{\mu_i,a,j,\ell}}\left[|\mathcal{T}'.\mathsf{kde}.B^*_{H_{\mu_i,a,j,\ell}}(\mathbf{z})|\right] = \widetilde{O}(2^{j+1}\mu_i),$$

and these points are only considered with probability at most $1/(2^{j+1}\mu_i)$.

Next, we analyse the running time of Lines 17–29 of Algorithm 6. For every $\mathbf{q} \in S$, the total running time for Lines 17–29 is $\widetilde{O}(\varepsilon^{-2} \cdot K_{2,j} + \varepsilon^{-2} \cdot \text{cost}(k)) = \widetilde{O}(\varepsilon^{-2} \cdot \text{cost}(k))$, due to Lines 28 and 29.

Hence, the expected total running time for running $\mathcal{T}'.\mathsf{kde}.\text{ADDDATAPOINT}(\mathbf{z})$ at a single $\mathcal{T}'$ is $\widetilde{O}\left(\varepsilon^{-2} \cdot n'^{o(1)} \cdot \text{cost}(k)\right)$. As there are at most $\lceil \log n \rceil$ nodes $\mathcal{T}'$ that are updated when $\mathbf{z}$ is added and $n' \leqslant n$, the running time guarantee of the lemma follows.

It remains to prove that $\mathbb{E}[|\mathcal{A}|] = \widetilde{O}(1)$. Notice that, the number of points $\mathbf{q} \in \mathcal{T}'.\mathsf{kde}.Q$ whose KDE estimate changes at $\mathcal{T}'$ is the number of recovered points in $S$ (Line 16 of Algorithm 6). From

the ADDPOINTANDUPDATEQUERIES procedure (Algorithm 6), it holds for every $\mu_i$ and $a$ that $\mathbb{E}[|S|] = \widetilde{O}(1)$; as such for every $\mathcal{T}'$ the expected number of KDE estimates that change – and therefore the number of paths that need to be resampled – is $\widetilde{O}(1)$. As there are at most $\lceil \log n \rceil$ trees $\mathcal{T}'$ that are updated when $\mathbf{z}$ is added, it holds that $\mathbb{E}[|\mathcal{A}|] = \widetilde{O}(1)$. $\qquad\square$

Next we bound the size of the set $\mathcal{B}_{\mathbf{x}_i}$ that keeps track of the neighbours $\mathbf{x}_j$ of $\mathbf{x}_i$ in the approximate similarity graph $G$ that have higher degree.

**Lemma C.4.** *It holds with high probability for all $\mathbf{x}_i \in X$ that $|\mathcal{B}_{\mathbf{x}_i}| \leqslant 14 \cdot L$.*

*Proof.* We first notice that

$$\mathcal{B}_{\mathbf{x}_i} = \left\{ \mathbf{x}_j \in X \mid \mathcal{T}.\mathsf{kde}.\hat{\mu}_{\mathbf{x}_j} > \mathcal{T}.\mathsf{kde}.\hat{\mu}_{\mathbf{x}_i} \text{ and } i \in Y_{\mathbf{x}_j} \right\},$$

where $Y_{\mathbf{x}_j} \triangleq \{y_{j,1}, \ldots y_{j,L}\}$ are the indices corresponding to the sampled neighbours of $\mathbf{x}_j$. For every pair of indices $i, j$, and every $1 \leqslant \ell \leqslant L$, we define the random variable $Z_{i,j,\ell}$ to be 1 if $j$ is the neighbour sampled from $i$ at iteration $\ell$, and 0 otherwise, i.e.,

$$Z_{i,j,\ell} \triangleq \left\{ \begin{array}{ll} 1 & \text{if } y_{i,\ell} = j \\ 0 & \text{otherwise.} \end{array} \right.$$

We fix an arbitrary $\mathbf{x}_i$, and notice that

$$|\mathcal{B}_{\mathbf{x}_i}| = \sum_{\ell=1}^{L} \sum_{\substack{j=1 \\ \hat{\mu}_{\mathbf{x}_j} > \hat{\mu}_{\mathbf{x}_i}}}^{n} Z_{j,i,\ell}, \tag{C.6}$$

where for ease of notation we set $\hat{\mu}_{\mathbf{x}_i} \triangleq \mathcal{T}.\mathsf{kde}.\hat{\mu}_{\mathbf{x}_i}$ and $\hat{\mu}_{\mathbf{x}_j} \triangleq \mathcal{T}.\mathsf{kde}.\hat{\mu}_{\mathbf{x}_j}$ to be the KDE estimates at the root $\mathcal{T}$. We have that

$$
\begin{aligned}
\mathbb{E}\left[ \sum_{\ell=1}^{L} \sum_{\substack{j=1 \\ \hat{\mu}_{\mathbf{x}_j} > \hat{\mu}_{\mathbf{x}_i}}}^{n} Z_{j,i,\ell} \right] &= \sum_{\ell=1}^{L} \sum_{\substack{j=1 \\ \hat{\mu}_{\mathbf{x}_j} > \hat{\mu}_{\mathbf{x}_i}}}^{n} \mathbb{E}[Z_{j,i,\ell}] \\
&= \sum_{\ell=1}^{L} \sum_{\substack{j=1 \\ \hat{\mu}_{\mathbf{x}_j} > \hat{\mu}_{\mathbf{x}_i}}}^{n} \mathbb{P}[y_{j,\ell} = i] \\
&\leqslant \sum_{\ell=1}^{L} \sum_{\substack{j=1 \\ \hat{\mu}_{\mathbf{x}_j} > \hat{\mu}_{\mathbf{x}_i}}}^{n} \frac{2k(\mathbf{x}_i, \mathbf{x}_j)}{\deg_{\mathsf{K}}(\mathbf{x}_j)} \\
&< \sum_{\ell=1}^{L} \sum_{\substack{j=1 \\ \hat{\mu}_{\mathbf{x}_j} > \hat{\mu}_{\mathbf{x}_i}}}^{n} \frac{4k(\mathbf{x}_i, \mathbf{x}_j)}{\deg_{\mathsf{K}}(\mathbf{x}_i)}. \\
&\leqslant 4 \cdot L. \tag{C.7}
\end{aligned}
$$

Here, the second last inequality holds by the fact that

$$\deg_{\mathsf{K}}(\mathbf{x}_j) \geqslant \frac{\hat{\mu}_{\mathbf{x}_j}}{1+\varepsilon} > \frac{\hat{\mu}_{\mathbf{x}_i}}{1+\varepsilon} \geqslant \frac{(1-\varepsilon)\deg_{\mathsf{K}}(\mathbf{x}_i)}{1+\varepsilon} \geqslant \frac{\deg_{\mathsf{K}}(\mathbf{x}_i)}{2},$$

where the last inequality follows by our choice of $\varepsilon \leqslant 1/6$. Employing the same analysis, we have that

$$R = \sum_{\ell=1}^{L} \sum_{\substack{j=1 \\ \hat{\mu}_{\mathbf{x}_j} > \hat{\mu}_{\mathbf{x}_i}}}^{n} \mathbb{E}\left[ Z_{j,i,\ell}^2 \right] = \sum_{\ell=1}^{L} \sum_{\substack{j=1 \\ \hat{\mu}_{\mathbf{x}_j} > \hat{\mu}_{\mathbf{x}_i}}}^{n} \mathbb{P}[y_{j,\ell} = i] \leqslant 4 \cdot L.$$

We apply the Bernstein's inequality, and have that

$$\mathbb{P}\left[\left|\sum_{\ell=1}^{L}\sum_{\substack{j=1\\\hat{\mu}_{\mathbf{x}_j}>\hat{\mu}_{\mathbf{x}_i}}}^{n} Z_{j,i,\ell} - \mathbb{E}\left[\sum_{\ell=1}^{L}\sum_{\substack{j=1\\\hat{\mu}_{\mathbf{x}_j}>\hat{\mu}_{\mathbf{x}_i}}}^{n} Z_{j,i,\ell}\right]\right| \geqslant 10L\right] \leqslant 2\exp\left(-\frac{100L^2/2}{4L + 10\cdot L/3}\right)$$

$$= 2\exp\left(-\frac{75L}{22}\right)$$

$$= o(1/n).$$

Hence, by the union bound, it holds with high probability for all $\mathbf{x}_i \in X$ that

$$\left|\sum_{\ell=1}^{L}\sum_{\substack{j=1\\\hat{\mu}_{\mathbf{x}_j}>\hat{\mu}_{\mathbf{x}_i}}}^{n} Z_{j,i,\ell} - \mathbb{E}\left[\sum_{\ell=1}^{L}\sum_{\substack{j=1\\\hat{\mu}_{\mathbf{x}_j}>\hat{\mu}_{\mathbf{x}_i}}}^{n} Z_{j,i,\ell}\right]\right| < 10L;$$

combining this with (C.6) and (C.7), we have with high probability that

$$||\mathcal{B}_{\mathbf{x}_i}| - 4L| < 10L,$$

which implies that $|\mathcal{B}_{\mathbf{x}_i}| < 14L$. $\qquad\square$

We are now ready to prove the running time guarantee of the update step.

**Lemma C.5.** *The expected running time of* UPDATEGRAPH$(G, \mathcal{T}, \mathbf{z})$ *is* $n^{o(1)} \cdot \text{cost}(k)$.

*Proof.* We analyse the running time of UPDATEGRAPH$(G, \mathcal{T}, \mathbf{z})$ step by step.

- The ADDDATAPOINTTREE procedure is dominated by the call to the ADDDATAPOINT procedure on Line 12 of Algorithm 10, which takes $\varepsilon^{-2} \cdot n^{o(1)} \cdot \text{cost}(k)$ time by Lemma C.3.

- Next, we analyse the running time of sampling $L$ new neighbours of the new data point $\mathbf{z}$ (Lines 5–11). The algorithm samples a neighbour $\mathbf{x}_j$ using the SAMPLE procedure, which takes $\varepsilon^{-2} \cdot n^{o(1)} \cdot \text{cost}(k)$ time (Lemma 4.2). To add the edge $(\mathbf{z}, \mathbf{x}_j)$, the algorithm computes the KDE estimate $\mathcal{T}.\text{kde}.\hat{\mu}_{\mathbf{z}}$, which takes $\varepsilon^{-2} \cdot n^{o(1)} \cdot \text{cost}(k)$ time, and the weight value $k(\mathbf{z}, \mathbf{x}_j)$ which takes $O(d) = \widetilde{O}(1)$ time. Since $L = \widetilde{O}(1)$, the total running time of Lines 5–11 is $\varepsilon^{-2} \cdot n^{o(1)} \cdot \text{cost}(k)$.

- For Lines 13–18, first note that the expected number of paths that need to be resampled is $\mathbb{E}[|\mathcal{A}|] = \widetilde{O}(1)$ (Lemma C.3), and the expected number of points $\mathbf{x}_i$ such that $\mathcal{T}.\text{kde}.\hat{\mu}_{\mathbf{x}_i}$ has changed is $\widetilde{O}(1)$. Since by Lemma C.4 it holds with high probability that $|\mathcal{B}_{\mathbf{x}_i}| \leqslant 4 \cdot L = \widetilde{O}(1)$, the total expected running time of Lines 13–18 is $\widetilde{O}(1)$.

- Finally, we analyse the running time of Lines 19–33. The running time of removing all the stored data about the path $\mathcal{P}_{\mathbf{x}_i,\ell}$ that needs to be resampled (Lines 21–27) is dominated by the time needed for removing all the stored information about $\mathbf{x}_i$ in $\mathcal{T}^*.\text{left.kde}$ and $\mathcal{T}^*.\text{right.kde}$ for every $\mathcal{T}^*$ (Line 21). Doing this for all $\mathcal{T}^*$ takes $\varepsilon^{-2} \cdot n^{o(1)} \cdot \text{cost}(k)$ time, since there are $O(\log n)$ such trees $\mathcal{T}^*$ and in the data structures $\mathcal{T}^*.\text{left.kde}$ and $\mathcal{T}^*.\text{right.kde}$, $\mathbf{x}_i$ is removed from all buckets $B^*_{H_{\mu_i,a,j,\ell}}(\mathbf{x}_i)$, and there are $\varepsilon^{-2} \cdot n^{o(1)} \cdot \text{cost}(k)$ such buckets. The running time of the rest of the loop (Lines 28–33) is dominated by the running time for resampling a path $\mathcal{P}_{\mathbf{x}_i,\ell}$, which is $\varepsilon^{-2} \cdot n^{o(1)} \cdot \text{cost}(k)$ (Lemma 4.2). Therefore, by the fact that $\mathbb{E}[|\mathcal{A}|] = \widetilde{O}(1)$ (Lemma C.3), the total expected running time of Lines 19–33 is $\varepsilon^{-2} \cdot n^{o(1)} \cdot \text{cost}(k)$.

Combining everything together proves the lemma. $\qquad\square$

### C.2.2 PROOF OF CORRECTNESS

**Lemma C.6.** *Let $G' = (X \cup \mathbf{z}, E', w_{G'})$ be the updated graph after running* UPDATEGRAPH$(G, \mathcal{T}, \mathbf{z})$ *for the new arriving* $\mathbf{z}$. *Then, it holds with probability at least $9/10$ that $G'$ is an approximate similarity graph on $X \cup \mathbf{z}$.*

*Proof.* We prove this statement by showing that running CONSTRUCTGRAPH$(X)$ followed by UPDATEGRAPH$(G, \mathcal{T}, \mathbf{z})$ is equivalent to running CONSTRUCTGRAPH$(X \cup \mathbf{z})$.

- First, we prove that the structure of the tree $\mathcal{T}$ is the same in both settings: when running CONSTRUCTGRAPH$(X)$, we ensure that the tree $\mathcal{T}$ is a complete binary tree. Then, when inserting a data point $\mathbf{z}$ using the ADDDATAPOINT$(\mathbf{z})$ procedure on Line 3 of Algorithm 9, $\mathbf{z}$ is inserted appropriately (by the condition on Line 15 of Algorithm 10) such that the updated tree is also a complete binary tree. Therefore, the structure of the tree $\mathcal{T}$ is identical in both settings.

- Next, on Line 12 of Algorithm 9, $\mathbf{z}$ is added to the relevant $\mathcal{T}'$.kde dynamic KDE data structures using the ADDDATAPOINT$(\mathbf{z})$ procedure of Algorithm 1. This ensures that the stored data points $\mathcal{T}'$.kde.$X$ at every internal node $\mathcal{T}'$ are identical in both settings and, by the guarantees of the dynamic KDE data structures (Theorem 3.1), the query estimates $\mathcal{T}'$.kde.$\hat{\mu}_{\mathbf{q}}$ for every internal node $\mathcal{T}'$ and any $\mathbf{q} \in \mathcal{T}'$.kde.$Q$ are the same in both settings.

- For the new data point $\mathbf{z}$, we sample $L$ new neigbours (Lines 5–11 of Algorithm 9). By the previous points, it holds that the tree $\mathcal{T}$ is identical in both settings, and therefore the sampling procedure on Lines 5–11 in Algorithm 9 for the new data point $\mathbf{z}$ is equivalent to the sampling procedure on Lines 41–48 of Algorithm 8 for the point $\mathbf{z}$ when executing INITIALISE$(X \cup \mathbf{z}, \varepsilon)$.

- Then, for any data point $\mathbf{x}_i \in X$, let $(\mathbf{x}_i, \mathbf{x}_j) \in E$ be one of its sampled neighbours edge after running CONSTRUCTGRAPH. It holds that the scaling factor for the edge weight $w_G(\mathbf{x}_i, \mathbf{x}_j)$ is

$$\widehat{w}(i, j) = \frac{L \cdot k(\mathbf{x}_i, \mathbf{x}_j)}{\min\{\mathcal{T}.\text{kde}.\hat{\mu}_{\mathbf{x}_i}, \mathcal{T}.\text{kde}.\hat{\mu}_{\mathbf{x}_j}\}}.$$

  Notice that after running UPDATEG$(\mathbf{z})$, the scaling factor $w_G(\mathbf{x}_i, \mathbf{x}_j)$ can change due to a change in $\min\{\mathcal{T}.\text{kde}.\hat{\mu}_{\mathbf{x}_i}, \mathcal{T}.\text{kde}.\hat{\mu}_{\mathbf{x}_j}\}$. Without loss of generality, let $\min\{\mathcal{T}.\text{kde}.\hat{\mu}_{\mathbf{x}_i}, \mathcal{T}.\text{kde}.\hat{\mu}_{\mathbf{x}_j}\} = \mathcal{T}.\text{kde}.\hat{\mu}_{\mathbf{x}_j}$. By Line 47 of Algorithm 8, in this case we have $\mathbf{x}_i \in \mathcal{B}_{\mathbf{x}_j}$. We further distinguish between the two cases:

  1. If $\mathcal{T}.\text{kde}.\hat{\mu}_{\mathbf{x}_i}$ changes after running UPDATEGRAPH$(\mathbf{z})$, then by the ADDDATAPOINTTREE$(\mathcal{T}, \mathbf{z})$ procedure all the paths $\mathcal{P}_{\mathbf{x}_i, \ell}$ for $1 \leqslant \ell \leqslant L$ will be resampled and updated on Lines 21–33.
  2. On the other hand, if $\mathcal{T}.\text{kde}.\hat{\mu}_{\mathbf{x}_j}$ changes and $\mathcal{T}.\text{kde}.\hat{\mu}_{\mathbf{x}_i}$ does not, then the paths $\mathcal{P}_{\mathbf{x}_j, \ell'}$ ending at the leaf corresponding to $\mathbf{x}_i$ are not necessarily resampled. In this case, the scaling factor is updated on Lines 13–18, and therefore $w_G(\mathbf{x}_i, \mathbf{x}_j)$ is appropriately rescaled.

- Let $\mathcal{P}^*_{\mathbf{x}_i, \ell} \in \mathcal{T}.\text{paths}$ be any sampling path that is not resampled, i.e., $\mathcal{P}^*_{\mathbf{x}_i, \ell} \notin \mathcal{A}$. This implies that the KDE estimate of $\mathcal{T}'.\text{kde}.\hat{\mu}_{\mathbf{x}_i}$ does not change at any internal $\mathcal{T}'$ where $\mathcal{P}_{\mathbf{x}_i, \ell}$ is stored, and therefore the sampling procedure for $\mathcal{P}^*_{\mathbf{x}_i, \ell}$ is identical in both settings.

- Finally, let $\mathcal{P}_{\mathbf{x}_i, \ell} \in \mathcal{A}$ be a sampling path that is resampled, and $(\mathbf{x}_i, \mathbf{x}_j)$ be the sampled edge (contribution) corresponding to $\mathcal{P}_{\mathbf{x}_i, \ell}$. Before resampling the path $\mathcal{P}_{\mathbf{x}_i, \ell}$ starting from $\mathcal{T}'$, on Lines 21–27 the algorithm removes the stored paths $\mathcal{P}_{\mathbf{x}_i, \ell}$ and query points $\mathbf{x}_i$ from every internal node $\mathcal{T}^*$ below $\mathcal{T}'$, and removes the weight contribution to $w_G(\mathbf{x}_i, \mathbf{x}_j)$ from $\mathcal{P}_{\mathbf{x}_i, \ell}$. Then, on Lines 28–33, we resample a new edge $(\mathbf{x}_i, \mathbf{x}_j^*)$, in an equivalent manner as sampling a new edge when running Lines 41–48 of Algorithm 8. Therefore, the resampling procedure for the path $\mathcal{P}_{\mathbf{x}_i, \ell}$ is identical to the sampling procedure for $\mathcal{P}_{\mathbf{x}_i, \ell}$ when running INITIALISE$(X \cup \mathbf{z}, \varepsilon)$, because the resampling procedure uses the updated KDE estimates at each internal node $\mathcal{T}'$, which are identical to the KDE estimates that would be computed in INITIALISE$(X \cup \mathbf{z}, \varepsilon)$.

Combining everything together proves the lemma. □

Finally, we are ready to prove the second statement of Theorem 4.1.

*Proof of the Second Statement of Theorem 4.1.* Lemma C.5 shows the time complexity of UPDATEGRAPH$(G, \mathcal{T}, \mathbf{z})$, and Lemma C.6 shows the correctness of our updated procedures. Combining these two facts together proves the second statement of Theorem 4.1. □

# D    ADDITIONAL EXPERIMENTAL RESULTS

In this section, we provide some more details about our experimental setup and give some additional experimental results. Table 4 provides additional information about all of the datasets used in our experiments.

## D.1    DYNAMIC KDE EXPERIMENTS

Tables 5 and 6 show the experimental evaluation of the dynamic KDE algorithms on several additional datasets. The results demonstrate that our algorithm scales better to larger datasets than the baseline algorithms.

Figures 4 and 5 show the relative errors and running times for all iterations, datasets, and algorithms for the dynamic KDE experiments.

## D.2    PLOTS FOR DYNAMIC SIMILARITY GRAPH EXPERIMENTS

Table 7 shows the results of the experiments for the dynamic similarity graph, evaluated with the Adjusted Rand Index (ARI) (Rand, 1971).

Table 4: Datasets used for experimental evaluation. $n$ is the number of data points, $d$ is the dimension, and $\sigma$ is the parameter we use in the Gaussian kernel.

| Dataset | n | d | $\sigma$ | License | Reference | Description |
|---|---|---|---|---|---|---|
| blobs | 20,000 | 10 | 0.01 | BSD | (Pedregosa et al., 2011) | Synthetic clusters from a mixture of Gaussian distributions. |
| cifar10 | 50,000 | 2,048 | 0.0001 | - | (He et al., 2016; Krizhevsky, 2009) | ResNet-50 embeddings of images. |
| mnist | 70,000 | 728 | 0.000001 | CC BY-SA 3.0 | (Lecun et al., 1998) | Images of handwritten digits. |
| shuttle | 58,000 | 9 | 0.01 | CC BY 4.0 | (NASA, 2002) | Numerical data from NASA space shuttle sensors. |
| aloi | 108,000 | 128 | 0.01 | - | (Geusebroek et al., 2005) | Images of objects under a variety of lighting conditions. |
| msd | 515,345 | 90 | 0.000001 | CC BY 4.0 | (Bertin-Mahieux et al., 2011) | Numerical and categorical features of songs. |
| covtype | 581,012 | 54 | 0.000005 | CC BY 4.0 | (Blackard & Dean, 1999) | Cartographic features used to predict forest cover type. |
| glove | 1,193,514 | 100 | 0.1 | PDDL 1.0 | (Pennington et al., 2014) | Word embedding vectors. |
| census | 2,458,285 | 68 | 0.01 | CC BY 4.0 | (Meek et al., 1990) | Categorical and numerical data from the 1990 US census. |

Table 5: Experimental results for dynamic KDE. For each dataset, the shaded results correspond to the algorithm with the lowest total running time.

| | CKNS | | DYNAMICRS | | OUR ALGORITHM | |
|---|---|---|---|---|---|---|
| dataset | Time (s) | Err | Time (s) | Err | Time (s) | Err |
| shuttle | $32.9_{\pm 2.1}$ | $0.146_{\pm 0.002}$ | $0.8_{\pm 0.0}$ | $0.078_{\pm 0.005}$ | $10.9_{\pm 0.3}$ | $0.159_{\pm 0.024}$ |
| aloi | $619.0_{\pm 10.7}$ | $0.050_{\pm 0.006}$ | $19.7_{\pm 0.3}$ | $0.010_{\pm 0.003}$ | $46.9_{\pm 0.7}$ | $0.060_{\pm 0.021}$ |
| msd | $14,360.0_{\pm 0.0}$ | $0.385_{\pm 0.000}$ | $1,887.8_{\pm 0.0}$ | $5.430_{\pm 0.000}$ | $306.4_{\pm 0.0}$ | $0.388_{\pm 0.000}$ |
| covtype | $5,650.3_{\pm 109.0}$ | $0.159_{\pm 0.002}$ | $309.2_{\pm 2.4}$ | $0.018_{\pm 0.003}$ | $151.7_{\pm 4.5}$ | $0.196_{\pm 0.017}$ |
| glove | $2,640.8_{\pm 1677.7}$ | $0.221_{\pm 0.229}$ | $1,038.6_{\pm 26.5}$ | $0.004_{\pm 0.005}$ | $445.6_{\pm 214.6}$ | $0.296_{\pm 0.469}$ |
| census | $10,471.5_{\pm 160.6}$ | $0.080_{\pm 0.000}$ | $3,424.8_{\pm 5.2}$ | $0.005_{\pm 0.001}$ | $836.5_{\pm 44.6}$ | $0.102_{\pm 0.021}$ |

Table 6: Running times for dynamic KDE with the exact algorithm.

| Dataset | Running Time |
|---|---|
| shuttle | $4.1_{\pm 0.1}$ |
| aloi | $164.5_{\pm 13.6}$ |
| msd | $2,715.6_{\pm 0.0}$ |
| covtype | $2,349.8_{\pm 101.2}$ |
| glove | $5,251.7_{\pm 0.0}$ |
| census | $16,202.6_{\pm 154.6}$ |

Table 7: ARI values for the dynamic similarity graph experiments.

| | FULLYCONNECTED | | kNN | | OUR ALGORITHM | |
|---|---|---|---|---|---|---|
| dataset | Time (s) | ARI | Time (s) | ARI | Time (s) | ARI |
| blobs | $72.8_{\pm 2.2}$ | $1.000_{\pm 0.000}$ | $383.6_{\pm 3.9}$ | $0.797_{\pm 0.287}$ | $21.2_{\pm 0.8}$ | $1.000_{\pm 0.000}$ |
| cifar10 | $19,158.2_{\pm 231.6}$ | $0.000_{\pm 0.000}$ | $3,503.0_{\pm 490.6}$ | $0.098_{\pm 0.001}$ | $1,403.5_{\pm 152.4}$ | $0.221_{\pm 0.013}$ |
| mnist | $1,328.3_{\pm 159.5}$ | $0.149_{\pm 0.000}$ | $5,796.3_{\pm 234.3}$ | $0.673_{\pm 0.001}$ | $1,470.3_{\pm 77.9}$ | $0.238_{\pm 0.011}$ |

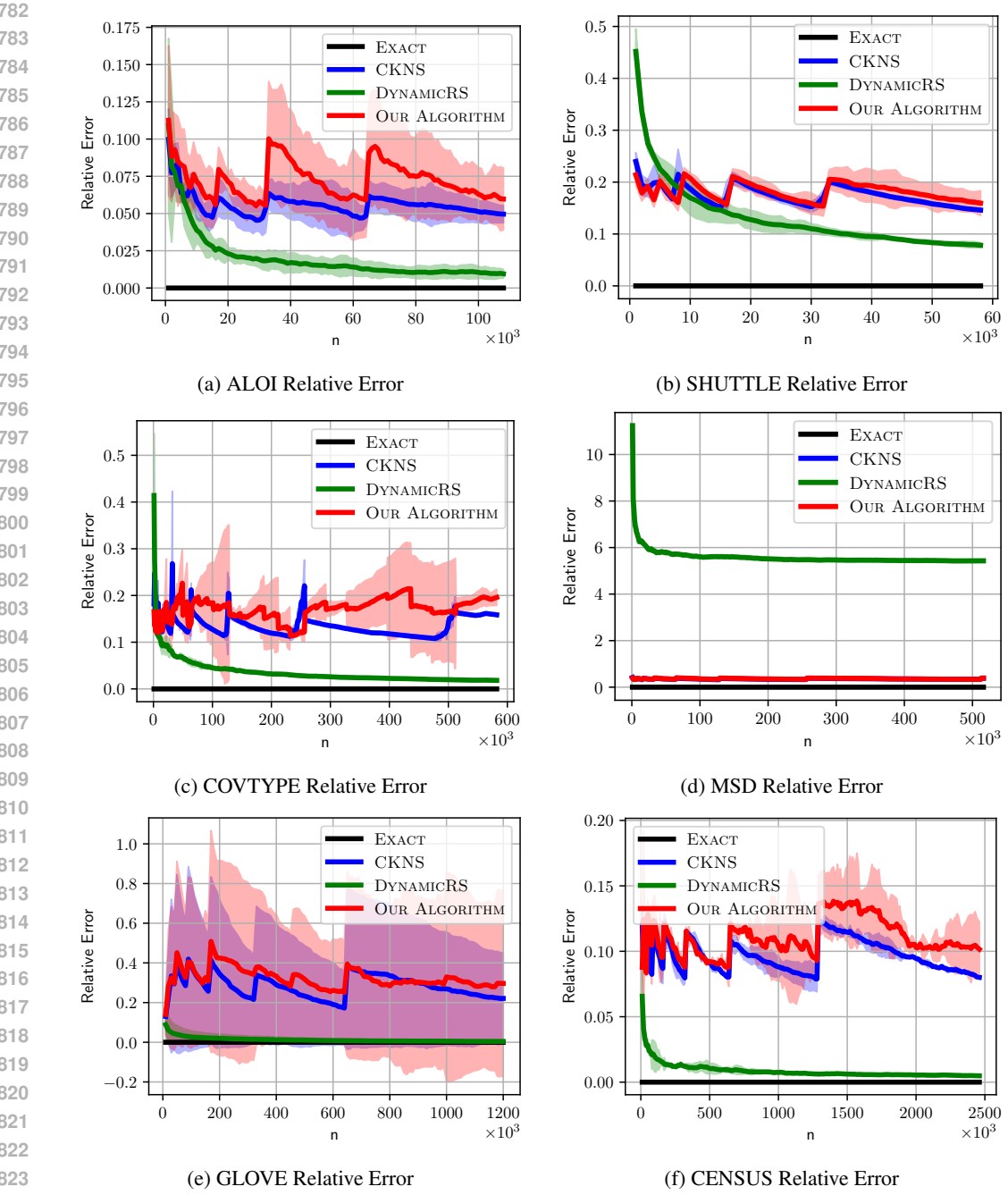

(a) ALOI Relative Error

(b) SHUTTLE Relative Error

(c) COVTYPE Relative Error

(d) MSD Relative Error

(e) GLOVE Relative Error

(f) CENSUS Relative Error

Figure 4: Relative errors for all datasets.

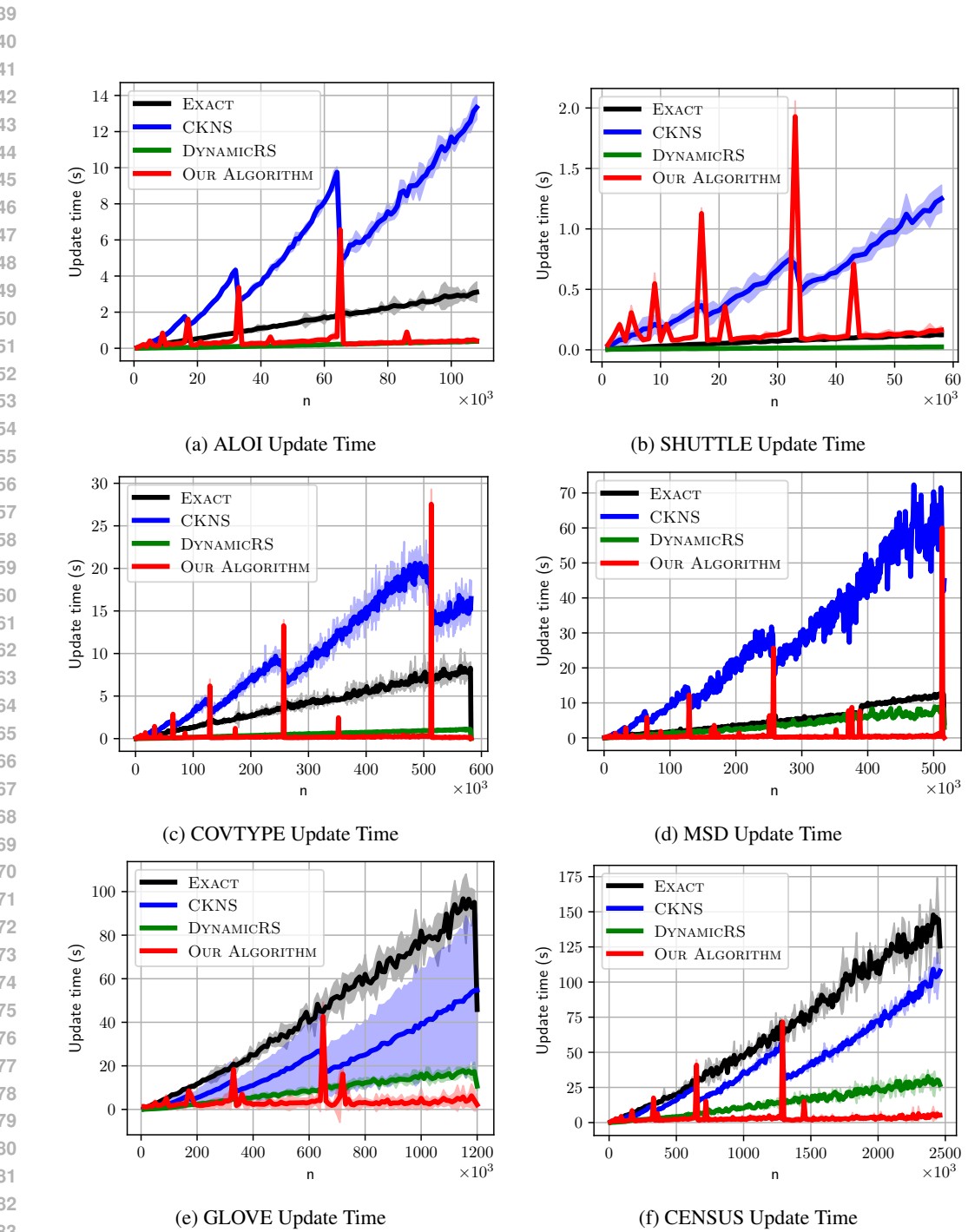

Figure 5: Running times for all data sets.

