# OpenReview forum: "Dynamic Similarity Graph Construction with Kernel Density Estimation"
_ICLR.cc/2025/Conference — Submitted to ICLR 2025_

### Official Review · Reviewer_GVfg · 2024-10-27

**Soundness:** 2
**Presentation:** 3
**Contribution:** 2
**Rating:** 3
**Confidence:** 4

**Summary:**

This paper studies the kernel density estimation problem, in which we are given a set $X$ of data points in $\mathbb{R}^d$, a kernel function $k : \mathbb{R}^d \times \mathbb{R}^d \to \mathbb{R}$, and a query point $q \in \mathbb{R}^d$, and the objective is to quickly output an estimate of $\sum_{x \in X} k(q, x)$. In particular, they consider the dynamic setting, where both points in the dataset $X$ and the queries in the query set $Q$ can change over time and the goal is to maintain $(1+\varepsilon)$-multiplicative approximations to the queries.

The paper gives a data structure for approximation of KDE queries that uses $\varepsilon^{-2} \cdot n^{1+o(1)} \cdot \text{cost}(k)$ initialization time and dynamic data point updates with time $\varepsilon^{-2} \cdot n^{o(1)} \cdot \text{cost}(k)$ per update. The paper also gives a data structure for sparse similarity graph approximation that uses $n^{1+o(1)} \cdot \text{cost}(k)$ time to initialize, as well  $n^{o(1)} \cdot \text{cost}(k)$ expected amortised update time. For both settings, $\text{cost}(k)$ is a function of the kernel function that measures how far points can be while still only contributing a certain amount to the KDE query.

**Strengths:**

+ KDE estimation is a well-motivated problem that would be of interest to multiple areas of the ML community
+ The dynamic setting is a specific direction that has recently received attention
+ The main statements of the paper are prove with mathematically rigorous claims
+ The experiments are comprehensive and shows good performance in practice

**Weaknesses:**

- Similar qualitative results for both the initialisation time and the update time were achieved by Liang et. al. (2024). It seems the techniques are reasonably similar too, looking to sample points in various distance levels around the queries. Although this paper cites the Liang et. al. paper and although it is true the latter paper does not handle dynamic updates of the query set, this paper fails to mention that the qualitative results of the Liang et. al. paper are similar to this paper, and it fails to compare the techniques between the papers.

Jiehao Liang, Zhao Song, Zhaozhuo Xu, Danyang Zhuo: Dynamic Maintenance of Kernel Density Estimation Data Structure: From Practice to Theory. CoRR abs/2208.03915 (2024)

- Though the claim is that the update time is sublinear, the cost function depends on the value of the smallest query. For many reasonable kernels such as the Gaussian kernel, these queries have very small value, in which case the resulting update time is not small. On the other hand, for queries with large value, sampling is a standard KDE technique. Given these points, it is not clear what are the technical novelties of this paper.

**Questions:**

N/A

---

> ### Author Response · Authors · 2024-11-18
>
> We thank the reviewer for their time and feedback, and we respond to points below.
>
> > **W1**: “Similar qualitative results for both the initialisation time and the update time were achieved by Liang et. al. (2024). It seems the techniques are reasonably similar too, looking to sample points in various distance levels around the queries. Although this paper cites the Liang et. al. paper and although it is true the latter paper does not handle dynamic updates of the query set, this paper fails to mention that the qualitative results of the Liang et. al. paper are similar to this paper, and it fails to compare the techniques between the papers.”
>
> **Response:**
> We appreciate the reviewer’s remark, and provide a detailed comparison between our approach and that of Liang et al. (2022) below.
>
> Both our work and that of Liang et al. (2022) are based on the static LSH-based KDE estimator of Charikar et al. (2020) (referred to as CKNS in the manuscript), who introduced the technique of sampling points at geometric weight levels $L_{j}^{\mathbf{q}}$.
>
> However, the key difference is that our algorithm supports dynamic updates of maintained query points as data points are added, while Liang et al. only consider updating the CKNS estimator. This difference, outlined briefly in Lines 226-230, is expanded here for clarity:
>
> 1. Liang et al. study the setting where a new data point $\mathbf{z}$ replaces an existing point $\mathbf{x}\_i \in X$. Under this dynamic setting, they show that  the CKNS _estimator_ can be maintained.  They achieve this by iterating through the maintained hash buckets $B_{H_{\mu\_i, a, j, \ell}} (\mathbf{x}\_i)$ and replacing every stored $\mathbf{x}\_i$ with $\mathbf{z}$.
>
> 2. On the other hand, we study the incremental setting where a new point $\mathbf{z}$ is added to $X$. To update the estimator, we take a similar approach, and iterate through the maintained hash buckets $B_{H_{\mu\_i, a, j, \ell}} (\mathbf{z})$ and add $\mathbf{z}$ accordingly (Lines 1-10 of Algorithm 6 in the appendix). Updating the estimator like this allows for fast estimation of KDE estimates on the updated data points. _However_, it is not clear how to efficiently update the estimates of a set of maintained query points, i.e., where the update time is independent of the number of added query points, as in our Theorem 3.1.
>
> 3. Therefore, in our work, we carefully design an additional _query hash_ (Algorithm 4) to decide when to update estimates of maintained points - illustrated in Figure 1. This additional data structure allows us to bound how often a query point is updated throughout the sequence of data point insertions, as the value of $\mu_\mathbf{q}$ changes throughout the sequence of updates (Lemma B.7).
>
> To address the reviewer's concern, we have added extra discussion at the end of page 5 (in blue) to better compare our approach to that of Liang et al.
>
> Importantly, although the actual update times of both results are qualitatively similar, our update time includes returning all the updated KDE estimates. Liang et al.’s update time only includes updating the buckets.
>
> This difference is crucial for the setting where $X=Q$, as is the case in our Theorem 4.1. Any new data point changes the true KDE value for all data/query points. However, our results show that for any data/query point insertion, only $\widetilde{O}(1)$ estimates have to be updated to maintain $(1\pm \varepsilon)$-approximations for every $\mathbf{q} \in Q$. This - combined with several other techniques - allows us to bound the number of sampling paths/edges that need to be resampled, leading to our efficient update times in Theorem 4.1.
>
> Our dynamic KDE experiments further illustrate this distinction. In the CKNS baseline, we first update the buckets with the new data points (as proposed in Liang et al.), and then recalculate the estimates for all query points after every data point update. On the other hand, our algorithm shows significant performance improvements by dynamically maintaining query estimates.
>
> As a final remark, we highlight that our work is the first to rigorously implement the static CKNS algorithm, and its adaptation to the dynamic setting.

---

> ### Author Response · Authors · 2024-11-18
>
> > **W2**: “Though the claim is that the update time is sublinear, the cost function depends on the value of the smallest query. For many reasonable kernels such as the Gaussian kernel, these queries have very small value, in which case the resulting update time is not small. On the other hand, for queries with large value, sampling is a standard KDE technique. Given these points, it is not clear what are the technical novelties of this paper.”
>
> **Response**: Our results indeed rely on the smallest KDE value. As is standard in the literature, we assume a lower bound for the smallest KDE value. For ease of presentation, we assume for every $\mathbf{q}$ it holds that $1 \leq \mu_\mathbf{q} \leq n$, but the results hold under the general condition that $\mu_\mathbf{q}/n = n^{-\theta(1)}$, like the work of Charikar et al. (2020).
>
> It’s important to note that this is the most interesting regime for this problem, since for $\mu_\mathbf{q}/n = n^{-\omega(1)}$ under the Orthogonal Vectors Conjecture [1], the problem cannot be solved faster than $n^{1-o(1)}$ time using space $n^{1-o(1)}$ [2]. And indeed, as noted by the reviewer, for larger values $\mu_\mathbf{q}/n = n^{-o(1)}$  random sampling solves the problem in $n^{o(1)}/\varepsilon^{2}$ time and space (Charikar et al., 2020).
>
>
> **References**
>
> [1] _Rubinstein, Aviad. "Hardness of approximate nearest neighbor search." Proceedings of the 50th annual ACM SIGACT symposium on theory of computing. 2018._
>
> [2] _Charikar, Moses, and Paris Siminelakis. "Multi-resolution hashing for fast pairwise summations." 2019 IEEE 60th Annual Symposium on Foundations of Computer Science (FOCS). IEEE, 2019._

---

> > ### Comment · Reviewer_GVfg · 2024-11-25
> >
> > Thanks for the response. Though this paper can handle the incremental setting where points are added to the dataset, I think there is also value in being able to replace points in the dataset, which is the scenario handled by Liang et. al. (2022).
> >
> > Looking at the author responses to other reviewers and the analysis within the paper, I agree there are non-trivial gaps in implementing the previous estimators for the static case, even given familiarity with standard techniques for implementing some sort of sampling on evolving datasets (as well as some other key statistics). However, it is not clear that these analyses are significant for the KDE literature. Perhaps the authors can elaborate on insights, structural properties, or techniques that might be of independent interest?

---

> > > ### Author Response · Authors · 2024-11-26
> > >
> > > We thank the reviewer for their feedback and taking the time to read our responses.
> > >
> > > > _" (...) there is also value in being able to replace points in the dataset, which is the scenario handled by Liang et. al. (2022)."_
> > >
> > > We agree that replacing points in a dataset is also a valuable setting. We do note that our qualitative KDE results are impossible to obtain in that setting for the general case.
> > >
> > > Suppose all query points are the same and their KDE is dominated by a single nearby data point. Then repeatedly placing this point at a distant location and then returning it to its original position forces updates for all query estimates at every update.  This results in an amortised update time of $|Q| \cdot \varepsilon^{-2} \cdot n^{o(1)} \cdot \mathrm{cost}(k)$ - equivalent to re-estimating all query points at every data point update. Our update time on the other hand is $\varepsilon^{-2} \cdot n^{o(1)} \cdot \mathrm{cost}(k)$.
> > >
> > > It would be interesting if a result similar to ours could be obtained with an extra condition on how far data points are allowed to move, however, that is beyond the scope of our work.
> > >
> > >
> > > >_"Perhaps the authors can elaborate on insights, structural properties, or techniques that might be of independent interest?"_
> > >
> > > We appreciate that the reviewer recognises the non-trivial aspects of our work, and we highlight the significance of our work below:
> > >
> > > - For the dynamic KDE result, the main insight is that the amortised update time for adding data points can be independent of the number of maintained query points, $|Q|$. To the best of our knowledge this is the first such result. This is because we show that using our query hash data structure, a query point only gets updated $ O(\log T)$ times after $ T $ updates (Lemma B.7, page 21).
> > >
> > > - For our dynamic approximate similarity result (Theorem 4.1), we use our dynamic KDE result for the setting where the set of data points is equal to the set of query points $ X = Q $. In this specific case, we use the structural property that any new data/query point updates $ \widetilde{O}(1) $ data/query points in expectation (Lemma 4.6). This key property, with additional required analysis for the graph construction, allows us to achieve our fast update times for the approximate similarity graph result.
> > >
> > > - We believe our designed technical procedures have broad potential applications in future work and could be of independent interest. For example, our dynamic KDE tree (Algorithms 7 to 10, pages 23–27) can maintain randomly sampled neighbors of a given vertex by edge weight for kernel graphs. This is a common primitive in many graph algorithms and could facilitate the design of dynamic variants of graph algorithms for kernel graphs (see e.g. Bakshi et al., 2023).
> > >
> > > **References**
> > >
> > > _Ainesh Bakshi, Piotr Indyk, Praneeth Kacham, Sandeep Silwal, and Samson Zhou. Subquadratic algorithms for kernel matrices via kernel density estimation. In 11th International Conference on Learning Representations (ICLR’23), 2023._

---

> ### Author Response · Authors · 2024-12-01
>
> Thank you once more for your feedback and comments on our work. As the review period is ending, we wanted to check if you’ve had a chance to read our response to your last question. We are happy to respond to any further questions you might have.

---

### Official Review · Reviewer_Qzjh · 2024-11-02

**Soundness:** 3
**Presentation:** 3
**Contribution:** 2
**Rating:** 5
**Confidence:** 3

**Summary:**

The paper studies the problem of approximately maintaining a dynamic similarity graph using kernel density estimation. A similarity graph over a set of points in $R^d$ weighs the edge between every pair of points $x,y$ by a similarity measure like the gaussian kernel $\exp(-|x-y|^2)$. Since the full graph is costly to compute and store, there is work on sparse approximations with approximate edge weights. Recently this line of work has focused on utilizing progress on hashing-based estimators for KDE.

This paper continues this line of work by allowing the similarity graph to be maintained dynamically under insertion and deletion of points.

**Strengths:**

Efficient construction of similarity graphs is an important problem, and the empirical results are favorable.

**Weaknesses:**

The paper seems technically somewhat incremental, in that there is not a lot of novelty needed in order to adapt existing work on the problem to the dynamic setting. The hashing-based estimators for KDE are naturally dynamically maintainable (just a point can simply be added or removed from its bucket in the hash table) and the paper basically builds this property into the Macgregor-Sun construction.

**Questions:**

N/A

---

> ### Author Response · Authors · 2024-11-18
>
> We thank the reviewer for their time and feedback.
>
> > **W1**: “The paper seems technically somewhat incremental, in that there is not a lot of novelty needed in order to adapt existing work on the problem to the dynamic setting. The hashing-based estimators for KDE are naturally dynamically maintainable (just a point can simply be added or removed from its bucket in the hash table) and the paper basically builds this property into the Macgregor-Sun construction.”
>
> **Response:**
> We emphasise that although the LSH-based _estimator_ of CKNS can be extended to a dynamic setting as data points get added (see e.g. Liang et al., 2022), it is not clear that the _estimates_ of a set of query points can be efficiently updated, i.e., where the update time is independent of the number of added query points, as in our Theorem 3.1. To obtain the amortised running time guarantees in Theorem 3.1, we carefully design an additional query hash to decide when to update estimates of points. In the analysis, one has to consider how often a query point $\mathbf{q}$ gets updated throughout the sequence of data point insertions/deletions (See Lemma B.7), as the value of $\mu_{\mathbf{q}}$ changes throughout the sequence of updates. This part of the discussion corresponds to the proofs found in Section B.2 (Pages 18 to Page 22)
>
> An important case is where $X=Q$, as is the case in our Theorem 4.1. Any new data point changes the true KDE value for all data/query points. However, our results show that for any data/query point insertion, only $\widetilde{O}(1)$ estimates have to be updated to maintain $(1\pm \varepsilon)$-approximations for every $\mathbf{q} \in Q$. Using this, we prove that the number of paths (edges) that need to be resampled is $\widetilde{O}(1)$ (Lemma 4.6). This requires careful analysis of the difference in the KDE estimate of $\mathbf{x}$ at the root and internal nodes of the sampling tree in order to bound the number of collisions (Lemmas 4.4, and 4.5) in the relevant buckets. This part of discussion corresponds to the proofs in Section C.2 from Page 25 to Page 30.
>
> Additionally, several novel analyses are needed on top of the static algorithm of Macgregor and Sun (2023) to ensure the sublinear update time of our approximate similarity graph: (i) we have to carefully track all the sampling paths corresponding to sampled edges in the tree; (ii) the reweighting factor of the sampled edges has to be changed (see Lines 350 - 355 for the discussion), in order to bound the number of edges whose weight needs to be updated if the degree/kde estimate of a neighbour changes (Lemma C.3).
>
> Taking all of this into account, we believe that our designed algorithms and their analyses form a significant contribution.

---

> > ### Comment · Reviewer_Qzjh · 2024-11-21
> >
> > Thank you for your replies.

---

> ### Author Response · Authors · 2024-11-25
>
> We appreciate the reviewer for taking the time to consider our response. There is one more point we would like to address in the original review, which states that our results hold "under insertion and deletion of points." We clarify that our KDE results are only stated for the incremental case of inserting data points. Our update times are impossible to obtain in the fully dynamic case.
>
> For instance, if all query points are the same, and their KDE is dominated by a single nearby data point, repeatedly inserting and deleting this point forces updates for all query estimates at every update. This results in an amortised update time of $|Q| \cdot \varepsilon^{-2} \cdot n^{o(1)} \cdot \mathrm{cost}(k)$ - equivalent to re-estimating all query points at every data point update. Our update time on the other hand is $\varepsilon^{-2} \cdot n^{o(1)} \cdot \mathrm{cost}(k)$.

---

> > ### Comment · Reviewer_Qzjh · 2024-11-27
> >
> > Thank you for this clarification. An further question is, could you please clarify the selection of the experimental setting in Table 2 and in the associated plots? For each algorithm and dataset we see some configuration of runtime and error in the table, but they are not compatible across the algorithms. Was a mutual parameter fixed here? Sorry if I've missed it, I do not see right now in the manuscript.

---

> > > ### Author Response · Authors · 2024-11-27
> > >
> > > Thank you for your question. Below, we clarify the experimental setup described in Table 2 and the associated plots.
> > >
> > >
> > > For each dataset, we set the parameter $\sigma$ of the Gaussian kernel such that the average kernel density $\mu_\mathbf{q} \cdot n^{-1} \approx 0.01$, This matches the experimental setup of previous work on KDE, including DEANN (Karppa et al., 2022). $\sigma$ can be considered the mutual parameter here (which varies across datasets). While we originally reported the specific used values of $\sigma$ in the Readme.md in the supplementary material, we acknowledge that this is not easy to find. Therefore we have updated Table 4 (page 32, changes in blue) in the manuscript to report our used values of $\sigma$.
> > >
> > > For the rest of the experimental setup, we split the data points into chunks of size 1,000 for aloi, msd, and covtype, and size 10,000 for glove and census. Then, we add one chunk at a time to both the set of data points $X$ and the set of query points $Q$. At each iteration, we evaluate the kernel density estimates $\hat{\mu}_\mathbf{q}$ produced by each algorithm with the relative error. Table 2 gives the total running time and final relative error for each algorithm, and in Figures 4 and 5 we show the relative errors and running times for all datasets and algorithms at every iteration.
> > >
> > > We describe this experimental setup on lines 482-513, however if the reviewer has any further suggestions to clarify the setup we will gladly update the manuscript.
> > >
> > > **References**
> > >
> > > _Matti Karppa, Martin Aumüller, and Rasmus Pagh. Deann: Speeding up kernel-density estima-
> > > tion using approximate nearest neighbor search. In 25th International Conference on Artificial
> > > Intelligence and Statistics (AISTATS’22), pp. 3108–3137, 2022._

---

> > > > ### Comment · Reviewer_Qzjh · 2024-11-27
> > > >
> > > > Thank you, though this was not quite my question. I meant to ask, what governs the trade-off between the running time and the accuracy in the tables and in plots. In the table, DynamicRS in most rows takes longer to run than yours but its error is smaller by 1-2 orders of magnitude. What governs this trade-off? For DynamicRS you could take less samples to speed it up while increasing the error. Is there a similar handle to govern the trade-off in your algorithm, and make it more accurate by paying with more running time?
> > > >
> > > > As another reviewer asks, is there a way to directly compare the algorithms by allotting them the same time to run and measuring the error they achieve within that time? Or allot them an error budget and measure how long they require to attain that error? Right now the algorithms seem incomparable in the table and in the plots.
> > > >
> > > > While it is too late in the rebuttal phase to add experiments, my question was how you chose the trade-off for each algorithm in these experiments to begin with.

---

> > > > > ### Author Response · Authors · 2024-11-28
> > > > >
> > > > > Thanks for clarifying the question.
> > > > >
> > > > > >_”Is there a similar handle to govern the trade-off in your algorithm, and make it more accurate by paying with more running time?”_
> > > > >
> > > > > There are number of parameters that govern the time/accuracy tradeoff in our algorithm
> > > > > 1. The error parameter $\varepsilon$.
> > > > > 2. The number of computed independent estimators $K_1$ (Line 7, Algorithm 1) - equivalent to the number of copies of the hash data structure.
> > > > > 3. The number of hash functions used in the hash table, $K_2$ (Line 6, Algorithm 2). A higher value reduces the variance of the estimates.
> > > > > 4. The assumption on the lower bound of $\mu_\mathbf{q}$. A smaller assumption on the lower bound gives more accurate estimates for query points with very small density estimates, at the cost of a longer preprocessing/query/update time complexity.
> > > > >
> > > > > > _”is there a way to directly compare the algorithms by allotting them the same time to run and measuring the error they achieve within that time? Or allot them an error budget and measure how long they require to attain that error?”_
> > > > >
> > > > > The only way to do that would be to perform an extensive grid search over the parameters + the error budgets. As our algorithm has so many hyperparameters this becomes computationally very expensive.
> > > > >
> > > > > >_”my question was how you chose the trade-off for each algorithm in these experiments to begin with”_
> > > > >
> > > > > For our dynamic KDE algorithm, we fixed the parameters to be $K_1 = \log(n)$ and $K_2 = \log(n) \cdot p^{-k_j}$, matching their theoretical asymptotic values while keeping the constants equal to $1$. We set the lower bound of $\mu_{\mathbf{q}}$ to be $1000 / n$. We found with some initial testing that these values gave a reasonable trade-off between running time and accuracy. We fixed the parameters of the CKNS algorithm to be the same as ours, and chose a random sampling probability of $0.1$ for the DynamicRS algorithm.
> > > > > Note that we did not optimise the algorithms’ parameters for the datasets in the experiments. Rather, we fixed the parameters for each algorithm and then reported their performance. In our view, this provides a fair comparison since it is not possible to optimise algorithms’ parameters for specific datasets when applying them in practice.
> > > > >
> > > > > As we replied to reviewer bEJW, the main point of our algorithm is that it has a provable approximation guarantee. On the other hand, for the DynamicRS baseline, there is only a guarantee if sufficiently many samples are drawn with respect to the smallest KDE value in the dataset. So while DynamicRS performs very well in many cases, it suffers significant degradation on the msd dataset, while our algorithm (without doing any hyperparameter grid search) has stable performance across the data sets.

---

> > > > > > ### Comment · Reviewer_Qzjh · 2024-12-01
> > > > > >
> > > > > > Thank you for your answers, I have read them. I believe the point stands that the comparison presented between the algorithm is not direct and does not clearly show an advantage of one over the other, and the fact your method has multiple parameters with no guidance for selecting them other than an exhaustive grid search is a considerable hindrance to its usability. If you intend the empirical evaluation to convey that your method is practically viable and preferable to existing methods, I encourage you to put more thought into the matter of hyperparameter selection.

---

> > > > > > > ### Author Response · Authors · 2024-12-04
> > > > > > >
> > > > > > > Thank you for your feedback on hyperparameter selection. For this work, our primary contributions are theoretical, focusing on rigorously proving the approximation guarantees and the sublinear update times of our algorithms. We did not try to optimize experimental performance, but we do believe our experiments highlight the sublinear update times of our algorithm. Furthermore, the difficulty of hyperparameter selection is not unique to our algorithm as it is a known challenge in the static setting. See e.g., Karppa et al., (2022) and Siminelakis et al. (2019). As a final remark, we note that our parameter settings do perform well in our experiments on dynamic similarity graphs.
> > > > > > >
> > > > > > >
> > > > > > > **References**
> > > > > > > _Matti Karppa, Martin Aumüller, and Rasmus Pagh. Deann: Speeding up kernel-density estimation using approximate nearest neighbor search. In 25th International Conference on Artificial Intelligence and Statistics (AISTATS’22). 2022._
> > > > > > >
> > > > > > > _Siminelakis, Paris, Kexin Rong, Peter Bailis, Moses Charikar, and Philip Levis. Rehashing kernel evaluation in high dimensions. In 36th International Conference on Machine Learning (ICML’2019). 2019._

---

> > > > > ### Author Response · Authors · 2024-12-01
> > > > >
> > > > > Thank you again for engaging in the review period and for your feedback and questions. As the review period is ending, we wanted to check if you’ve had a chance to read our response to your last questions. We are happy to respond to any further questions you might have.

---

### Official Review · Reviewer_bEJW · 2024-11-05

**Soundness:** 3
**Presentation:** 3
**Contribution:** 3
**Rating:** 6
**Confidence:** 3

**Summary:**

Overview: The paper presents efficient dynamic data structures for two kernel problems over a dynamic set of points X and a kernel function k(.,.):
-	Kernel density estimation (KDE) problem, where the goal is to maintain a set of points X and a set of queries Q so that for each q in Q the data structure holds an estimate of the sum_{x in X} k(x,q), and
-	Kernel sparsification, where the goal is to maintain X and a sparse weighted graph G with vertices in X such that clustering the graph G produces approximately the same result as clustering the fully connected graph with edge weights induced by k(.,.).

Techniques:  Both data structures use recently developed algorithms for the *static* KDE problem. The KDE data structure is obtained by “dynamizing” the algorithm of Charikar et al, 2020, while the kernel sparsification data structure follows the algorithm of Macgregor and Sun, 2023.

The authors complement the theoretical development with empirical evaluation of their algorithms. For KDE, the proposed algorithm is compared on a few benchmark data sets, to the following baselines:

 (a) EXACT: an algorithm that recomputes the kernel values after every update to X using the naive (quadratic-time) algorithm.

 (b) DYNAMICRS: an algorithm that (I am guessing) solves the static problem exactly on a random sample containing 10% of the points, after every update.

(c) CKNS: an algorithm that recomputes the estimates after every update using the static algorithm of Charikar et al 2020.

For each data set and each algorithm, the authors report running times and estimation errors.

**Strengths:**

S1: Both problems have been studied extensively, and they have many applications in machine learning. The dynamic variants of those problems have been studied less, but they are well-motivated and important as well.

S2: see below.

**Weaknesses:**

W1: The experimental design is a good starting point, but it makes it impossible to answer the following important question: for a given data set and a given error bound, which algorithm is the fastest? For example, for the GLOVE data set, the proposed algorithm is twice as fast as DYNAMICRS, but it also has a much higher error, so it is plausible that DYNAMICRS with a smaller sampling rate would be faster *and* have smaller error. Overall, the experimental section demonstrates that the proposed algorithm is implementable, but it is unclear whether it improves empirically over the state of the art.

W2/S2: Converting static algorithms into dynamic data structures does not seem to require significantly new ideas, but it is not trivial either.

**Questions:**

What are the main new algorithmic ideas needed to dynamize the static algorithms of Charikar et al and Macgregor-Sun ?

---

> ### Author Response · Authors · 2024-11-18
>
> We thank the reviewer for their time and feedback, and we respond to their points/question below.
>
> > **W1**: “The experimental design is a good starting point, but it makes it impossible to answer the following important question: for a given data set and a given error bound, which algorithm is the fastest? For example, for the GLOVE data set, the proposed algorithm is twice as fast as DYNAMICRS, but it also has a much higher error, so it is plausible that DYNAMICRS with a smaller sampling rate would be faster and have smaller error. Overall, the experimental section demonstrates that the proposed algorithm is implementable, but it is unclear whether it improves empirically over the state of the art. ”
>
> **Response:**
> In practice, the objective is to approximate the KDE efficiently without computing the true KDE values, as this is computationally expensive. A key strength of our algorithm is that it has a provable approximation guarantee. On the other hand, for the DynamicRS baseline, there is only a guarantee if sufficiently many samples are drawn with respect to the smallest KDE value in the dataset. Determining this fraction in practice is not clear and dataset-dependent. For instance, while DynamicRS performs very well in many cases, it suffers significant degradation on the msd dataset. In contrast, our algorithm has stable performance across all tested datasets.
>
>
> > **S2/W2/Q1**: “Converting static algorithms into dynamic data structures does not seem to require significantly new ideas, but it is not trivial either.” & “What are the main new algorithmic ideas needed to dynamize the static algorithms of Charikar et al and Macgregor-Sun ?”
>
> **Response:**
>
> We list the new algorithmic ideas below.
>
> The key algorithmic idea in our Dynamic KDE result is the introduction of an additional _query hash_ (Algorithm 4) to decide when to update estimates of maintained points - illustrated in Figure 1. We emphasise that although the LSH-based _estimator_ of CKNS can be extended to a dynamic setting as data points get added (see e.g. Liang et al., 2022), it is not as clear that the _estimates_ of a set of query points can be efficiently updated, i.e., where the update time is independent of the number of added query points, as in our Theorem 3.1. To obtain the amortised running time guarantees in Theorem 3.1, we therefore carefully design the additional query hash to decide when to update estimates of points. In the analysis, one then has to consider how often a query point $\mathbf{q}$ gets updated throughout the sequence of data point insertions/deletions (See Lemma B.7), as the value of $\mu_{\mathbf{q}}$ changes throughout the sequence of updates. This part of the discussion corresponds to the proofs found in Section B.2 (Pages 18 to Page 22)
>
> An important case is where $X=Q$, as is the case in our Theorem 4.1. Any new data point changes the true KDE value for all data/query points. However, our results show that for any data/query point insertion, only $\widetilde{O}(1)$ estimates have to be updated to maintain $(1\pm \varepsilon)$-approximations for every $\mathbf{q} \in Q$. Using this, we prove that the number of paths (edges) that need to be resampled is $\widetilde{O}(1)$ (Lemma 4.6). This requires a careful analysis of the difference in the KDE estimate of $\mathbf{x}$ at the root and internal nodes of the sampling tree in order to bound the number of collisions (Lemmas 4.4, and 4.5) in the relevant buckets. This part of discussion corresponds to the proofs in Section C.2 from Page 25 to Page 30.
>
> Additionally, several novel analysis are needed on top of the static algorithm of Macgregor and Sun (2023) to ensure the sublinear update time of our approximate similarity graph: (i) we have to carefully track all the sampling paths corresponding to sampled edges in the tree; (ii) the reweighting factor of the sampled edges has to be changed (see Lines 350 - 355 for the discussion), in order to bound the number of edges whose weight needs to be updated if the degree/kde estimate of a neighbour changes (Lemma C.3).
>
> Taking all of this into account, we believe that our designed algorithms and their analyses form a significant contribution.

---

> > ### Comment · Reviewer_bEJW · 2024-11-26
> > **Thank you for the answers**
> >
> > Thank you for elaborating on the technical challenges that needed to be overcome in order to "dynamize" the KDE data structure. I am comfortable with the paper being accepted.

---

### Official Review · Reviewer_Rhdf · 2024-11-08

**Soundness:** 3
**Presentation:** 3
**Contribution:** 3
**Rating:** 8
**Confidence:** 3

**Summary:**

The paper develops a technique to maintain the kernel density estimates arising from a set of data points X at a set of query points Q, as updates occur dynamically to X and Q. Using this, they design a dynamic data structure that can maintain a sparse approximation to the "kernel weighted" similarity graph for a set of points X (as X changes).

The main idea is to adapt the approach of Charikar, Kapralov, Nouri, and Siminelakis (CKNS) to handle dynamic updates to both the X and the Q. For the application to approximating similarity graphs, the paper builds off the recent work of Macgregor & Sun (2023).

**Strengths:**

- Kernel density estimation is a ubiquitous problem in data analysis, and dynamic models are also interesting to study. The update guarantees that the paper obtains are quite strong.

- In spite of being based on prior work, there is sufficient technical novelty in the algorithms.

- The application to dynamic similarity matrix estimation is also interesting.

**Weaknesses:**

No significant weaknesses; please see the questions below.

**Questions:**

- The paper cites the work of Macgregor and Sun 2023 for dynamic similarity matrices (and the algorithm is based on that work), but I didn't notice experimental comparison to that work. I may have missed it, but if the authors can provide a clear comparison (both experimentally and theoretically), it would make the paper stronger.

- About the n^{1/4} time required for the KDE step for Gaussian kernels (which is what leads to the final update time), are there any lower bounds suggesting that the time cannot be improved?

---

> ### Author Response · Authors · 2024-11-18
>
> We thank the reviewer for their time and positive evaluation of our paper. We respond to the questions below.
>
> > **Q1**: “The paper cites the work of Macgregor and Sun 2023 for dynamic similarity matrices (and the algorithm is based on that work), but I didn't notice experimental comparison to that work. I may have missed it, but if the authors can provide a clear comparison (both experimentally and theoretically), it would make the paper stronger.”
>
> **Response:** The work of Macgregor and Sun (2023) is a static algorithm for approximate similarity graph construction. Our result generalises their algorithm to the dynamic setting.
>
> Regarding the theoretical comparison, we achieve the same guarantees on the returned approximate similarity graph as their work. One notable difference is that their algorithm can use any KDE estimator as a black box, whereas our algorithm requires our dynamic KDE algorithm.
>
> Regarding an experimental comparison, directly comparing our dynamic method to their static approach would involve re-running their algorithm after each data point update, which is computationally intensive. Given that the theoretical guarantees for the both approaches are the same, we expect that both algorithms would yield similarity graphs with similar performance in terms of NMI and ARI. Our focus is to demonstrate the efficiency gains in the dynamic context. However, if the reviewer believes that including this experiment would improve the paper, we are willing to conduct this experiment and include the findings.
>
> > **Q2**: “About the n^{1/4} time required for the KDE step for Gaussian kernels (which is what leads to the final update time), are there any lower bounds suggesting that the time cannot be improved?”
>
> **Response:** This is an interesting question, and to the best of our knowledge there is no known running time lower bound for KDE estimation for the Gaussian kernel, in both the static and dynamic setting.
>
> In their paper, Charikar et al (2020) do provide a KDE estimator which uses data dependent LSH, and they achieve a KDE query time of $n^{0.173}$ for the Gaussian kernel in the static setting; generalising this result to the dynamic setting would be an interesting direction for future work.

---

### Author Response · Authors · 2024-11-18

Dear reviewers, thank you all for your feedback. We have revised the paper pdf based on your reviews and highlighted the changes in blue.

---

### Author Response · Authors · 2024-11-24
**End of discussion period**

Dear reviewers, with the discussion period ending soon, we’d really value your feedback on our responses - let us know if there’s anything else we should clarify or fix. Thank you again for your time.

---

### Meta-Review · Area_Chair_SyEV · 2024-12-23

**Metareview:**

This paper creates similarity graphs using fast LSH based data structures to do KDE dynamically. They show how this can translate to a fast dynamic spectral clustering method. Reviews are mixed, with the highest score being a very brief and shallow review, and the most negative reviewer also conceding there are some nontrivial parts of this work. My own view is that it is somewhat unsurprising combination of existing ideas, though the implementation is well executed. I would have liked to see more reviewer discussion, but overall this paper could benefit from a deeper analysis or stronger baselines as it is a borderline submission. I hope the reviewer remarks will help in a stronger resubmission

**Additional Comments On Reviewer Discussion:**

The positive review contained zero information, and reviewers did not engage in discussion.

---

### Decision · Program_Chairs · 2025-01-22

Reject